# Risk Bounds For Distributional Regression

**Carlos Misael Madrid Padilla**
Department of Statistics and Data Science
Washington University in St Louis
St Louis, MO 63130
`carlosmisael@wulst.edu`

**Oscar Hernan Madrid Padilla**
Department of Statistics
University of California, Los Angeles
Los Angele, CA 90095
`oscar.madrid@stat.ucla.edu`

**Sabyasachi Chatterjee**
Department of Statistics
University of Illinois at Urbana-Champaign
Champaign, IL 61820
`sc1706@illinois.edu`

## Abstract

This work examines risk bounds for nonparametric distributional regression estimators. For convex-constrained distributional regression, general upper bounds are established for the continuous ranked probability score (CRPS) and the worst-case mean squared error (MSE) across the domain. These theoretical results are applied to isotonic and trend filtering distributional regression, yielding convergence rates consistent with those for mean estimation. Furthermore, a general upper bound is derived for distributional regression under non-convex constraints, with a specific application to neural network-based estimators. Comprehensive experiments on both simulated and real data validate the theoretical contributions, demonstrating their practical effectiveness.

## 1   Introduction

While regression methods are widely popular across statistics and machine learning, it is well recognized that the conditional mean alone often fails to capture the full relationship between a response variable and a set of covariates. As noted by Shaked and Shanthikumar [57], "the ultimate goal of regression analysis is to obtain information about the conditional distribution of a response given a set of explanatory variables." A common framework is quantile regression, which estimates conditional quantiles to provide a more detailed view of the response distribution [35]. A more direct approach is distributional regression, which estimates the conditional distribution of the response given the covariates.

Distributional regression has found applications in diverse areas, including electricity spot price analysis [33], understanding income determinants [34], modeling weather data [63], and improving precipitation forecasts [28, 55].

The estimation of distributions of random variables under structural constraints is a fundamental problem in many statistical and machine learning tasks, including nonparametric regression, density estimation, and probabilistic forecasting [26, 62, 23]. Consider the sequence distributional model, in which we observe independent random variables $y_1, \ldots, y_n \in \mathbb{R}$, each drawn from an unknown distribution $F_1^*, \ldots, F_n^*$, respectively. The objective of this paper is to estimate the vector $F^*(t) = (F_1^*(t), \ldots, F_n^*(t))^\top$ for $t \in \mathbb{R}$. Here, $F_i^*(t) = \mathbb{P}(y_i \leq t)$ represents the cumulative distribution function (CDF) at a specified $t$ for each observation $y_i$.

39th Conference on Neural Information Processing Systems (NeurIPS 2025).

In this paper, we will explore different structural constraints to estimate $F^*(t)$. These constraints not only ensure interpretable and robust estimators but also prevent overfitting, making them essential in domains such as signal processing, medical diagnostics, and probabilistic weather forecasting. For example, in survival analysis, monotonicity reflects the cumulative nature of survival probabilities, while in genomics, smoothness helps capture gradual trends in gene expression data [3, 62]. To rigorously evaluate the quality of the estimators, we employ the continuous ranked probability Score (CRPS), a widely used metric for assessing the accuracy of probabilistic forecasts [22]. By quantifying the distance between the estimated and true CDFs, CRPS provides an interpretable and robust framework for comparing estimators under various structural constraints.

## 1.1 Summary of Results

We now provide a brief summary of the contributions in this paper.

**Unified Framework for Estimation:** We study a unified framework for estimating $F^*(t)$ by minimizing a quadratic loss over a convex set $K_t \subset \mathbb{R}$. The convex set $K_t \subset \mathbb{R}$ enforces structural constraints on the parameter $F^*(t)$. For the resulting estimator, we provide rigorous theoretical guarantees, including non-asymptotic bounds on both the mean squared error (MSE) and the CRPS.

**Applications to Monotonicity and Bounded Variation:** We demonstrate the applicability of our framework to convex constraints arising in monotonicity [9, 2, 12] and bounded total variation [42, 62, 23]. These examples illustrate the flexibility and practical utility of the proposed approach for structured parameter estimation.

**General Theory for Distributional Regression:** We establish a general theory for distributional regression with constraints encoded by arbitrary sets. The main result provides a uniform bound on the empirical $\ell_2$ error of $F^*(t)$ across $t$, with the upper bound explicitly dependent on the approximation error and the complexity of the sets $K_t$.

**Convergence Rates for Neural Networks:** Exploiting the general results for arbitrary sets, we derive convergence rates for distributional regression using dense neural networks. This extends the framework of Kohler and Langer [37] to the context of distributional regression.

## 1.2 Other Related Work

Distributional regression involves modeling the cumulative distribution function (CDF) of a random variable, whereas quantile regression focuses on estimating the inverse CDF. Koenker et al. [36] provides a comprehensive review of the distinctions between these two approaches. For instance, if the outcome variable is income and the covariate is a binary variable representing educational attainment, distributional regression would model the probability that income falls below a certain threshold for each educational group. In contrast, quantile regression would estimate income differences between individuals ranked at the same quantile within the two groups. For further discussion on these differences, see also Peracchi [50]. The approach we adopt in this paper is based on modeling the mean of the random variables of the form $1_{\{y_i \leq t\}}$, which represent the indicator of the events $\{y_i \leq t\}$ for $i = 1, \ldots, n$. This idea was first introduced in [20] and has since been explored in various contexts, including Firpo and Sunao [19], Rothe [52], Rothe and Wied [53], and Chernozhukov et al. [15]. More recently, distributional regression has been studied in diverse settings, such as isotonic regression [28], random forests [10], and neural networks [58, 31]. In particular, Henzi [27] develop consistent estimators of conditional CDFs under increasing concave and convex stochastic orders, providing a flexible framework that accommodates heterogeneous variance structures and distributional crossings—situations where traditional stochastic dominance assumptions may fail. These works highlight the versatility and applicability of distributional regression across a range of methodologies and problem domains. Finally, other nonparametric approaches to distributional regression include Dunson et al. [17] and Hall et al. [24].

## 2 Notation

Througout, for a vector $v \in \mathbb{R}^n$, we denote by $\|v\|$ and $\|v\|_1$ the $\ell_2$ and $\ell_1$ norms, respectively. Thus, $\|v\| = \sqrt{\sum_{i=1}^n v_i^2}$ and $\|v\|_1 = \sum_{i=1}^n |v_i|$. Furthermore, given two functions $G, H : \mathbb{R} \to \mathbb{R}$, we define $\mathrm{CRPS}(G, H) = \int_{-\infty}^{\infty} (G(t) - H(t))^2 dt$. Also, for for $\eta > 0$ and $v \in \mathbb{R}^n$, we write

$B_\eta(v) := \{u \in \mathbb{R}^n : \|v - u\| \leq \eta\}$. For a metric space $(\mathcal{X}, d)$, let $K$ be a subset of $\mathcal{X}$, and $r > 0$ be a positive number. Let $B_r(x, d)$ be the ball of radius $r$ with center $x \in \mathcal{X}$. We say that a subset $C \subset \mathcal{X}$ is an $r$-external covering of $K$ if $K \subset \cup_{x \in C} B_r(x, d)$. Then external covering number of $K$, written as $N(r, K, d)$, is defined as the minimum cardinality of any $r$-external covering of $K$. Furthermore, for a function $f : \mathcal{X} \to \mathbb{R}$, we define its $\ell_\infty$ norm as $\|f\|_\infty := \sup_{x \in \mathcal{X}} |f(x)|$. Also, for two sequences $a_n$ and $b_n$, we write $a_n \lesssim b_n$ if $a_n \leq cb_n$ for a positive constant $c$, and if $a_n \lesssim b_n$ and $b_n \lesssim a_n$, then we write $a_n \asymp b_n$. The indicator function of a set $A$ is denoted as $1_A(t)$, which takes value 1 if $t \in A$ and 0 otherwise. Finally, the indicator of an event $A$ is $1_A$ which takes value 1 if $A$ holds and 0 otherwise.

## 2.1 Outline

The paper is organized as follows: Section 3 establishes a unified framework for distributional regression under structural constraints. In particular, Section 3.1 introduces the general methodology. Section 3.2 derives statistical risk guarantees for the convex case. Sections 3.2.1 and 3.2.2 provide concrete examples of convex estimators, focusing on isotonic regression and trend filtering, respectively. Section 3.3 extends the framework to the non-convex setting. Specifically, Section 3.3.1 develops the general theory for non-convex estimators, and Section 3.3.2 explores an estimator based on deep neural networks. Section 4 and 5 present simulation studies and real-data experiments, respectively, comparing the proposed methods to state-of-the-art competitors. Finally, Section 6 concludes with a discussion of future research directions, and the Appendix provides additional theoretical results and experimental findings, including two additional real-data applications.

## 3 Theory

### 3.1 General Result for Constrained Estimators

We begin this section by addressing general problems in distributional regression under structural constraints. Using the notation introduced in Section 1, our goal is to estimate the vector of distribution functions evaluations $F^*(t)$. To achieve this, we adopt an empirical risk minimization framework based on the continuous ranked probability score (CRPS), a widely used tool for evaluating distributional forecasts [43, 22]. Specifically, we consider the estimator

$$\widehat{F} := \underset{\{F_i\}_{i=1}^n \,:\, F(t) \in K \text{ for all } t}{\arg\min} \sum_{i=1}^n \mathrm{CRPS}(F_i, 1_{\{y_i \leq \cdot\}}), \tag{1}$$

where $F(t) = (F_1(t), \ldots, F_n(t))^\top \in \mathbb{R}^n$, and $K \subset \mathbb{R}^n$ is a set encoding the structural constraint. Related CRPS-based formulations have been proposed in the context of isotonic distributional regression [28].

This formulation can be viewed as a form of M-estimation over function-valued parameters, where the loss is defined via a proper scoring rule. Specifically, the estimator $\widehat{F}$ minimizes the empirical CRPS loss, computed relative to the observed point masses $1_{\{y_i \leq \cdot\}}$. Because CRPS is strictly proper [22], the population version of this loss—where expectations are taken over the distribution of each $y_i$—is minimized at the true distribution functions $F_i^*$. Hence, the estimator in (1) can be interpreted as an empirical risk minimizer, and is statistically and decision-theoretically justified. The constraint $F(t) \in K$ for all $t$ imposes additional structure on the estimator, promoting regularity and interpretability. We now show that the solution to the empirical CRPS minimization problem can be obtained via a simple projection estimator. For each $t \in \mathbb{R}$, define $w(t) = (1_{\{y_1 \leq t\}}, \ldots, 1_{\{y_n \leq t\}})^\top \in \mathbb{R}^n$, and consider the projection

$$\widehat{F}(t) := \underset{\theta \in K}{\arg\min} \left\{ \|w(t) - \theta\|^2 \right\}. \tag{2}$$

**Lemma 1.** *For $K \subset \mathbb{R}^n$, the function-valued estimator defined by (2), with $\widehat{F}_i(t) := [\widehat{F}(t)]_i$, solves the empirical CRPS minimization problem in (1).*

Thus, for each $t \in \mathbb{R}$, the solution $\widehat{F}(t)$ in problem (1) is obtained by projecting the empirical vector $w(t)$ onto the convex set $K$, making the estimation process both computationally efficient and conceptually transparent. In the more general case where the structural constraint may vary with $t$,

i.e., $K_t \subset \mathbb{R}^n$, we extend the estimator to

$$\widehat{F}(t) := \arg\min_{\theta \in K_t} \left\{ \|w(t) - \theta\|^2 \right\}. \tag{3}$$

This projection-based formulation allows us to define flexible estimators tailored to varying structural assumptions.

## 3.2 Risk bounds for convex case

In this subsection, we focus on the special case where each constraint set $K_t \subset \mathbb{R}^n$ is convex and satisfies $F^*(t) \in K_t$. As described in (3), the corresponding estimator $\widehat{F}(t)$ is obtained by projecting the empirical vector $w(t)$ onto $K_t$, which amounts to minimizing the empirical $\ell_2$-loss under convex constraints.

Our next result provides an upper bound on the expected value of the CRPS error. This is a consequence of a modified version of Theorem A.1 in Guntuboyina et al. [23], see Theorem 5 in the Appendix. While our result is conceptually related to Theorem A.1 in Guntuboyina et al. [23], our setting and proof strategy differ in several important ways; see Appendix E for a detailed discussion comparing both results.

**Theorem 1.** *Suppose that* $(0, \ldots, 0), (1, \ldots, 1) \in K_t$, *and*

$$\mathbb{P}(y_i \in \Omega) = 1, \;\; \text{for all} \;\; i, \tag{4}$$

*and for some fixed compact set* $\Omega \subset \mathbb{R}$. *Then, there exists a constant* $C > 0$ *such that*

$$\mathbb{E}\left( \frac{1}{n} \sum_{i=1}^n \text{CRPS}(\widehat{F}_i, F_i^*) \right) \leq \frac{C\eta^2}{n} \tag{5}$$

*for every* $\eta > 1$ *satisfying*

$$\sup_{t \in \mathbb{R}} \mathbb{E}\left[ \sup_{\theta \in K_t \,:\, \|\theta - F^*(t)\| \leq \eta} g^\top (\theta - F^*(t)) \right] \leq \frac{\eta^2}{L} \tag{6}$$

*where* $g \sim N(0, I_n)$, *for a universal constant* $L > 0$.

Thus, Theorem 1 shows that bounding the expected CRPS error can be reduced to analyzing the local Gaussian complexity, as given on the left-hand side of Equation (6).

As a direct consequence of Theorem 1, we can derive the same upper bound as in (5) for a rearrangement (see e.g. Lorentz [39], Bennett and Sharpley [6]) of a truncated version of $\widehat{F}_i$, which is non-decreasing by construction. This result is formalized in the following theorem.

**Corollary 1.** *Suppose that* $(0, \ldots, 0), (1, \ldots, 1) \in K_t$ *for all* $t \in \mathbb{R}$, *and that* $F_i^*(\cdot)$ *is continuous for all* $i$. *Given* $i \in \{1, \ldots, n\}$, *let* $\widehat{F}_i^+(t) = \max\{0, \widehat{F}_i(t)\}$. *Let* $y_{(1)} \leq \ldots \leq y_{(n)}$ *be the order statistics of* $y$. *Define* $a_{i,j} = \widehat{F}_i^+(y_{(j)})$ *for* $j = 1, \ldots, n - 1$ *and sort the vector* $a_{i,\cdot}$ *as*

$a_{i,j_1} \geq \ldots \geq a_{i,j_{n-1}}$, *and let* $\widetilde{F}_i$ *be defined as* $\widetilde{F}_i(t) = \begin{cases} 0 & \text{if } t < y_{(1)} \\ \sum_{l=1}^{n-1} a_{i,j_l} 1_{[v_l, v_{l-1})}(t) & \text{if } y_{(1)} \leq t < y_{(n)} \\ 1 & \text{if } y_{(n)} \leq t, \end{cases}$

*where* $v_0 = y_{(n)}$ *and* $v_l = y_{(n)} - \sum_{k=1}^{l} (y_{(j_k + 1)} - y_{(j_k)})$ *for* $l = 1, \ldots, n - 1$. *Then, with the notation from Theorem 1, we have that* $\mathbb{E}\left( \frac{1}{n} \sum_{i=1}^n \text{CRPS}(\widetilde{F}_i, F_i^*) \right) \leq \frac{C\eta^2}{n}$.

The function $\widetilde{F}_i(t)$ defined above is non-decreasing and is constructed by modifying the original estimator $\widehat{F}_i(t)$ through a rearrangement of its values. Specifically, this is achieved by applying a change of variable to $\widehat{F}_i(t)$, followed by the Hardy–Littlewood decreasing rearrangement. This classical construction [25] preserves the level set measures and minimizes the $L^2$ distance among all decreasing equimeasurable functions. An example of $\widetilde{F}_i(t)$ and $\widehat{F}_i(t)$ is given in Figure 2 in the Appendix. We now present the final result from this section.

**Theorem 2.** *Suppose that $K_t \subset K$ for all t. For any $\eta > 0$ it holds that*

$$\mathbb{P}\left(\sup_{t \in \mathbb{R}} \sum_{i=1}^{n} \left(\widehat{F}_i(t) - F_i^*(t)\right)^2 > 2\eta^2\right) \leq \frac{C}{\eta^2} \int_0^{\eta/4} \sqrt{\log N(\varepsilon, (K-K) \cap B_\eta(0), \|\cdot\|)} d\varepsilon + \frac{C\sqrt{\log n}}{\eta}, \tag{7}$$

*for a positive constant C.*

Theorem 2 provides a high-probability concentration bound for the MSE in estimating $F^*(t)$, holding uniformly over all $t \in \mathbb{R}$. This sets it apart from standard sub-Gaussian bounds, which typically yield control at a fixed evaluation point $t$, see for example [5, 14]. To ensure that the bound in Theorem 2 is small, it suffices to upper bound the local entropy of $K - K$, where $K$ is an upper set that contains the sets $K_t$.

### 3.2.1 Isotonic Regression

In this subsection, we present the first application of our general theory from Section 3, focusing on distributional isotonic regression—a topic that has garnered significant attention in the literature [16, 18, 29, 32]. The most relevant works to our results are Mösching and Dümbgen [44], which examined distributional isotonic regression under smoothness constraints, and Henzi et al. [28], which proposed an interpolation method equivalent to the formulation in (3) with $K_t$ enforcing a monotonicity constraint.

Setting

$$K_t = K := \{\theta \in \mathbb{R}^n : \theta_1 \leq \theta_2 \leq \ldots \leq \theta_n\}, \tag{8}$$

we now consider the case of isotonic distributional regression assuming that $F^*(t) \in K_t$, which is equivalent to $\mathbb{P}(Y_1 \leq t) \leq \ldots \leq \mathbb{P}(Y_n \leq t)$. With the constraint sets as in (8), the resulting estimator in (3) can be found with the pool adjacent violators algorithm from Robertson [51].

**Corollary 2.** *Consider the estimators $\{\widehat{F}_t\}_{t \in \mathbb{R}}$ defined in (3) with $K_t$ defined as (8). If $F^*(t) \in K$ and (4) holds for some fixed compact set $\Omega$, then*

$$\mathbb{E}\left(\frac{1}{n}\sum_{i=1}^{n} \mathrm{CRPS}(\widehat{F}_i, F_i^*)\right) \leq Cn^{-2/3}, \tag{9}$$

*for some positive constant $C > 0$. Moreover, (9) holds replacing $\widehat{F}$ with the corresponding $\widetilde{F}$ as defined in Corollary 1, provided that each function $F_i^*$ is continuous. Finally,*

$$\sup_{t \in \mathbb{R}} \sum_{i=1}^{n} \frac{1}{n}\left(\widehat{F}_i(t) - F_i^*(t)\right)^2 = O_{\mathbb{P}}\left(\frac{1}{n^{2/3}} + \frac{\log n}{n}\right), \tag{10}$$

*where (10) holds without requiring (4) nor continuity of the $F_i^*$'s.*

The result in Corollary 2 establishes that distributional isotonic regression achieves an estimation rate of $n^{-2/3}$ for both the expected average CRPS and the worst-case MSE, as shown in (10). This result improves upon Theorem 3 in Henzi et al. [28] in the univariate case, which only demonstrated convergence in probability for isotonic distributional regression. However, we emphasize that Henzi et al. [28] study the more general setting of multivariate covariates, while our analysis is restricted to the univariate case ($d = 1$).

To further contextualize our theoretical guarantees, we compare them with Theorem 3.3 in Mösching and Dümbgen [44], which establishes uniform consistency for estimating the conditional distribution function. Considering the fixed design case in their formulation and assuming without loss of generality that in their notation $X_1 < \ldots < X_n \in \mathbb{R}$, the goal is to estimate an unknown family of distributions $(F_x)_{x \in \mathbb{R}}$, where for each fixed $t \in \mathbb{R}$, the map $x \mapsto F_x(t)$ is assumed to be non-decreasing and $\alpha$-Hölder continuous with constant $C > 0$ that is the same across $t$. Additionally, the design is assumed to be asymptotically dense—i.e., the covariate values $X_1 < \cdots < X_n$ sufficiently cover the domain. Translating their setup into our notation, their target $F_{X_i}(t) := \mathbb{P}(Y_i \leq t \mid X_i)$ corresponds to $F_i^*(t) := \mathbb{P}(y_i \leq t)$ in our sequence model. Their assumption that $x \mapsto F_x(t)$ is non-decreasing for each $t$ aligns with our isotonic regression framework, where we impose monotonicity of the sequence $F_1^*(t) \leq \cdots \leq F_n^*(t)$. However, in contrast to their setting, we do not require any smoothness assumptions such as Hölder continuity on the sequence $\{F_i^*(t)\}_{i=1}^{n}$, nor do we require

the covariates to be dense. Despite that, our method achieves a faster convergence rate of order $n^{-2/3}$ for both the average CRPS risk and the worst-case MSE, compared to their rate $n^{-2\alpha/(2\alpha+1)}$ (up to logarithmic factors) for the worst-case MSE, with $\alpha \in (0, 1]$, established under their stronger regularity assumptions. A summary of these comparisons is provided in Table 2 in Appendix E.

We also show in Appendix A that faster rates are achievable under additional structural assumptions on $F^*(t)$. Specifically, if $F^*(t)$ has few strict increases—e.g., if it is piecewise constant with a small number of jumps—then the estimator can attain nearly parametric risk rates up to logarithmic factors.

### 3.2.2 Trend Filtering

In this subsection, we apply the theory from Section 3 to distributional regression under a total variation constraint. Total variation-based methods were independently introduced by Rudin et al. [54], Mammen and Van De Geer [42], and Tibshirani et al. [61]. These methods have been extensively studied in various contexts within the statistics literature, including univariate settings [62, 38, 23, 41, 46], grid graphs [30, 11], and general graphs [65, 48].

Before establishing our proposed total variation estimators, we introduce some additional notation. For a vector $\theta \in \mathbb{R}^n$, define $D^{(0)}(\theta) = \theta$, $D^{(1)}(\theta) = (\theta_2 - \theta_1, \ldots, \theta_n - \theta_{n-1})^\top$ and $D^{(r)}(\theta)$, for $r \geq 2$, is recursively defined as $D^{(r)}(\theta) = D^{(1)}(D^{(r-1)}(\theta))$, where $D^{(r)}(\theta) \in \mathbb{R}^{n-r}$. With this notation, for $r \geq 1$, the $r$th order total variation of a vector $\theta$ is given as

$$\mathrm{TV}^{(r)}(\theta) = n^{r-1}\|D^{(r)}(\theta)\|_1. \tag{11}$$

The concept of the $r$th total variation can be understood as follows. Consider $\theta$ as the evaluations of an $r$ times differentiable function $f : [0, 1] \to \mathbb{R}$ on the grid $(1/n, 2/n, \ldots, n/n)$. In this case, a Riemann approximation of the integral $\int_{[0,1]} |f^{(r)}(t)|dt$ corresponds precisely to $\mathrm{TV}^{(r)}(\theta)$, where $f^{(r)}$ denotes the $r$th derivative of $f$. Therefore, for natural instances of $\theta$, it is reasonable to expect that $\mathrm{TV}^{(r)}(\theta) = O(1)$. The above discussion motivates us to define the sets

$$K_t := \left\{\theta \in \mathbb{R}^n \,:\, \mathrm{TV}^{(r)}(\theta) \leq V_t\right\}, \tag{12}$$

for some $V_t > 0$, and consider the corresponding estimator in (3). This estimator is referred to as *trend filtering* following standard terminology in the literature (see, for example, [65, 62, 61]), where methods based on bounded total variation of order $r$ are collectively known as trend filtering estimators. The intuition here is that if $F^*(t) \in K_t$ then the probabilities $F_1^*(t), \ldots, F_n^*(t)$ change smoothly over $i$ in the sense that $F^*(t)$ has bounded $r$th total variation. We also highlight that the set $K_t$ defined in Equation 12 is convex; see Appendix G.7 for a formal proof. The resulting set in (12) allows us to define the trend filtering distributional regression estimator subject of our next corollary which follows from the results in Section 3.1.

Refined risk bounds under additional sparsity assumptions are presented in Appendix A, where we show that trend filtering estimators can achieve near-parametric rates when the signal is both smooth and piecewise sparse. These results extend recent adaptive risk bounds in trend filtering; see, for example, Guntuboyina et al. [23].

**Corollary 3.** *Consider the estimator in (3) with $K_t$ as in (12) for an integer $r$ satisfying $r \geq 1$. If $F^*(t) \in K_t$, (4) holds for some fixed compact set $\Omega$, and $\sup_t V_t \leq V$, then*

$$\mathbb{E}\left(\frac{1}{n}\sum_{i=1}^n \mathrm{CRPS}(\widehat{F}_i, F_i^*)\right) \leq C\left[\frac{V^{\frac{2}{2r+1}}}{n^{\frac{2r}{2r+1}}} + \frac{\log n}{n}\right] \tag{13}$$

*for a positive constant $C$. Moreover, the upper bound in (13) also holds for the corresponding sorted estimators $\widetilde{F}_i$ as defined in Corollary 1, if in addition each function $F_i^*$ is continuous. Finally,*

$$\sup_{t \in \mathbb{R}} \sum_{i=1}^n \frac{1}{n}\left(\widehat{F}_i(t) - F_i^*(t)\right)^2 = O_{\mathbb{P}}\left(\frac{V^{\frac{2}{2r+1}}}{n^{\frac{2r}{2r+1}}} + \frac{\log n}{n}\right), \tag{14}$$

*where (14) holds without requiring (4), nor continuity of the $F_i^*$'s.*

Corollary 3 establishes that the constrained version of trend filtering for distributional regression achieves the rate $V^{1/(2r+1)}n^{-2r/(2r+1)}$, ignoring logarithmic factors, for both the CRPS and the worst-case MSE. This result aligns with the convergence rate of trend filtering in one-dimensional regression, where the same rate is attained when the regression function has $r$th-order total variation [42, 62, 23]. A summary of these comparisons is provided in Table 2 in Appendix E. Additionally, per Corollary 7 in Appendix B, the penalized version of trend filtering for distributional regression achieves the same rate in terms of the worst-case MSE, further reinforcing its consistency with classical trend filtering results.

## 3.3 Risk Bounds for the General Case

### 3.3.1 General Result

This subsection aims to present our main result on constrained distributional regression in scenarios where the constraint sets $K_t$ are arbitrary, not necessarily convex, and potentially misspecified for $F^*(t)$.

**Theorem 3.** *Let $\widehat{F}(t)$ be the estimator defined in (3) for all $t \in \mathbb{R}$ but with $K_t$ not necessarily convex and with $F^*(t)$ not necessarily in $K_t$. Suppose that $\sup_{t \in \mathbb{R}} \sup_{F(t) \in K_t} \|F(t)\|_\infty \le B$ for some constant $B \ge 1$, and $K_t \subset K$ for all $t$ and some set $K$. Let $G(t)$ be defined as $G(t) \in \arg\min_{F(t) \in K_t} \|F(t) - F^*(t)\|_\infty$. Then, for $\eta > 1$, with $K(\eta) = (K - K) \cap B_\eta(0)$, we have that*

$$
\mathbb{P}\left(\sup_{t \in \mathbb{R}} \|\widehat{F}(t) - F^*(t)\| > \eta + \sup_{t \in \mathbb{R}} \sqrt{n}\|F^*(t) - G(t)\|_\infty\right)
$$
$$
\le \frac{C}{\eta^2} \sum_{j=1}^J \frac{1}{2^{j-2}} \int_0^{2^{j/2}\eta/4} \sqrt{\log N(\varepsilon, K(\eta), \|\cdot\|)} d\varepsilon + \frac{C\sqrt{\log n}}{\eta} + \frac{C\sqrt{n}}{\eta} \sup_{t \in \mathbb{R}} \|G(t) - F^*(t)\|_\infty
$$

*for some positive constant $C > 0$, and where $J = \left\lceil \frac{\log(2nB/\eta^2)}{\log 2} \right\rceil$.*

The intuition behind Theorem 3 is that $\eta$ captures the estimation error, which depends on the local covering complexity of the sets $K(\eta)$. The second term, $\sup_{t \in \mathbb{R}} \sqrt{n}\|F^*(t) - G(t)\|_\infty$, corresponds to the approximation error, measuring how well the true $F^*(t)$ can be approximated within the model class $K_t$. While such a decomposition into estimation and approximation error is standard in nonparametric theory, our result provides a uniform guarantee over all $t \in \mathbb{R}$, in contrast to classical bounds that control error only at a fixed evaluation point, as in [47]. This distinction is particularly relevant for distributional regression problems, where the goal is to control the entire CDF path. The proof of Theorem 3, provided in Appendix G.8, relies on a peeling argument and extends techniques originally developed for the convex case.

### 3.3.2 Dense ReLU Networks

We now turn to the application of Theorem 3 to the problem of distributional regression using dense neural networks. The results in this section add to the literature on statistical theory for rectified linear unit (ReLU) networks as in Bauer and Kohler [4], Schmidt-Hieber [56], Kohler and Langer [37], Padilla et al. [49], Ma and Safikhani [40], Zhang et al. [67], Padilla et al. [47].

Before presenting our main result, we first introduce some notation. Suppose we are given i.i.d. data $\{(x_i, y_i)\}_{i=1}^n \subset [0, 1]^{d_0} \times \mathbb{R}$ and let

$$
G^*(x, t) := \mathbb{P}(y_i \le t | x_i = x) = f_t^*(x) \tag{15}
$$

for functions $f_t^* : [0, 1]^{d_0} \to \mathbb{R}$ for all $t$. We set the conditional cdf $F_i^*(t) = G^*(x_i, t)$.

To define the constraint set $K_t$, we follow Kohler and Langer [37] and assume all hidden layers have the same width. Let $\mathcal{F}(L, \nu)$ denote the set of neural networks with depth $L$, width $\nu$, and ReLU activation, restricting the functions in $\mathcal{F}(L, \nu)$ to satisfy $\|f\|_\infty \le 1$. The structure of these networks is described in Appendix C. Then, for all $t \in \mathbb{R}$, we define

$$
K_t = K := \{\theta \in \mathbb{R}^n : \theta_i = f(x_i), \ i = 1, \ldots, n, \text{ for some } f \in \mathcal{F}(L, \nu)\}. \tag{16}
$$

We next summarize the main assumptions underlying the analysis, stated formally in Appendix C. Specifically, for each threshold $t$, we assume that the true conditional distribution function $G^*(\cdot, t)$ belongs to a hierarchical composition model class $\mathcal{H}(l, \mathcal{P})$ as defined in Kohler and Langer [37]. Here, $l$ denotes the number of hierarchical composition levels, and $\mathcal{P} \subset (0, \infty) \times \mathbb{N}$ specifies allowable pairs $(p, M)$ corresponding to the smoothness $p$ and input dimension $M$ of each component function. These assumptions, ensure that the neural network class $\mathcal{F}(L, \nu)$ can approximate $G^*(\cdot, t)$ at the rate $\phi_n = \max_{(p,M) \in \mathcal{P}} n^{-2p/(2p+M)}$, which determines the convergence behavior established in the result below.

We now present our main result concerning distributional regression with ReLU neural networks.

**Corollary 4.** *Let $\widehat{F}(t)$ be the estimator from (3) with the set $K_t$ as in (16) for all $t \in \mathbb{R}$ with $F^*(t)$ not necessarily in $K_t$. Suppose that Assumption 1, described in Appendix C, holds. Let $\phi_n = \max_{(p,M) \in \mathcal{P}} n^{\frac{-2p}{(2p+M)}}$. Under the choices of $L$ and $\nu$ specified in equations (51) or (52) in Appendix G.9, the estimator satisfies*

$$\sup_{t \in \mathbb{R}} \sum_{i=1}^{n} \frac{1}{n} \left( \widehat{F}_i(t) - F_i^*(t) \right)^2 = O_{\mathbb{P}} \left( \frac{\log n}{n} + \phi_n \log^4 n \right). \tag{17}$$

Corollary 4 demonstrates that dense ReLU neural network estimators for distributional regression uniformly achieve the rate $\phi_n$, up to logarithmic factors, in terms of the worst-case MSE for estimating the true parameters $\{F^*(t)\}_{t \in \mathbb{R}}$, provided these parameters belong to a hierarchical composition class. Importantly, while this rate matches that of Kohler and Langer [37] for mean regression under sub-Gaussian error assumptions, our result strengthens the guarantee by holding uniformly over all thresholds $t \in \mathbb{R}$, rather than for a fixed target. A summary of this comparison is provided in Table 2 in Appendix E. This extension is essential for distributional learning tasks where uniform control is required, such as CRPS-based risk bounds.

## 4 Simulated data analysis

We evaluate the performance of the proposed methods against state-of-the-art approaches across diverse simulation settings that reflect various practical challenges and structural assumptions. Specifically, six distinct scenarios are considered to evaluate different aspects of the distributional regression problem. The implementation of all experiments is available at `https://github.com/cmadridp/UnifDR`. We refer to our proposed approach as **UnifDR** which adapts different methods based on the scenario. In the first two scenarios, **UnifDR** applies the isotonic regression method from Section 3.2.1; in the next two, it uses the trend filtering approach from Section 3.2.2; and in the final two scenarios, it employs the Dense ReLU Networks method described in Section 3.3.2.

**Scenario 1 (S1).** We generate data $y_i \sim \text{Normal}(\mu_i, 1)$ where $\mu_i = 1 - i/n$ for $i = 1, \ldots, n$.

**Scenario 2 (S2).** We consider $y_i \sim \text{Unif}(a_i, b_i)$, where $a_i = (n - i)/n$ and $b_i = a_i + 1$.

**Scenario 3 (S3).** The true CDFs are modeled as $F_i^*(t) = 1 - \exp(-t/\mu_i)$, $\mu_i = 1 + 0.5 \sin(2\pi i/n)$.

**Scenario 4 (S4).** Consider $F_i^*(t) = \text{Gamma}(\text{shape} = 0.7, \text{scale} = \mu_i)$, where $\mu_i = 6 \cdot \mathbf{1}_{\{i \le n/4\}} + 2 \cdot \mathbf{1}_{\{n/4 < i \le n/2\}} + 8 \cdot \mathbf{1}_{\{n/2 < i \le 3n/4\}} + 4 \cdot \mathbf{1}_{\{i > 3n/4\}}$.

**Scenario 5 (S5).** Let $\mathbf{x}_i \sim \text{Unif}([0, 1]^5)$. The true CDFs are given by $F_i^*(t) = \Phi((t - h(\mathbf{x}_i))/0.5)$, where $h(\mathbf{x}_i) = -3x_i^{(1)} + 2\log(1 + x_i^{(2)}) + x_i^{(3)} + 5x_i^{(4)} + (x_i^{(5)})^2$, with $x_i^{(j)}$ denoting the $j$ th coordinate of $x_i$. The function $\Phi$ represents the standard normal cumulative distribution function.

**Scenario 6 (S6).** Let $\mathbf{x}_i \sim \text{Unif}([0, 1]^{10})$ and $y_i \sim \chi^2(h(\mathbf{x}_i))$. Here, $h(\mathbf{x}_i) = \log\left(\left| -0.5 \cdot \sum_{j=1}^{3} \sin(\pi x_i^{(j)}) - 0.5 \sum_{j=4}^{9} x_i^{(j)} + 0.5 \cos(x_i^{(10)}) \right| + 2\right)$, with $x_i^{(j)}$ denoting the $j$ th coordinate of $x_i$.

To implement our proposed approach **UnifDR** we proceed as follows. The isotonic method introduced in Section 3.2.1 is implemented in R using the pool adjacent violators algorithm (PAVA) from Robertson [51]. For this approach there are no direct competitors in the distributional regression problem. The Trend Filtering estimator in Section 3.2.2 is implemented using the `trendfilter` function from the `glmgen` package in R, and we compare it with additive smoothing splines (AddSS)

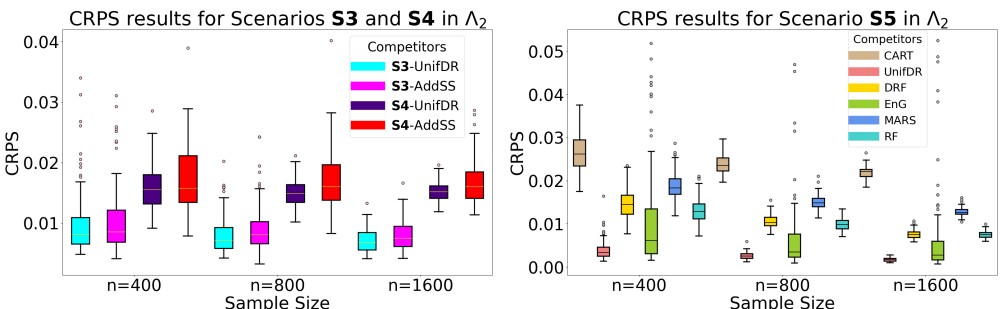

Figure 1: Box plots of CRPS results in $\Lambda_2$. The left plot corresponds to **S3** and **S4**, while the right plot displays the results for **S5**.

Table 1: Evaluation metrics for **UnifDR** (Trend Filtering approach) and its competitor AddSS on the 2015 Chicago crime dataset.

| METHOD | CRPS (MEAN ± STD) | MSD (MEAN ± STD) |
|---|---|---|
| UNIFDR | **0.0976±0.0017** | **0.2509±0.0025** |
| ADDSS | 0.1223±0.0078 | 0.2850±0.0391 |

via the `smooth.spline` function in R. For the Dense ReLU Networks method in Section 3.3.2, we use a fully connected feedforward architecture with an input layer, two hidden layers (64 units each), and an output layer. The network is implemented in `Python` and trained using the Adam optimizer with a learning rate of 0.001. In this case, we compare the proposed **UnifDR** method with five benchmark methods. First, we consider Classification and Regression Trees (CART) [8], implemented in R via the `rpart` package, with the complexity parameter used for tuning. Second, we evaluate Multivariate Adaptive Regression Splines (MARS) [21], available in the `earth` package, where the penalty parameter serves as the tuning parameter. Third, we assess Random Forests (RF) [7], implemented in R via the `randomForest` package, using 500 trees and tuning the minimum terminal node size. Additionally, we consider two recent methods. Distributional Random Forests (DRF) [10] is implemented via the `drf` package in `Python`. DRF employs tree-based ensemble with tuning parameters including the splitting rule and the number of trees. Lastly, Engression (EnG) [58] is implemented using the `engression` package in `Python`. EnG utilizes hierarchical structured neural networks, and we adopt the same training hyperparameters as our deep learning approach to ensure optimization consistency. EnG also requires a sampling procedure, with the number of samples set to 1000 for accurate distribution estimation.

**Performance Evaluation:** For each scenario, datasets with sample sizes $n \in \{400, 800, 1600\}$ are generated, with each experiment repeated 100 times using Monte Carlo simulations. Evaluations are conducted at 100 evenly spaced points $t$ from three fixed intervals: $\Lambda_1 = [-1, 0.4]$, $\Lambda_2 = [-2, 2]$, and $\Lambda_3 = [0.8, 10]$. Each dataset is randomly split into 75% training and 25% test sets. Competing models undergo 5-fold cross-validation on the training data for hyperparameter tuning, with performance assessed on the test set. For the isotonic regression method, test set predictions are obtained via naive nearest neighbor interpolation. The accuracy of the estimated CDFs $\widehat{F}_i(t)$ relative to the true CDFs $F_i^*(t)$ is evaluated using the following performance metrics, averaged over 100 Monte Carlo repetitions. **CRPS:** CRPS evaluates the overall fit of $\widehat{F}_i(t)$ to $F_i^*(t)$ across all evaluation points in $\mathbb{R}$, see Section 2. Since the evaluations in experiments are performed over a finite set of 100 values in $\Lambda$, CRPS is approximated via a Riemann sum: $\text{CRPS} = \frac{1}{|\text{Test}|} \sum_{i \in \text{Test}} \frac{1}{100} \sum_{t \in \Lambda} \left( \widehat{F}_i(t) - F_i^*(t) \right)^2$, where $|\text{Test}| = \frac{n}{4}$ is the size of the test set. **Maximum Squared Difference (MSD):** MSD captures the worst-case discrepancy between $\widehat{F}_i(t)$ and $F_i^*(t)$, and is approximated as: $\text{MSD} = \max_{t \in \Lambda} \frac{1}{|\text{Test}|} \sum_{i \in \text{Test}} \left( \widehat{F}_i(t) - F_i^*(t) \right)^2$.

The results below focus on the CRPS metric for Scenarios **S3**, **S4**, and **S5**, where the CDFs are evaluated at the points in $\Lambda_2$. Additional results, including MSD performance and CRPS evaluations at $\Lambda_1$ and $\Lambda_3$ across all scenarios, as well as the $\Lambda_1$-based CRPS for Scenarios **S1**, **S2**, and **S6**, are provided in Appendix F. Figure 1 presents the performance of **UnifDR** in **S3** and **S4**, where we compare the Trend Filtering approach with AddSS. The same figure also includes results for **S5**, where **UnifDR** utilizes the Dense ReLU Networks method against five state-of-the-art competitors: CART, MARS, RF, DRF, and EnG. In all scenarios, **UnifDR** consistently outperforms competing

methods, with its performance superiority becoming more pronounced as the sample size increases. This dominance is further confirmed by the extended evaluations in Appendix F, reinforcing the robustness of **UnifDR** across various conditions.

# 5 Real data application

In this section, we evaluate the performance of the proposed **UnifDR** method using both the Trend Filtering and Dense ReLU network procedures on real-world datasets.

## 5.1 Chicago crime data

We analyze the 2015 Chicago crime dataset, available at `https://data.gov/open-gov/`, which records reported crimes in Chicago throughout the year. Following Tansey et al. [60], the spatial domain is discretized into a $100 \times 100$ grid, where each grid cell aggregates crime counts within its spatial boundary. The response variable is defined as the log-transformed total crime counts per grid cell. Grid cells with zero observed crimes are excluded, yielding a final dataset of 3,844 grid cells. The **UnifDR'** Trend Filtering procedure does not use covariates. Instead, the spatial grid is treated as an ordered sequence, assuming a smooth spatial trend. The grid cells are ordered lexicographically. The dataset is randomly partitioned into 100 train (75%)–test (25%) splits, and evaluation is conducted at evenly spaced points $\Lambda$ in the interval [-1,6]. Performance is assessed using the Continuous Ranked Probability Score (CRPS) and Maximum Squared Difference (MSD) metrics, comparing estimated CDFs $\widehat{F}_i(t)$ against empirical indicators $w_i(t)$, where $t \in \Lambda$. AddSS is used as a benchmark for the Trend Filtering approach on the same dataset. Table 1 presents CRPS and MSD metrics, demonstrating **UnifDR** superior performance. Furthermore, Figure 4 in Appendix F.1.1 provides a visualization of $\widehat{F}_i(t)$ at $t = 3$ for both competing methods. The same data set is analyzed using **UnifDR** with the Dense ReLU network framework, with further details in Appendix F.1.

## 5.2 Other real data examples

Beyond the Chicago crime dataset, **UnifDR** is further evaluated on California housing prices and daily Ozone measurements. Detailed descriptions, pre-processing steps, and results are provided in the Appendix F.1.

# 6 Conclusion

This paper introduced a unified framework for nonparametric distributional regression under convex and non-convex structural constraints. We established theoretical risk bounds for the estimation of cumulative distribution functions (CDFs) in various settings, including isotonic regression, trend filtering, and deep neural networks. Our analysis leveraged continuous ranked probability scores (CRPS) and worst-case mean squared error (MSE) to quantify estimation accuracy, demonstrating that structured constraints such as monotonicity, bounded total variation, and hierarchical function composition lead to improved estimation accuracy. The resulting upper bounds are general and apply to a broad class of convex and non-convex estimators. While we do not derive lower bounds at this level of generality, the rates obtained for specific cases—such as isotonic and trend filtering estimators—match known minimax results for mean estimation, providing evidence for the tightness and near-optimality of our bounds in these canonical examples (see Table 2). In particular, we establish explicit convergence guarantees for isotonic, strengthening prior findings. Moreover, the trend filtering estimator achieves rates consistent with classical one-dimensional regression results. For deep neural networks, we show that our estimator achieve comparable rates under hierarchical composition constraints, aligning with existing results in structured regression. Experiments on simulated and real datasets further validated the theoretical guarantees, with our proposed methods consistently outperforming alternative approaches. In particular, **UnifDR**, the distributional regression framework we study, demonstrated superior performance across all considered settings.

An important avenue for future research is extending our theory to dependent data settings. Many real-world applications involve time-series data, spatial data, or network-structured data, where dependencies among observations must be accounted for.

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

# A  Fast Rates

This appendix develops refined risk bounds for distributional regression estimators under additional structural assumptions on the true signal. While our general theory establishes minimax-optimal rates under convex constraints such as monotonicity or bounded total variation, certain low-complexity signal classes allow for significantly faster convergence.

We provide two canonical examples illustrating this phenomenon. The first focuses on isotonic regression, where we show that if the true distribution functions have only a small number of strict increases, the estimator achieves a nearly parametric rate. The second concerns trend filtering, where we demonstrate that sparsity in the higher-order differences of the signal—combined with a minimum segment length condition—leads to similarly fast convergence.

These results extend analogous adaptive risk bounds from classical point estimation [12, 23] to the distributional regression setting.

## A.1  Isotonic Regression

We begin by revisiting the isotonic case. In the main text, Corollary 2 established a general risk bound of order $n^{-2/3}$ under minimal monotonicity assumptions. Here, we refine this analysis by showing that much faster rates, nearly parametric, are achievable when the true distribution function $F^*(t)$ exhibits low complexity, in the sense of having few strict increases.

To formalize this, consider the isotonic constraint set

$$K := \{\theta \in \mathbb{R}^n : \theta_1 \leq \theta_2 \leq \cdots \leq \theta_n\}, \tag{18}$$

and assume $F^*(t) \in K$ for all $t \in \mathbb{R}$. Define

$$k(t) := \left|\left\{i \in \{1, \ldots, n-1\} : F_i^*(t) < F_{i+1}^*(t)\right\}\right|.$$

This quantity counts the number of strict increases in $F^*(t)$, and hence controls the complexity of the signal.

**Corollary 5.** *Let $\widehat{F}_t$ be the estimator defined in* (3) *with $K_t = K$ as in* (18)*. Assume $F^*(t) \in K$ and that* (4) *holds for a compact set $\Omega$. Then*

$$\mathbb{E}\left(\frac{1}{n}\sum_{i=1}^n \mathrm{CRPS}(\widehat{F}_i, F_i^*)\right) \leq C \sup_{t \in \mathbb{R}} \cdot \frac{1+k(t)}{n} \log\left(\frac{en}{1+k(t)}\right)$$

*for some constant $C > 0$. Moreover, this bound also applies to the sorted estimator $\widetilde{F}$ from Corollary 1, provided each $F_i^*$ is continuous.*

This result improves on the general $n^{-2/3}$ bound when the number of strict increases $k(t)$ is small. For example, if $F^*(t)$ is piecewise constant with at most $s$ jumps, then $k(t) \leq s$, and the bound becomes $\frac{1+s}{n} \log\left(\frac{en}{1+s}\right)$, nearly achieving the parametric rate when $s = O(1)$. The proof of Corollary 5 is provided in Appendix G.10.

## A.2  Trend Filtering

We now turn to the trend filtering setting. In the main text, Corollary 3 established a general risk bound of order $n^{-2r/(2r+1)}$ under a total variation constraint of order $r$, matching the minimax-optimal rate for function estimation under such constraints [42, 62].

Here, we show that significantly faster rates are achievable in the distributional regression setting when the signal exhibits additional structure. In particular, when the $r$th-order differences of the vector $F^*(t)$ are sparse and well-separated, the estimator can attain nearly parametric accuracy up to logarithmic factors.

Define

$$K_t := \left\{\theta \in \mathbb{R}^n : \mathrm{TV}^{(r)}(\theta) \leq V_t^*\right\}, \tag{19}$$

where $V_t^* = \mathrm{TV}^{(r)}(F^*(t))$.

**Corollary 6.** *Consider the estimator in (3) with $K_t$ as in (19) for an integer $r$ satisfying $r \geq 1$. Suppose $s = \left\| D^{(r)} F^*(t) \right\|_0$ and $S = \left\{ j : \left( D^{(r)} F^*(t) \right)_j \neq 0 \right\}$ for all $t$. Let $j_0 < j_1 < \ldots < j_{s+1}$ be such that $j_0 = 1, j_{s+1} = n - r$ and $j_1, \ldots, j_s$ are the elements of $S$. With this notation define $\eta_{j_0} = \eta_{j_{s+1}} = 0$. Then for $j \in S$ define $\eta_j$ to be 1 if $\left( D^{(r-1)} F^*(t) \right)_j < \left( D^{(r-1)} F^*(t) \right)_{j+1}$, otherwise set $\eta_j = -1$. Suppose that $F^*(t)$ satisfies the following minimum length assumption*

$$\min_{l \in [s], \eta_{j_l} \neq \eta_{j_l+1}} (j_{l+1} - j_l) \geq \frac{cn}{s+1}$$

*for some constant $c$ satisfying $0 \leq c \leq 1$. Then, for $\sup_t V_t^* \leq V^*$,*

$$\mathbb{E} \left( \frac{1}{n} \sum_{i=1}^n \text{CRPS}(\widehat{F}_i, F_i^*) \right) \leq C \left[ \frac{(s+1)}{n} \log \left( \frac{en}{s+1} \right) \right] \tag{20}$$

*for a positive constant $C$. Moreover, the upper bound in (20) also holds for the corresponding sorted estimators $\widetilde{F}_i$ as defined in Corollary 1, provided that each function $F_i^*$ is continuous and $V_t^* \leq V^*$ for all $t$.*

This result shows that, under a sparse difference structure and a minimum segment length condition, trend filtering estimators for distributional regression can achieve nearly parametric accuracy (up to logarithmic factors). The bound in (20) mirrors the adaptive rates derived for standard trend filtering in point estimation problems; see, for example, Guntuboyina et al. [23]. Our extension demonstrates that the same refined rate behavior persists in the more general context of distributional regression, where the goal is to estimate full conditional distributions rather than scalar means.

The proof of Corollary 6 is provided in Appendix G.11.

## B  General Result for Penalized Estimators

A natural alternative to shape-constrained estimators is the use of penalized estimators, where the penalty term promotes a desired behavior in the signal being estimated. Motivated by this approach, we present a general result for distributional regression using penalized estimators in this subsection. Specifically, consider estimators of the form

$$\hat{F}(t) := \arg\min_{\theta \in \mathbb{R}^n} \left\{ \frac{1}{2} \|w(t) - \theta\|^2 + \lambda_t \text{pen}_t(\theta) \right\}, \tag{21}$$

where $\lambda_t > 0$ is a tuning parameter and $\text{pen} : \mathbb{R}^n \to \mathbb{R}$ is a penalty function. We now present our main result for the penalized estimator defined in (21).

**Theorem 4.** *Suppose that $\text{pen}_t(\cdot)$ is convex for all $t$ and it is a semi-norm. In addition, assume that $\sup_{t \in \mathbb{R}} \text{pen}_t(F^*(t)) \leq V$. Let $K := \{\theta \in \mathbb{R}^n : \text{pen}_t(\theta) \leq 6V\}$. Then for any $\eta > 0$, it holds that*

$$\mathbb{P} \left( \sup_{t \in \mathbb{R}} \sum_{i=1}^n \left( \widehat{F}_i(t) - F_i^*(t) \right)^2 > 2\eta^2 \right) \leq \frac{C}{\eta^2} \int_0^{\eta/4} \sqrt{\log N(\varepsilon, K \cap B_\eta(0), \| \cdot \|)} d\varepsilon + \frac{C\sqrt{\log n}}{\eta}, \tag{22}$$

*for some constant $C > 0$, proivide that we set $\lambda_t = \eta^2/4\text{pen}(F^*(t))$.*

Theorem 4 demonstrates that achieving a uniform upper bound on the MSE can be accomplished by controlling the covering number of sets of the form $K \cap B_\varepsilon(0)$, as outlined on the right-hand side of (22).

We now turn to a statistical guarantee for the penalized version of trend filtering in distributional regression. This result follows directly from Theorem 4 and involves a calculation analogous to the proof of (14).

**Corollary 7.** *Consider the estimator in (21) with $\text{pen}_t(\theta) := \text{TV}^{(r)}(\theta)$ and $\lambda_t$ chosen as in Theorem 4. Then*

$$\sup_{t \in \mathbb{R}} \sum_{i=1}^n \frac{1}{n} \left( \widehat{F}_i(t) - F_i^*(t) \right)^2 = O_\mathbb{P} \left( \frac{V^{\frac{2}{2r+1}}}{n^{\frac{2r}{2r+1}}} + \frac{\log n}{n} \right), \tag{23}$$

*where $V := \sup_{t \in \mathbb{R}} \text{pen}_t(F^*(t))$.*

# C Dense ReLU Networks: assumption and definitions

In this appendix, we provide additional details for Section 3.3.2. Before outlining our assumptions on the functions $F^*(t)$, we introduce notation related to dense ReLU networks. To that end, we describe a dense neural network with architecture $(L, k)$ employing the ReLU activation function given as $\rho(s) = \max\{0, s\}$ for any $s \in \mathbb{R}$. Such a network is represented as a real-valued function $f : \mathbb{R}^d \to \mathbb{R}$ satisfying the following properties:

$$f(x) = \sum_{i=1}^{k_L} c_{1,i}^{(L)} f_i^{(L)}(x) + c_{1,0}^{(L)} \tag{24}$$

for weights $c_{1,0}^{(L)}, \ldots, c_{1,k_L}^{(L)} \in \mathbb{R}$ and for $f_i^{(L)}$'s recursively defined by

$$f_i^{(s)}(x) = \rho \left( \sum_{j=1}^{k_{s-1}} c_{i,j}^{(s-1)} f_j^{(s-1)}(x) + c_{i,0}^{(s-1)} \right) \tag{25}$$

for some $c_{i,0}^{(s-1)}, \ldots, c_{i,k_{s-1}}^{(s-1)} \in \mathbb{R}$, $s \in \{2, \ldots, L\}$, and $f_i^{(1)}(\mathbf{x}) = \rho \left( \sum_{j=1}^d c_{i,j}^{(0)} x^{(j)} + c_{i,0}^{(0)} \right)$ with $c_{i,0}^{(0)}, \ldots, c_{i,d}^{(0)} \in \mathbb{R}$.

Next we provide some notation necessary for define the class of signals where the $F^*(t)$'s belong.

**Definition 1** $((p, C)$-smoothness$)$**.** *Let $p = q + s$ for some $q \in \mathbb{N} = \mathbb{Z}^+ \cup \{0\}$ and $0 < s \le 1$. We say that a function $g : \mathbb{R}^d \to \mathbb{R}$ is $(p, C)$-smooth, if for every $\alpha = (\alpha_1, \ldots, \alpha_d) \in \mathbb{N}^d$, with $d \in \mathbb{Z}^+$, where $\sum_{j=1}^d \alpha_j = q$, the partial derivative $\partial^q g / (\partial u_1^{\alpha_1} \ldots \partial u_d^{\alpha_d})$ exists and*

$$\left| \frac{\partial^q g}{\partial u_1^{\alpha_1} \ldots \partial u_d^{\alpha_d}} (u) - \frac{\partial^q g}{\partial u_1^{\alpha_1} \ldots \partial u_d^{\alpha_d}} (v) \right| \le C \|u - v\|^s$$

*for all $u, v \in \mathbb{R}^d$.*

Let us now define the generalized hierarchical interaction models $\mathcal{H}(l, \mathcal{P})$.

**Definition 2** (Space of Hierarchical Composition Models, [37])**.** *For $l = 1$ and smoothness constraint $\mathcal{P} \subseteq (0, \infty) \times \mathbb{N}$, the space of hierarchical composition models is defined as*

$$\mathcal{H}(1, \mathcal{P}) := \left\{ h : \mathbb{R}^d \to \mathbb{R} : h(a) = m \left( a_{(\pi(1))}, \ldots, a_{(\pi(M))} \right), \text{ where} \right.$$
$$\left. m : \mathbb{R}^M \to \mathbb{R} \text{ is } (p, C)\text{-smooth for some } (p, M) \in \mathcal{P} \text{ and } \pi : \{1, \ldots, M\} \to \{1, \ldots, d\} \right\}.$$

*For $l > 1$, we set*

$$\mathcal{H}(l, \mathcal{P}) := \left\{ h : \mathbb{R}^d \to \mathbb{R} : h(\mathbf{x}) = m \left( f_1(a), \ldots, f_M(a) \right), \text{ where} \right.$$
$$\left. m : \mathbb{R}^M \to \mathbb{R} \text{ is } (p, C)\text{-smooth for some } (p, M) \in \mathcal{P} \text{ and } f_i \in \mathcal{H}(l - 1, \mathcal{P}) \right\}.$$

With the notation above, we are ready to state our assumption on the true signals in the spirit of [37].

**Assumption 1.** *Suppose that for all $t$ the function $G^*(\cdot, t)$ is in the class $\mathcal{H}(l, \mathcal{P})$ as in Definition 2. In addition, assume that each function $g^t$ in the definition of $G^*(\cdot, t)$ can have different smoothness $p_{g^t} = q_{g^t} + s_{g^t}$, for $q_{g^t} \in \mathbb{N}$, $s_{g^t} \in (0, 1]$, and of potentially different input dimension $M_{g^t}$, so that $(p_{g^t}, M_{g^t}) \in \mathcal{P}$. Let $M_{\max}$ be the largest input dimension and $p_{\max}$ the largest smoothness of any of the functions $g^t$ for all $t$. Suppose that for each $g^t$ all the partial derivatives of order less than or equal to $q_{g^t}$ are uniformly bounded by constant $C_{\text{Smooth}}$, and each function $g^t$ is Lipschitz continuous with Lipschitz constant $C_{\text{Lip}} \ge 1$. Also, assume that $\max\{p_{\max}, M_{\max}\} = O(1)$.*

# D Rearrangement of estimates

Figure 2 shows an example of the original $\widehat{F}_i$ and its rearrangement $\widetilde{F}_i$ as in Corollary 1, which is guaranteed to be non-decreasing.

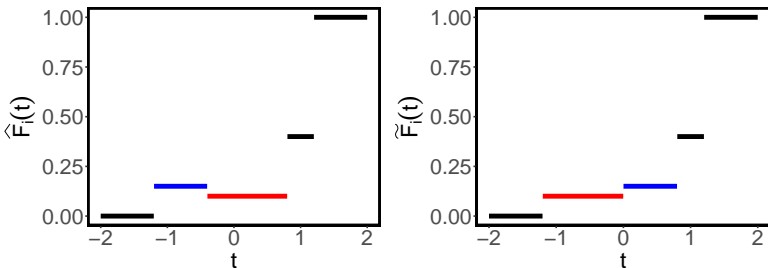

Figure 2: The plot in the left shows a display of an example of a function $\widehat{F}_i$ and the right panel shows the corresponding rearrangement $\widetilde{F}_i$ as described in Corollary 1.

## E  Summary of Results and Theoretical Comparisons with Prior Work

For a summary of all theoretical results presented in the paper and comparisons with prior work such as [28, 44, 42, 62, 23, 37, 10, 58], see Table 2. This table synthesizes the convergence rates and underlying assumptions for our estimators (isotonic, trend filtering, and dense ReLU networks) alongside those of key existing methods discussed throughout the manuscript.

In addition, Appendix E.1 provides a complementary comparison that focuses specifically on methods designed for conditional distribution estimation. In particular, we contrast our Dense ReLU approach from Section 3.3.2 with the main competing estimators evaluated in the simulation section, Section 4, including those from [10] and [58], highlighting differences in assumptions, estimation strategies, and theoretical guarantees.

Finally, this appendix also includes a dedicated subsection (Appendix E.2) that clarifies the conceptual relation and technical distinctions between our Theorem 1 and Theorem A.1 of Guntuboyina et al. [23]. This comparison emphasizes how our framework extends the convex-projection setting to distributional regression with Bernoulli-type observations and CRPS loss, relying on different concentration and normalization arguments.

### E.1  Error Bounds for Conditional Distribution Estimation

The result in Corollary 4 establishes that the dense ReLU network-based estimator achieves a convergence rate of order

$$\frac{\log n}{n} + \phi_n \log^4 n,$$

where $\phi_n = \max_{(p,M)\in\mathcal{P}} n^{-2p/(2p+M)}$, for the worst-case mean squared error over thresholds $t \in \mathbb{R}^p$, as shown in (17). This result provides a high-probability bound on the squared $\ell_2$ error between the estimated and true conditional CDF vectors

$$\widehat{F}(t) = \left(\widehat{F}_1(t), \ldots, \widehat{F}_n(t)\right)^\top$$

and

$$F^*(t) = (F_1^*(t), \ldots, F_n^*(t))^\top,$$

where $F_i^*(t) := \mathbb{P}(y_i \le t \mid x_i) = G(x_i, t)$, uniformly across $t \in \mathbb{R}$. Here $\{(x_i, y_i)\}_{i=1}^n \subset [0,1]^{d_0} \times \mathbb{R}$ are i.i.d. data.

To contextualize this result, we compare it with the result obtained in [10] and [58].

#### E.1.1  Comparison with Distributional Random Forests (DRF) [10]

Work [10] studies conditional distribution estimation using Distributional Random Forests (DRF). Their framework assumes access to i.i.d. samples $\{(x_i, y_i)\}_{i=1}^n \subset [0,1]^p \times \mathbb{R}^d$, where each $x_i \in [0,1]^p$ denotes a covariate vector and $y_i \in \mathbb{R}^d$ a multivariate response. The goal is to estimate the conditional distribution $\mathbb{P}(Y \mid X = x)$, where $Y \in \mathbb{R}^d$ and $X \in [0,1]^p$, using a weighted empirical measure of the form

$$\hat{\mathbb{P}}(Y \mid x) := \sum_{j=1}^n w_x(x_j) \, \delta_{y_j},$$

where $\delta_{y_i}$ is the Dirac measure at $y_i$, assigning unit mass to the observation $y_i \in \mathbb{R}^d$. Moreover, $w_x(x_j)$ denotes the DRF-induced weight assigned to training point $x_j$. Specifically, after constructing a forest of $N$ randomized trees, each test point $\mathbf{x} \in [0,1]^p$ defines, for every tree $\mathcal{T}_k$, a leaf $\mathcal{L}_k(\mathbf{x})$ consisting of training points $x_j$ that fall in the same terminal node as $\mathbf{x}$. The DRF weighting function is then defined as

$$w_{\mathbf{x}}(x_j) := \frac{1}{N} \sum_{k=1}^{N} \frac{1_{\{x_j \in \mathcal{L}_k(x)\}}}{|\mathcal{L}_k(x)|}.$$

This leads to the following plug-in estimator of the conditional CDF at a threshold $\mathbf{t} \in \mathbb{R}^d$

$$\hat{F}_{\mathbf{Y}|\mathbf{X}=\mathbf{x}}^{\mathrm{DRF}}(\mathbf{t}) := \sum_{j=1}^{n} w_{\mathbf{x}}(x_j) \cdot 1_{\left\{(y_j)_1 \leq t_1, \ldots, (y_j)_d \leq t_d\right\}},$$

which computes a weighted empirical CDF based on the forest-derived weights. To align with our setup, consider the univariate case $d = 1$ with $p = d_0$, where each observation $(x_i, y_i) \in [0,1]^{d_0} \times \mathbb{R}$. In this setting, their goal becomes estimating the conditional CDF $F_{Y|X=x}(t) := \mathbb{P}(Y \leq t \mid X = x)$ for any threshold $t \in \mathbb{R}$. The DRF estimator in this case simplifies to $\hat{F}_{Y|X=x}^{\mathrm{DRF}}(t) = \sum_{j=1}^{n} w_x(x_j) \cdot 1\{y_j \leq t\}$, providing a weighted empirical estimate of the conditional distribution function at point $x$. Viewed from our perspective, where the focus is on the sequence model $F^*(t) := (F_1^*(t), \ldots, F_n^*(t))^\top$, with $F_i^*(t) := \mathbb{P}(y_i \leq t \mid x_i)$, the DRF method yields estimates

$$\hat{F}_i^{\mathrm{DRF}}(t) := \sum_{j=1}^{n} w_{x_i}(x_j) \cdot 1\{y_j \leq t\},$$

where the weight vector $w_{x_i}(\cdot)$ is centered around the design point $x_i$. These estimates $\hat{F}_i^{\mathrm{DRF}}(t)$ are directly comparable to our estimator $\hat{F}_i(t)$. In terms of assumptions, their theoretical guarantee relies on several conditions about how the weights $w_{x_i}(x_j)$ are constructed. One key requirement is that each tree in the forest is built using a random subsample of the data, rather than sampling with replacement. The size of this subsample $s_n$ is allowed to grow with $n$, specifically with $s_n \asymp n^\beta$ for some $0 < \beta < 1$. They also assume that the covariate distribution on $[0,1]^{d_0}$ has a density bounded away from zero and infinity. This contrasts with our setting, where no such condition on the distribution of the covariates is required. Another assumption is that for each threshold $t$, the conditional CDF function $x \mapsto F_{Y|X=x}(t)$ is Lipschitz. Translated to our notation, this corresponds to requiring that for every $t \in \mathbb{R}$, the function $G(\cdot, t)$, where $F_i^*(t) = G(x_i, t)$, is Lipschitz in $x$. In terms of our assumption that $G(\cdot, t) \in \mathcal{H}(l, \mathcal{P})$, as stated in Assumption 1, this is equivalent to assuming that $G(\cdot, t) \in \mathcal{H}(1, \{(1, d_0)\})$, that is, $G(\cdot, t)$ belongs to a Lipschitz class, which is a strict subset of the more general hierarchical composition model we allow.

Under these conditions, the theoretical guarantees in [10] for the DRF method are limited to convergence in probability. Specifically, Corollary 5 in their paper establishes that the DRF estimator $\hat{F}_i^{\mathrm{DRF}}(t)$ converges in probability to the true conditional CDF $F_i^*(t)$ at fixed thresholds $t \in \mathbb{R}$. However, this result does not provide explicit rates of convergence in mean squared error or uniform guarantees over $t$.

In contrast, our result provides a high-probability bound on the $\ell_2$ error over the full vector $F^*(t)$, with a convergence rate

$$\sup_{t \in \mathbb{R}} \sum_{i=1}^{n} \frac{1}{n} \left( \hat{F}_i(t) - F_i^*(t) \right)^2 = O_{\mathbb{P}} \left( \frac{\log n}{n} + \phi_n \log^4 n \right),$$

where $\phi_n = n^{-2/(2+d)}$ matches the approximation rate for the Lipschitz class assumed in their setup. More generally, our framework also provides rates under weaker smoothness conditions, allowing each threshold function $G(\cdot, t)$ to belong to the hierarchical composition model class $\mathcal{H}(l, \mathcal{P})$, as defined in Assumption 1.

A summary of this comparison is provided in Table 2.

### E.1.2  Comparison with Engression (EnG) [58]

The work on [58] introduces a method for estimating conditional distributions using energy score minimization. Their framework assumes access to i.i.d. data $\{(x_i, y_i)\}_{i=1}^{n} \subset [0,1]^d \times \mathbb{R}$, where

each $x_i \in [0, 1]^d$ is a covariate vector and $y_i \in \mathbb{R}$ is a real-valued response. The main objective is to estimate the full conditional distribution $\mathbb{P}(Y \mid X = x)$.

To solve this problem, Engression models the conditional distribution $Y \mid X = x$ as a transformation of an independent noise variable $\varepsilon \sim \text{Unif}[0, 1]$, i.e., it posits the model

$$Y = g^*(X, \varepsilon)$$

for some unknown function $g^* \colon [0, 1]^d \times [0, 1] \to \mathbb{R}$.

The goal is to learn an approximation $g_\theta$ of $g^*$, parameterized by a deep neural network, such that the conditional distribution of $g_\theta(X, \varepsilon) \mid X$ matches that of $Y \mid X$. The model is trained by minimizing an empirical energy score loss, which quantifies the discrepancy between the observed responses and the model-generated samples. For each data point $(x_i, y_i)$, the method samples $m$ independent noise variables $\varepsilon_{i,1}, \ldots, \varepsilon_{i,m} \overset{\text{i.i.d.}}{\sim} \text{Unif}[0, 1]$. The resulting empirical loss function is

$$\min_\theta \frac{1}{n} \sum_{i=1}^{n} \left[ \frac{1}{m} \sum_{j=1}^{m} |y_i - g_\theta(x_i, \varepsilon_{i,j})| - \frac{1}{2m(m-1)} \sum_{j \neq j'} |g_\theta(x_i, \varepsilon_{i,j}) - g_\theta(x_i, \varepsilon_{i,j'})| \right].$$

Once the network $g_\theta$ is trained, the conditional distribution at a test point $x$ can be approximated by generating $M$ independent samples $\varepsilon_1, \ldots, \varepsilon_M \overset{\text{i.i.d.}}{\sim} \text{Unif}[0, 1]$ and evaluating $g_\theta(x, \varepsilon_m)$ for each $m \in \{1, \ldots, M\}$.

To align with our setting, we interpret their covariate dimension $d$ as our notation $d_0$, so that each design point $x_i \in [0, 1]^{d_0}$. In our sequence-based setting, this leads to the following estimator of the conditional cumulative distribution function:

$$\hat{F}_i^{\text{EnG}}(t) := \frac{1}{M} \sum_{m=1}^{M} \mathbf{1}\{g_\theta(x_i, \varepsilon_m) \leq t\},$$

which provides an approximation to the true conditional CDF $F_i^*(t) := \mathbb{P}(y_i \leq t \mid x_i)$. This estimator is directly comparable to our method's output $\hat{F}_i(t)$, and both aim to recover the conditional distribution function evaluated at $x_i$ for any threshold $t \in \mathbb{R}$.

Specifically, under the assumptions that the true function $g^*$ is Lipschitz continuous in both arguments and that the marginal distribution of covariates has a density bounded away from zero and infinity on its support, [58] shows that the estimator $\hat{F}_i^{\text{EnG}}(t)$ converges in probability to $F_i^*(t)$ for any fixed threshold $t \in \mathbb{R}$.

Translating these conditions to our notation, where $F_i^*(t) = G^*(x_i, t)$, the assumption that $g^*$ is Lipschitz in its first argument implies that the map $x \mapsto G^*(x, t)$ is Lipschitz for every fixed $t$. In contrast, our structural assumption in Assumption 1 posits that $G(\cdot, t) \in \mathcal{H}(l, \mathcal{P})$, a rich class of hierarchical compositions of functions. The Lipschitz class assumed in Engression corresponds to the special case $\mathcal{H}(1, \{(1, d_0)\})$. Therefore, our framework subsumes the setting considered in Engression and allows for more expressive function classes beyond global Lipschitz continuity. Additionally, we do not require the covariate distribution to be bounded away from zero. Importantly, our estimator comes with high-probability guarantees on the $\ell_2$ error over the full CDF vector $F^*(t) = (F_1^*(t), \ldots, F_n^*(t))^\top$. Specifically, we establish that

$$\sup_{t \in \mathbb{R}} \sum_{i=1}^{n} \frac{1}{n} \left( \hat{F}_i(t) - F_i^*(t) \right)^2 = O_{\mathbb{P}} \left( \frac{\log n}{n} + \phi_n \log^4 n \right),$$

where the term $\phi_n$ depends on the complexity of the class $\mathcal{H}(l, \mathcal{P})$. In particular, when $G(\cdot, t)$ is Lipschitz, we obtain $\phi_n = n^{-2/(2+d_0)}$.

A summary of this comparison is provided in Table 2.

## E.2 Relation between Theorem 1 and Theorem A.1 in Guntuboyina et al. [23]

Our Theorem 1 is conceptually related to, but technically distinct from, the high-probability bound established in Theorem A.1 of Guntuboyina et al. [23] (restated as Theorem 5). Their setting considers

the estimation of a mean vector $\mu \in \mathbb{R}^n$ based on noisy Gaussian observations $Y = \mu + Z$, where $Z \sim \mathcal{N}(0, I_n)$. The estimator $\widehat{\mu}$ is the Euclidean projection of $Y$ onto a convex set $K \subset \mathbb{R}^n$, and the goal is to bound the squared error $\|\widehat{\mu} - \mu\|_2^2$ in expectation, which is then strengthened to a high-probability bound.

To achieve this, they define the random process

$$H(t) := \sup_{\nu \in K : \|\nu - \mu\| \leq t} Z^\top (\nu - \mu) - \frac{t^2}{2},$$

which is strictly concave in $t$ and attains its unique maximum at $t^* = \|\widehat{\mu} - \mu\|_2$. The corresponding deterministic function

$$f_\mu(t) := \mathbb{E}[H(t)]$$

is also concave and attains its maximum at $t_\mu$, which characterizes the typical estimation scale. The key analytical step in Guntuboyina et al. [23] is to show that $t^*$ concentrates sharply around $t_\mu$ via a peeling argument: within a local window $[t_\mu - x\sqrt{t_\mu}, \, t_\mu + x\sqrt{t_\mu}]$, stochastic fluctuations of $H(t) - f_\mu(t)$ are controlled by Gaussian concentration, and outside this window, the deterministic function $f_\mu(t)$ exhibits quadratic decay. This interplay between curvature and Gaussian concentration yields strong probabilistic control of the projection error.

Our framework preserves the geometric structure of convex projection but differs in several essential respects. First, the observations in our setting are

$$w(t) = (1\{y_i \leq t\})_{i=1}^n,$$

which are Bernoulli random variables that play the role of $Y$. The associated noise vector $\epsilon(t) = w(t) - F^*(t)$ has independent, mean-zero, heteroskedastic components with variances $F_i^*(t)(1 - F_i^*(t))$. This noise is sub-Gaussian but not Gaussian, and thus lacks the rotational invariance and linearity properties exploited in Guntuboyina et al. [23].

Second, while their goal is to control the squared Euclidean error $\|\widehat{\mu} - \mu\|_2^2$, our analysis focuses on bounding the expected CRPS loss between the true conditional distribution functions $F_i^*$ and the estimators $\widehat{F}_i$, which serve as the natural analogues of $\mu$ and $\widehat{\mu}$ in our context. The non-Gaussianity of the noise, the use of the CRPS loss, and the absence of rotational symmetry make the direct application of their argument infeasible.

In Theorem 1, our proof reduces the problem to controlling the same central quantity, namely the squared error $\|\widehat{F} - F^*\|_2$, which corresponds to $t^* = \|\widehat{\mu} - \mu\|_2$ in their notation. The key scale parameter in our analysis, denoted $\eta$, balances stochastic variability and approximation bias, serving a role analogous to $t_\mu$ in their framework. We then show that $\|\widehat{F} - F^*\|_2^2$ concentrates sharply around $\eta^2$ using a peeling-style argument similar in spirit to theirs.

However, because of the heteroskedastic and non-Gaussian nature of our noise, the technical realization of this step is entirely different. Our analysis relies on

1. a sub-Gaussian concentration inequality for Lipschitz and separately convex functions, used to control the local empirical process $\sup_{\theta \in \Theta_{F^*(t)}(\eta)} \epsilon(t)^\top (\theta - F^*(t))$; and

2. a star-shaped self-normalization argument that extends this local control uniformly over the full constraint set $K_t$.

In summary, both approaches employ a peeling argument to control the projection error, but they differ fundamentally in probabilistic structure and analytical tools. Theorem A.1 of Guntuboyina et al. [23] exploits Gaussian concavity and symmetry, whereas our proof adapts non-Euclidean empirical process methods to accommodate sub-Gaussian, heteroskedastic noise within the CRPS-based framework of distributional regression. The novelty of our result lies in extending the high-probability convex-projection analysis to this more general and statistically heterogeneous setting.

Therefore, Theorem 1 can be viewed as a CRPS-based analogue of Theorem A.1 in Guntuboyina et al. [23], adapted to the non-Gaussian, heteroscedastic, and distributional-regression setting considered in this paper.

Table 2: Comparison of convergence rates for isotonic, trend filtering, and dense ReLU network methods in distributional regression. The "Old" rows refer to existing methods in the literature, while "Our" rows describe the corresponding results introduced in this work.

| | | | |
|---|---|---|---|
| **ISOTONIC** | | | |
| | METHOD | ASSUMPTIONS | CONVERGENCE RATE |
| PRIOR WORK | [28] | $K_t$ IS A MONOTONE CONE $F^*(t) \in K_t$ | $\mathrm{o}(1)$ |
| | [44] | $K_t$ IS A MONOTONE CONE $F^*(t) \in K_t$, DENSE DESIGN HÖLDER CONTINUITY | $n^{-2\alpha/(2\alpha+1)}$ (WORSE-CASE MSE) |
| THIS WORK | GENERAL RATE | $K_t$ IS A MONOTONE CONE $F^*(t) \in K_t$ | $n^{-2/3}$ (CRPS, WORSE-CASE MSE) |
| | FAST RATE | FEW STRICT INCREASES WITH THE NUMBER OF STRICT INCREASES $k(t) \ll n$ | $\frac{1+k(t)}{n} \log\left(\frac{en}{1+k(t)}\right)$ (CRPS) |
| **TREND FILTERING** | | | |
| | METHOD | ASSUMPTIONS | CONVERGENCE RATE |
| PRIOR WORK | [42, 62, 23] | $F_r := \left\{\theta \in \mathbb{R}^n : \mathrm{TV}^{(r)}(\theta) \le V\right\}$ $f \in F_r$ WITH $f$ THE REGRESSION FUNCTION $y_i = f(x_i) + \epsilon_i$ | $\frac{V^{2/(2r+1)}}{n^{2r/(2r+1)}}$ (MSE) |
| | [23] | SPARSE $D^{(r)}f$ MIN SEGMENT LENGTH | $\frac{s+1}{n}\log\left(\frac{en}{s+1}\right)$ (MSE) |
| THIS WORK | GENERAL RATE | $K_t := \left\{\theta \in \mathbb{R}^n : \mathrm{TV}^{(r)}(\theta) \le V_t\right\}$ $F^*(t) \in K_t$ | $\frac{V^{2/(2r+1)}}{n^{2r/(2r+1)}} + \frac{\log n}{n}$ (CRPS, WORSE-CASE MSE) |
| | FAST RATE | SPARSE $D^{(r)}F^*(t)$ MIN SEGMENT LENGTH | $\frac{s+1}{n}\log\left(\frac{en}{s+1}\right)$ (CRPS) |
| **DENSE ReLU NETWORKS** | | | |
| | METHOD | ASSUMPTIONS | CONVERGENCE RATE |
| PRIOR WORK | [37] | REGRESSION $y_i = f(x_i) + \epsilon_i$ $f \in \mathcal{H}(l, \mathcal{P})$ | $\phi_n$ (MSE) |
| | [10] | $F_i^*(t) = G^*(x_i, t) = \mathbb{P}(y_i \le t \mid x_i)$ $G^*(\cdot, t) \in \mathcal{H}(1, \{1, d_0\})$ FOREST-WEIGHTED EMPIRICAL CDF | $\mathrm{o}(1)$ |
| | [58] | $F_i^*(t) = G^*(x_i, t) = \mathbb{P}(y_i \le t \mid x_i)$ $G^*(\cdot, t) \in \mathcal{H}(1, \{1, d_0\})$ NEURAL NET-BASED SAMPLING | $\mathrm{o}(1)$ |
| THIS WORK | GENERAL RATE | $F_i^*(t) = G^*(x_i, t) = \mathbb{P}(y_i \le t \mid x_i)$ $G^*(\cdot, t) \in \mathcal{H}(l, \mathcal{P})$ | $\frac{\log n}{n} + \phi_n \log^4 n$ (CRPS, WORSE-CASE MSE) |

# F  Additional Numerical Results

This appendix provides an extensive evaluation of the proposed method, **UnifDR**, including additional results and analyses omitted from Sections 4 and 5. These supplementary results further demonstrate the effectiveness and robustness of our methods across diverse settings. The appendix presents:

- Additional real data applications to illustrate the practical utility of the proposed methods (Appendix F.1).

- Comprehensive evaluations on alternative evaluation sets ($\Lambda_1$ and $\Lambda_3$) supplementing the results for $\Lambda_2$ in Section 4. This section also includes missing evaluation results for $\Lambda_2$ related to the CRPS metric (Appendix F.2).

- Performance results based on the Maximum Squared Difference (MSD) metric across all scenarios, which were not included in Section 4 (Appendix F.3).

## F.1 Additional Real Data Applications

This appendix presents two additional real-world data applications to further demonstrate the effectiveness of the proposed methods. These examples span different domains, illustrating the versatility and robustness of our approach. Each case study includes a description of the dataset, the experimental setup, and a comparative performance analysis.

For each dataset, the data is randomly divided into a training subset (75%) and a testing (25%) subset. Model performance is evaluated using the empirical cumulative distribution function (CDF), $w_i(t)$, computed over 100 evenly spaced points in a predefined set $\Lambda$. The performance of each competitor is assessed using the Continuous Ranked Probability Score (CRPS) and the Maximum Squared Difference (MSD) metrics, defined as follows:

$$\text{CRPS} = \frac{1}{|\text{Test}|} \sum_{i \in \text{Test}} \frac{1}{100} \sum_{t \in \Lambda} \left( \widehat{F}_i(t) - w_i(t) \right)^2,$$

and

$$\text{MSD} = \max_{t \in \Lambda} \frac{1}{|\text{Test}|} \sum_{i \in \text{Test}} \left( \widehat{F}_i(t) - w_i(t) \right)^2.$$

The proposed **UnifDR** method is implemented in two variants: Trend Filtering, which captures smooth variations through total variation regularization, and Dense ReLU Networks, which leverages a deep neural network to incorporate covariate information.

### F.1.1 Chicago Crime Data with Dense ReLU Networks Approach

We revisit the `2015 Chicago crime dataset`, previously analyzed in the main text using the Trend Filtering approach. We remember that the dataset contains reported crimes in Chicago throughout 2015. As before, the spatial domain is discretized into a $100 \times 100$ grid, where each grid cell represents an aggregated crime count. The response variable remains the log-transformed total crime counts per grid cell, and grid cells with zero observed crimes are excluded, yielding a final dataset of 3,844 grid cells.

Unlike the Trend Filtering approach which assumes a smooth index trend, the Dense ReLU network approach leverages covariate information for modeling crime intensity. The following covariates are included. Latitude and Longitude Bins, encapsulating spatial crime patterns. Day of the Week, represented using dummy variables for each weekday (Monday through Sunday). Beat, a categorical identifier for Chicago's policing districts. Arrest Indicator, a binary variable denoting whether an arrest was made (1) or not (0).

The dataset is randomly split into 100 train (75%) – test (25%) partitions, with evaluation conducted at evenly spaced points $\Lambda$ ranging from -1 to 6. The Dense ReLU Networks approach employs a fully connected feedforward architecture with five hidden layers of 64 neurons each, using ReLU activations. The model is trained using the Adam optimizer with a learning rate of 0.001 over 1,000 epochs, minimizing the Binary Cross-Entropy (BCE) loss function for improved CDF estimation.

Performance is assessed using the Continuous Ranked Probability Score (CRPS) and Maximum Squared Difference (MSD) metrics, comparing estimated CDFs $\widehat{F}_i(t)$ against empirical indicators $w_i(t)$, where $t \in \Lambda$. The methods CART, MARS, RF, DRF, and EnG serve as competitors for the Dense ReLU Networks approach. Table 3 summarizes the results demonstrating the superior performance of the Dense ReLU network relative to classical nonparametric regression methods. Additionally, Figure 3 visualizes $\widehat{F}_i(t)$ for $t = 3$ across test grid cells for all competitors.

### F.1.2 California Housing Prices

We evaluate the effectiveness of the proposed **UnifDR** method by analyzing the 1990 California housing dataset, which contains demographic and economic information from various neighborhoods across California. Originally introduced in Pace and Barry [45], the dataset comprises 20,640 observations and is publicly available via the Carnegie Mellon StatLib reposi-

**Predictions of the CDF of Chicago Crime Data (Dense ReLU Networks)**

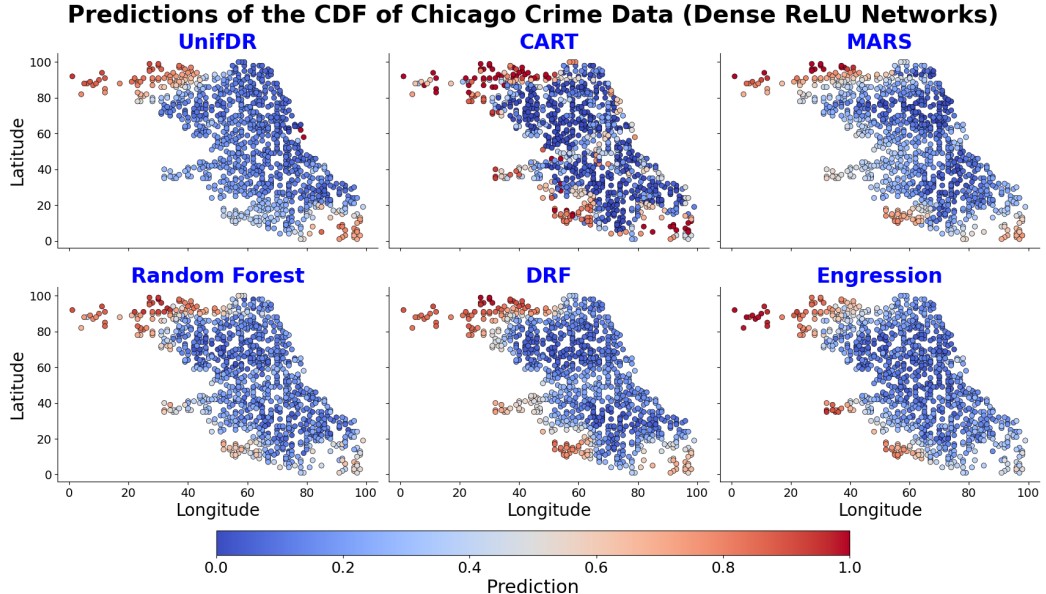

Figure 3: $\widehat{F}_i(t)$ for $t = 3$ and all $i \in$ Test, for all competitors.

**Predictions of the CDF of Chicago Crime Data (Trend Filtering)**

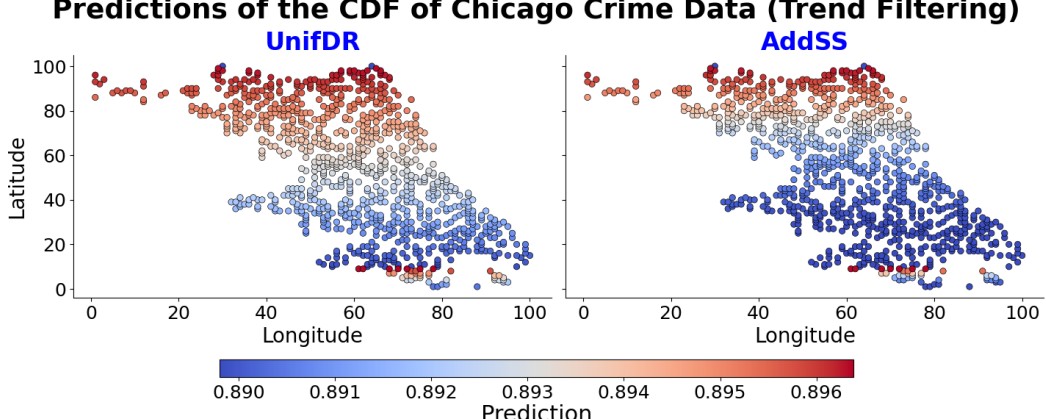

Figure 4: $\widehat{F}_i(t)$ for $t = 3$ and all $i \in$ Test, for all competitors for the example in Section 5.

tory at `http://lib.stat.cmu.edu/datasets/`, as well as the `https://www.dcc.fc.up.pt/~ltorgo/Regression/cal_housing.html` portal.

Following the approach of Ye and Padilla [66], the geographic area is discretized into a $200 \times 200$ spatial grid based on latitude and longitude coordinates. The response variable is derived by applying a logarithmic transformation to the median house values within each grid cell to enhance numerical stability and interpretability. Grid cells lacking valid data are excluded from further analysis, resulting in a final sample size of 3,165 grid cells.

In the Trend Filtering approach no covariates are included. Instead, the spatial grid is treated as an ordered sequence, allowing total variation regularization to capture smooth spatial variations in housing prices. In contrast, the Dense ReLU Networks method integrates spatial features such as latitude and longitude with socioeconomic attributes like `median_income` and `average_occupancy`, which is computed as population divided by households. This approach enables the model to capture complex relationships influencing housing prices. The Dense ReLU neural network consists of three hidden layers, each containing 30 neurons followed by a ReLU activation function.

Evaluations are performed over 100 evenly spaced points within the range $\Lambda = [5, 15]$.

Table 3: Evaluation metrics for UnifDR (Dense ReLU networks) and its competitors on the Chicago crime dataset.

| Method | CRPS (Mean ± Std) | MSD (Mean ± Std) |
|--------|-------------------|-------------------|
| UnifDR | **0.0811± 0.0018** | **0.2133 ± 0.0031** |
| CART | 0.0951 ± 0.0017 | 0.2622 ± 0.0071 |
| DRF | 0.0906 ± 0.0018 | 0.2477 ± 0.0042 |
| ENG | 0.1014 ± 0.0028 | 0.2652 ± 0.0045 |
| MARS | 0.0974 ± 0.0019 | 0.2732 ± 0.0047 |
| RF | 0.0934 ± 0.0020 | 0.2581 ± 0.0031 |

Table 4: Evaluation metrics for UnifDR (Trend Filtering) and its competitor AddSS on the California housing dataset.

| Method | CRPS (Mean ± Std) | MSD (Mean ± Std) |
|--------|-------------------|-------------------|
| UnifDR | **0.0343± 0.0013** | **0.2505 ± 0.0097** |
| AddSS | 0.0357 ± 0.0028 | 0.2652 ± 0.0407 |

Figure 5 presents the estimated cumulative distribution functions $\widehat{F}_i(t)$ at $t = 12$, comparing the performance of **UnifDR** (Trend Filtering) against its competitor, AddSS. Similarly, Figure 6 illustrates the estimated cumulative distribution functions at the same evaluation point, this time comparing **UnifDR** (Dense ReLU networks) against CART, MARS, RF, DRF, and EnG. The corresponding evaluation metrics, summarized in Tables 4 and 5, demonstrate that **UnifDR** consistently outperforms all competitors in terms of both CRPS and MSD.

**Predictions of the CDF of California Housing Prices (Trend Filtering)**

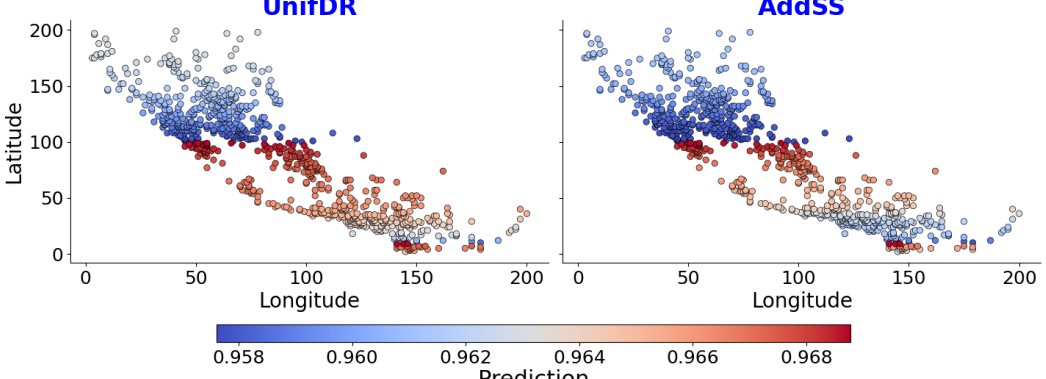

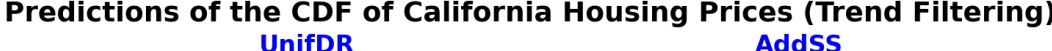

Figure 5: Estimated distribution function $\widehat{F}_i(t)$ at $t = 12$ for all grid cells in the test set, comparing UnifDR (Trend Filtering) and the AddSS competitor.

### F.1.3 Ozone Data Analysis

We further evaluate the effectiveness of the proposed **UnifDR** method by analyzing ozone concentration data collected from the Environmental Protection Agency (EPA) Regional dataset. This dataset consists of daily ozone measurements collected across various monitoring stations in different regions of the United States for the year 2024. The dataset includes measurements of ozone concentration along with associated variables such as Air Quality Index (AQI), wind speed, temperature, latitude, and longitude. The available variables include: State Code, County Code, Site Number, Latitude, Longitude, Date (Year, Month, Day), Ozone concentration (in parts per million), AQI, Wind Speed (in miles per hour), Temperature (in Fahrenheit), Observation Percentage (percentage of valid observations for a given day), First Maximum Value (highest ozone level recorded in a day), First Maximum Hour (time at which the maximum value was recorded), and Observation Count (number of valid ozone measurements per day). The data are publicly accessible via the EPA's AirData portal

**Predictions of the CDF of California Housing Prices (Dense ReLU Networks)**

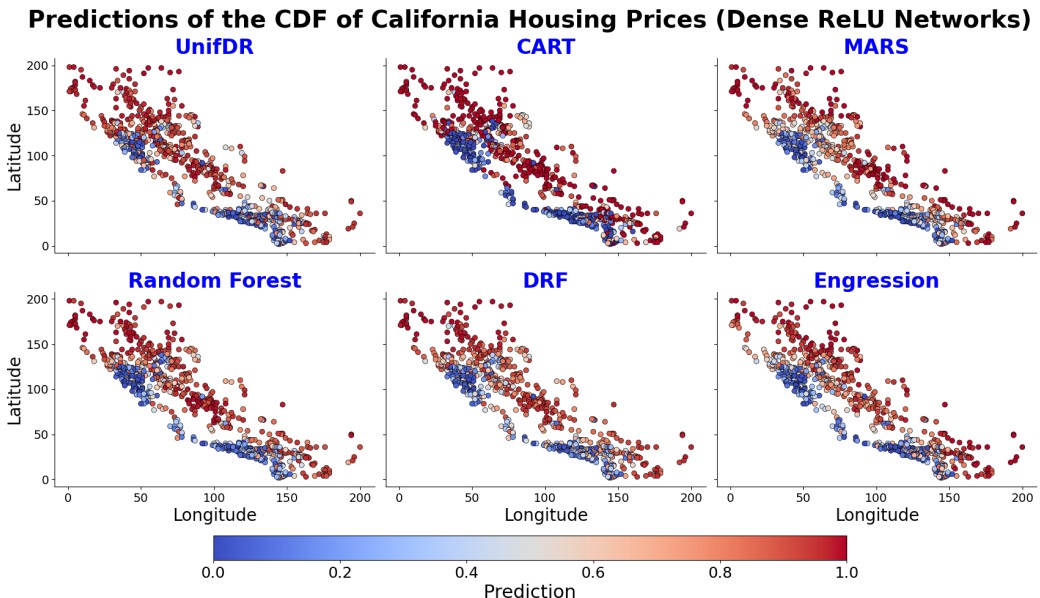

Figure 6: $\widehat{F}_i(t)$ for $t = 12$ and all $i \in$ Test, for all competitors.

Table 5: Evaluation metrics for UnifDR (Dense ReLU networks) and its competitors on the California housing price dataset.

| METHOD | CRPS (MEAN ± STD) | MSD (MEAN ± STD) |
|--------|-------------------|-------------------|
| UNIFDR | **0.0209± 0.00055** | **0.1450 ± 0.0047** |
| CART | 0.0258 ± 0.00053 | 0.1984 ± 0.0067 |
| DRF | 0.0229 ± 0.00051 | 0.1755 ± 0.0045 |
| ENG | 0.0244 ± 0.00059 | 0.1924 ± 0.0057 |
| MARS | 0.0233 ± 0.00057 | 0.1798 ± 0.0099 |
| RF | 0.0221 ± 0.00046 | 0.1650 ± 0.0047 |

at `https://aqs.epa.gov/aqsweb/airdata/download_files.html`, where historical records spanning multiple years can be downloaded.

Following the approach of previous studies, the geographic region of interest is discretized based on latitude and longitude coordinates, specifically within the range $30°N$ to $50°N$ and $-153°W$ to $-70°W$. In the Trend Filtering setup, each monitoring site within these bounds is uniquely identified using a lexicographic ordering of its latitude and longitude values. The response variable $y_i$ for each site $i$ is defined as the mean of the daily recorded ozone levels, ensuring robustness against short-term fluctuations and missing data. This approach results in a total of 1,189 unique monitoring sites, which serve as the basis for spatial trend estimation.

In contrast, the Dense ReLU network method method allows for a flexible spatial fit by incorporating spatial location data as covariates, capturing complex relationships that influence ozone concentration. Specifically, the model uses latitude, longitude, mean AQI, mean percentage of valid observations, mean 1st maximum ozone value, mean 1st maximum hour, and mean observation count as input features. The Dense ReLU network consists of two hidden layers, each containing 100 neurons followed by a ReLU activation function.

Evaluations are performed over 100 evenly spaced points within the range $\Lambda = [0, 1]$. Figures 7 and 8 present the estimated cumulative distribution functions $\widehat{F}_i(t)$ at $t = 0.03$, comparing the performance of **UnifDR** using Trend Filtering and Dense ReLU network against their competitors. Tables 6 and 7 summarize the evaluation metrics, demonstrating that **UnifDR** achieves superior performance in terms of both CRPS and MSD.

**Predictions for Ozone Concentration CDF in the USA (Trend Filtering)**

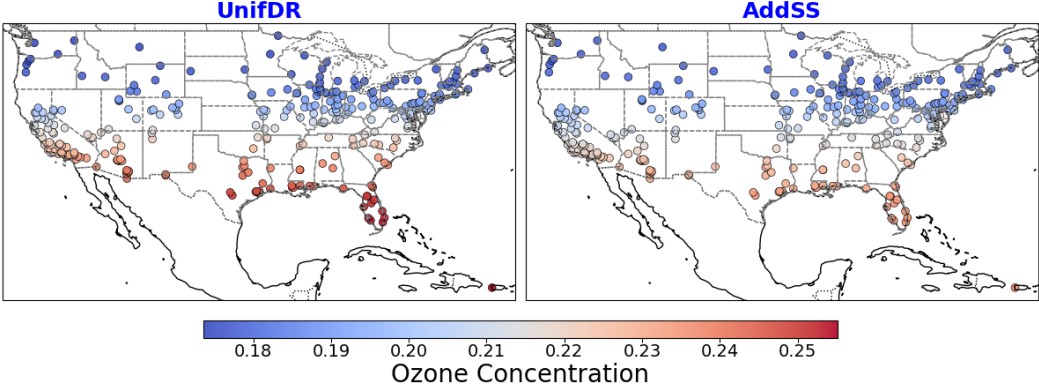

Figure 7: Estimated distribution function $\widehat{F}_i(t)$ for all monitoring sites in the test set, comparing UnifDR (Trend Filtering) and the AddSS competitor.

Table 6: Evaluation metrics for UnifDR (Trend Filtering) and its competitor AddSS on the ozone concentration dataset.

| METHOD | CRPS (MEAN $\pm$ STD) | MSD (MEAN $\pm$ STD) |
|---|---|---|
| UNIFDR | **0.0027$\pm$ 0.0002** | **0.1581 $\pm$ 0.0124** |
| ADDSS | 0.0035 $\pm$ 0.0002 | 0.1987 $\pm$ 0.0127 |

**Predictions for Ozone Concentration CDF in the USA (Dense ReLU Networks)**

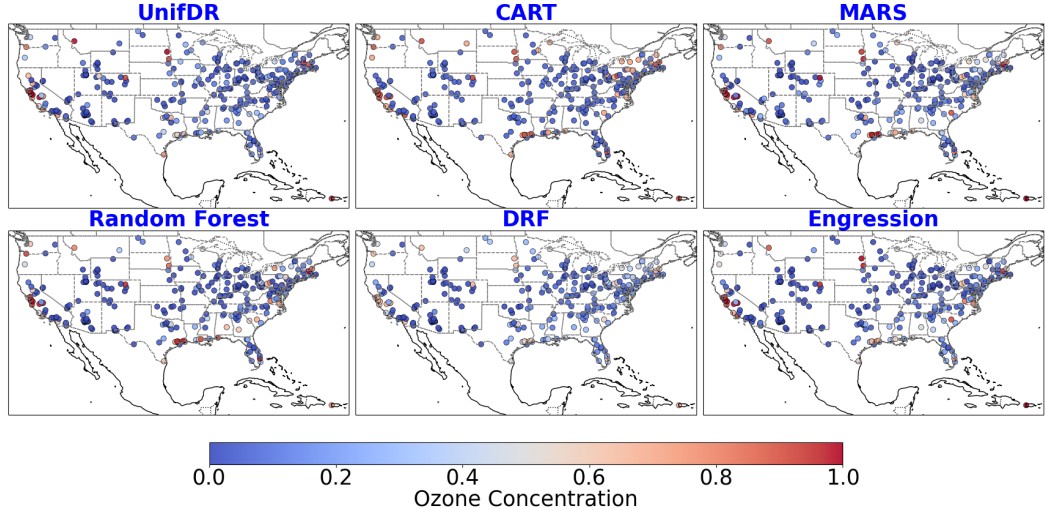

Figure 8: $\widehat{F}_i(t)$ for $t = 0.03$ and all $i \in$ Test, for all competitors.

Table 7: Evaluation metrics for UnifDR (Dense ReLU networks) and its competitors on the Ozone dataset.

| METHOD | CRPS (MEAN $\pm$ STD) | MSD (MEAN $\pm$ STD) |
|---|---|---|
| UNIFDR | **0.0016$\pm$ 0.00021** | **0.0959 $\pm$ 0.0136** |
| CART | 0.0026 $\pm$ 0.00028 | 0.1529 $\pm$ 0.0209 |
| DRF | 0.0031 $\pm$ 0.00016 | 0.1593 $\pm$ 0.0118 |
| ENG | 0.0022 $\pm$ 0.00099 | 0.1097 $\pm$ 0.0603 |
| MARS | 0.0022 $\pm$ 0.00039 | 0.1288 $\pm$ 0.0376 |
| RF | 0.0021 $\pm$ 0.00023 | 0.1012 $\pm$ 0.0157 |

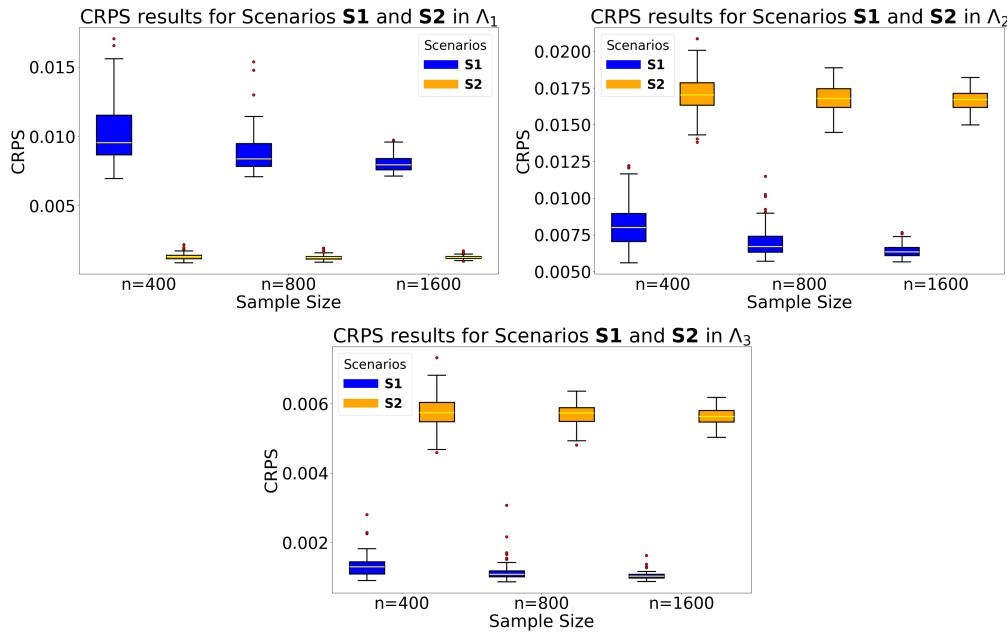

Figure 9: Box plots for CRPS results in Scenarios **S1** and **S2**. The top row shows results for $\Lambda_1$ (left) and $\Lambda_2$ (right), while the bottom row displays results for $\Lambda_3$.

### F.2 CRPS results on evaluation sets $\Lambda_1$, and $\Lambda_3$, and missing results for S1, S1 and S6 in $\Lambda_2$

Section 4 presented results for the evaluation set $\Lambda_2$ using the Continuous Ranked Probability Score (CRPS). However the results for scenarios **S1**, **S2** and **S6** were omitted due to space constraints. Moreover, it is worth noting that the evaluation set $\Lambda_2$ represents a balanced range of values centered around zero. To provide a more comprehensive analysis, this appendix includes the omitted CRPS results for $\Lambda_2$, and the CRPS results for the alternative evaluation sets $\Lambda_1$ and $\Lambda_3$, which emphasize distinct distributional regions.

The new evaluation sets are defined as follows:

- $\Lambda_1$: 100 points evenly spaced between $-1$ and $0.4$, focusing on the lower and middle ranges of the distribution.

- $\Lambda_3$: 100 points evenly spaced between $0.8$ and $10$, capturing the upper tail of the distribution.

This extended analysis provides deeper insights into the robustness of the proposed methods in varying distributional regimes, including regions with lower densities and heavier tails. As described in Section 4, for each evaluation set ($\Lambda_1$, $\Lambda_2$ and $\Lambda_3$), the CRPS is computed and averaged over 100 Monte Carlo repetitions. Figures 9 through 14 summarize the CRPS results across different scenarios and evaluation regions. Specifically, Figure 9 correspond to Scenarios **S1** and **S2** in $\Lambda_1$, $\Lambda_2$, and $\Lambda_3$. Figure 10 presents Scenarios **S3** and **S4** in $\Lambda_1$ and $\Lambda_3$. Figure 11 focuses on Scenario **S5** in $\Lambda_1$ and $\Lambda_3$. Figures 12, 13. and 14 consider Scenario **S6** in $\Lambda_1$, $\Lambda_2$, and $\Lambda_3$, respectively.

Across all scenarios, **UnifDR** is implemented using different estimation methods: the isotonic estimator (Section 3.2.1) for **S1** and **S2**, the trend filtering estimator (Section 3.2.2) for **S3** and **S4**, and the Dense ReLU Networks method (Section 3.3.2) for **S5** and **S6**. Our proposed framework **UnifDR** outperforms competing approaches across all scenarios, evaluation regions, and sample sizes, regardless of the specific estimation method employed. These findings highlight the versatility and robustness of **UnifDR** in adapting to diverse structural patterns within the data. Overall, performance trends remain consistent across $\Lambda_1$, $\Lambda_2$, and $\Lambda_3$, with minor variations due to differences in the underlying data distributions. This extended analysis further reinforces the effectiveness of the proposed methods under varying evaluation conditions.

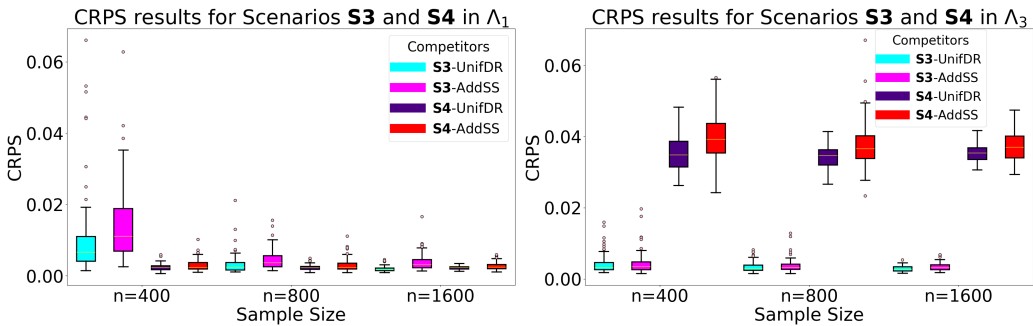

Figure 10: Box plots for simulation results of **S3-S4** for the CRPS metric. The row shows results for $\Lambda_1$ (left) and $\Lambda_2$ (right).

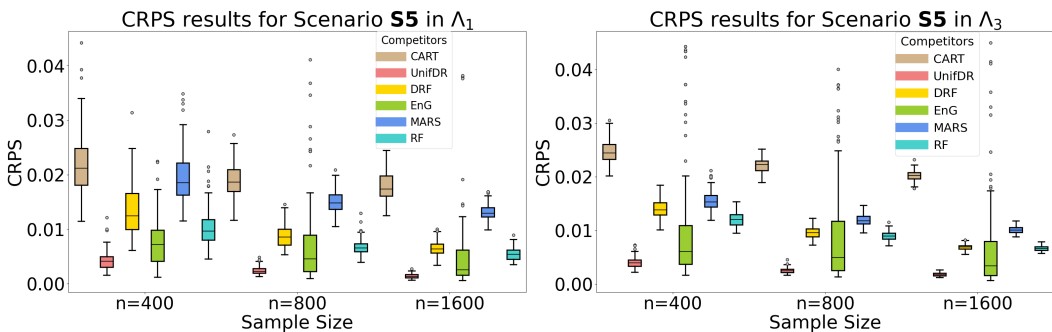

Figure 11: Box plots for simulation results of **S5** for the CRPS metric. The row shows results for $\Lambda_1$ (left) and $\Lambda_2$ (right).

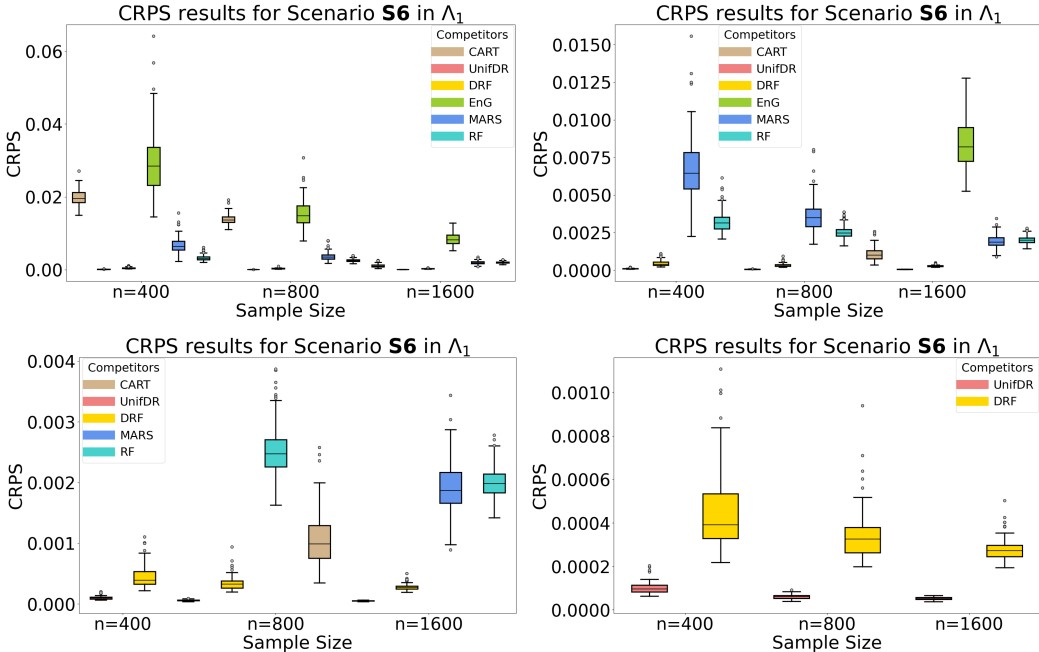

Figure 12: Box plots for simulation results of **S6** for the CRPS metric for the set $\Lambda_1$. The top row shows results for the all the competitors (left) and, competitors with median below $0.01$ (right). The bottom row displays results for competitors with median below $0.0025$ (left), and best two competitors (right).

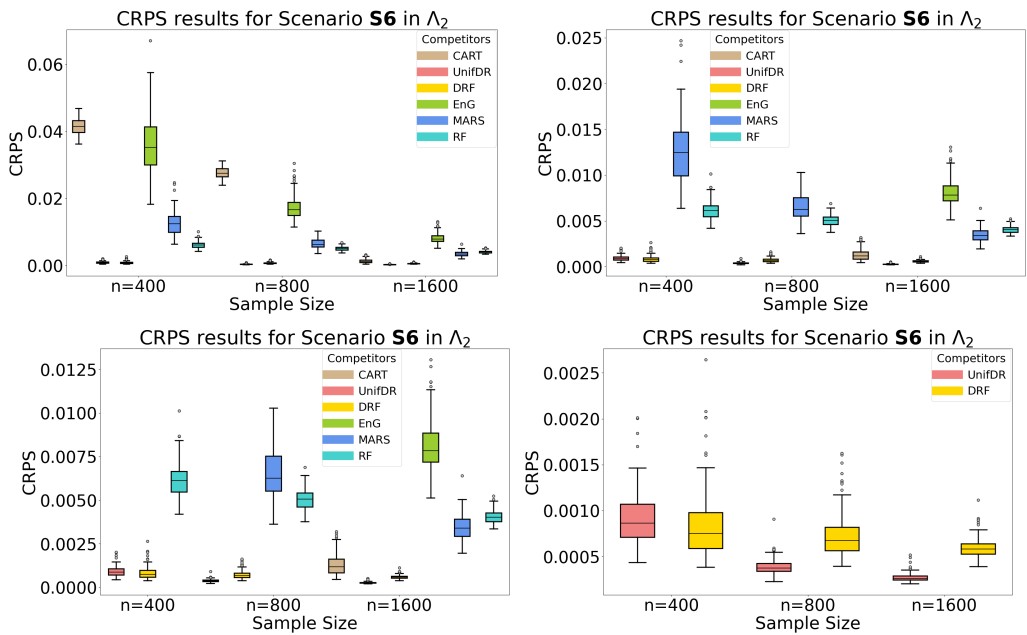

Figure 13: Box plots for CRPS results for **S6** in $\Lambda_2$. The top row shows results for the all the competitors (left) and, competitors with median below $0.02$ (right). The bottom row displays results for competitors with median below $0.01$ (left), and best two competitors (right).

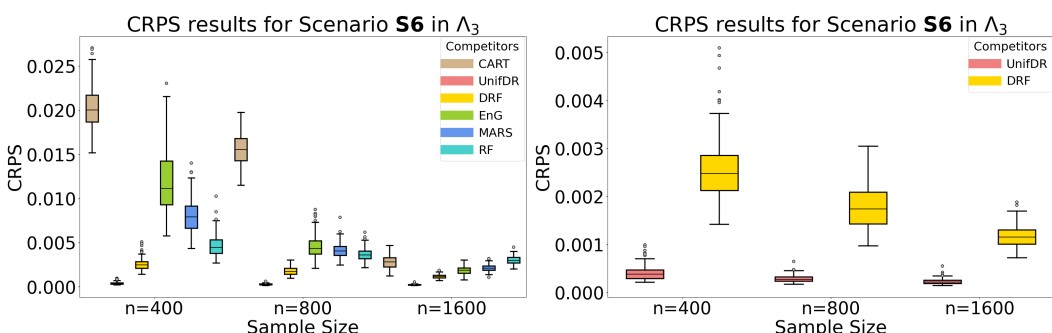

Figure 14: Box plots for simulation results of **S6** for the CRPS metric using evaluation set $\Lambda_3$. The left plot corresponds to all competitors performance, while the right plot corresponds to best two competitors.

### F.3 Additional Results for Maximum Squared Difference (MSD) Metric

This appendix extends the results presented in Section 4 and Appendix F.2 by providing evaluations of the Maximum Squared Difference (MSD) metric across all scenarios for the evaluation sets $\Lambda_1$, $\Lambda_2$, and $\Lambda_3$. The MSD metric measures the worst-case discrepancy between the estimated and true cumulative distribution functions (CDFs), offering a stringent assessment of model accuracy and robustness.

Figure 15 displays box plots of the MSD results for Scenarios **S1** and **S2** across all evaluation sets. As outlined in Section 4, **UnifDR** employs isotonic regression for these scenarios, which lacks direct competitors. The results indicate a decreasing trend in MSD values as the sample size increases, demonstrating the consistency and improved accuracy of isotonic regression with larger datasets. Moreover, variations in MSD across $\Lambda_1$, $\Lambda_2$, and $\Lambda_3$ reflect natural differences in the underlying data distributions.

Figure 16 presents MSD results for Scenarios **S3** and **S4**, where **UnifDR** employs trend filtering and is compared against the additive smoothing splines (AddSS) method. Across all evaluation sets and

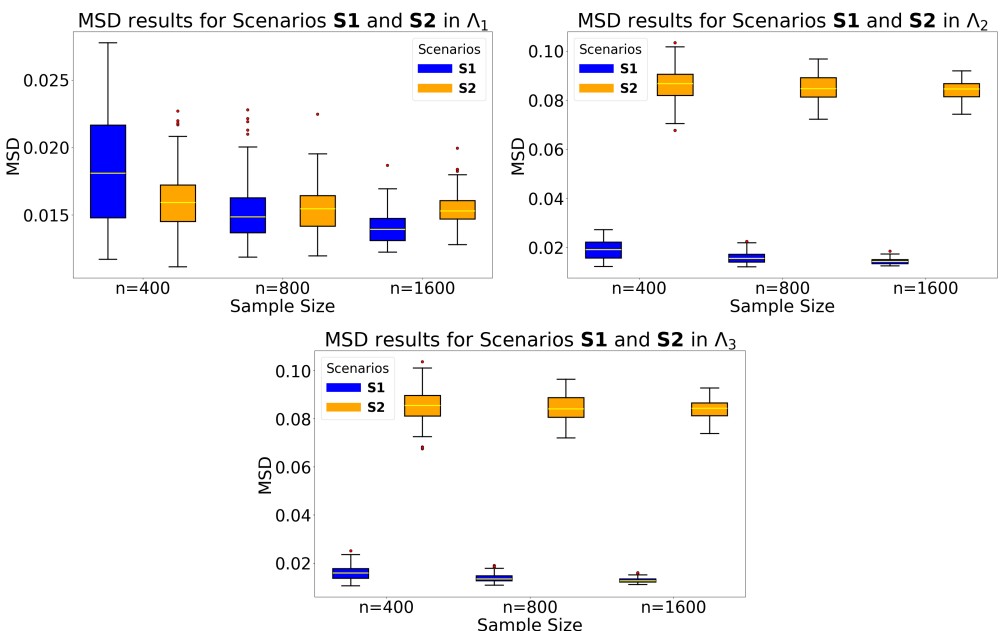

Figure 15: Box plots for MSD results in Scenarios **S1** and **S2**. The top row shows results for $\Lambda_1$ (left) and $\Lambda_2$ (right), while the bottom row displays results for $\Lambda_3$.

sample sizes, **UnifDR** consistently outperforms AddSS, demonstrating its ability to adapt to complex structural variations. Notably, trend filtering exhibits particularly strong performance in regions with sparse or heavy-tailed data distributions.

Figures 17 to 20 summarize MSD results for Scenarios **S5** and **S6**, where **UnifDR** is implemented via Dense ReLU Networks (Section 3.3.2). The results further confirm the superiority of **UnifDR** over all competing methods, maintaining its advantage across different sample sizes and evaluation sets. As observed with the CRPS metric, **UnifDR** effectively captures diverse structural patterns and remains robust in challenging scenarios involving data sparsity and heavy tails.

The inclusion of MSD results provides a comprehensive performance evaluation of **UnifDR**, reinforcing its effectiveness and reliability across various experimental conditions.

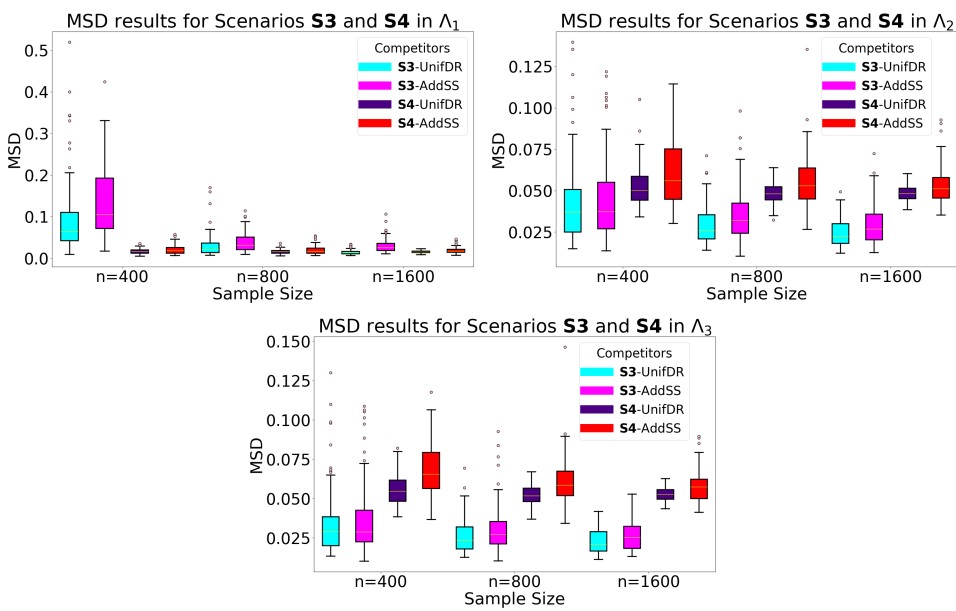

Figure 16: Box plots for MSD in **S3** and **S4**. The top row shows results for $\Lambda_1$ (left) and $\Lambda_2$ (right), while the bottom row displays results for $\Lambda_3$.

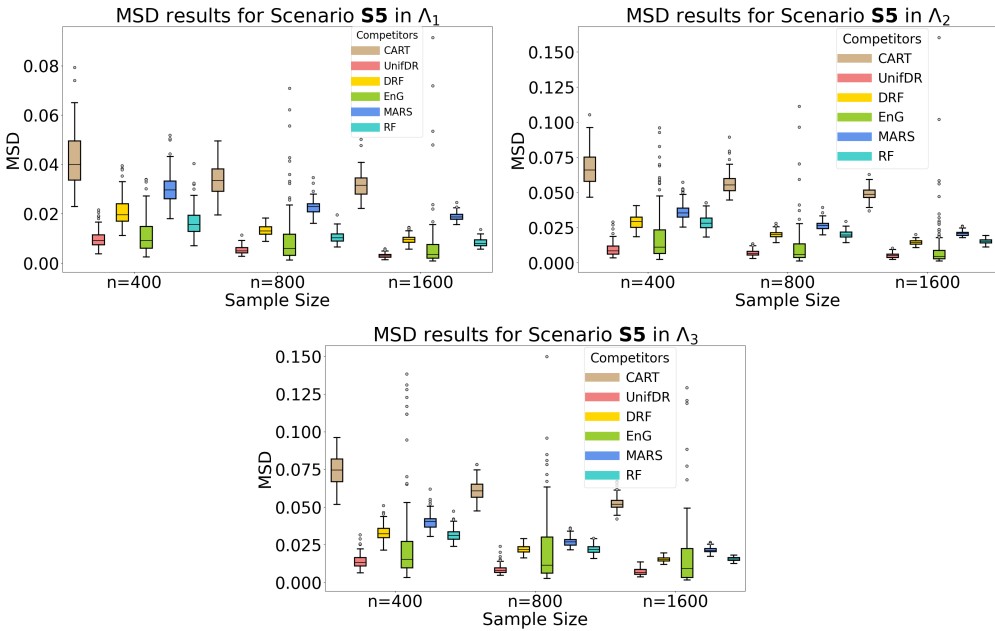

Figure 17: Box plots for MSD in **S5**. The top row shows results for $\Lambda_1$ (left) and $\Lambda_2$ (right), while the bottom row displays results for $\Lambda_3$.

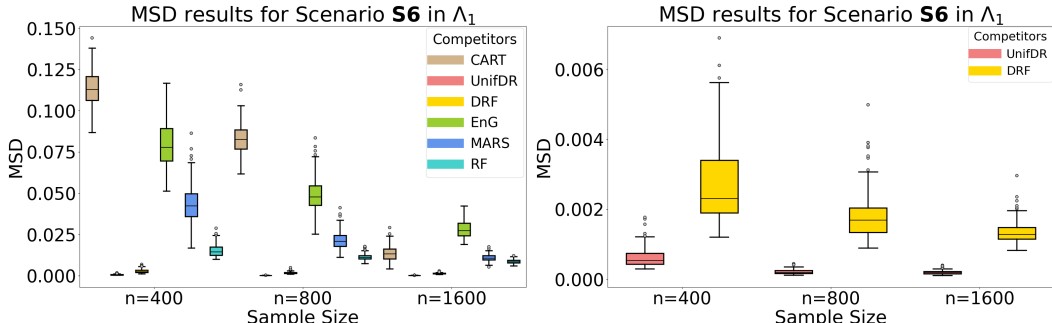

Figure 18: Box plots for MSD in **S6** using evaluation set $\Lambda_1$. The left plot corresponds to all competitors performance, while the right plot corresponds to best two competitors.

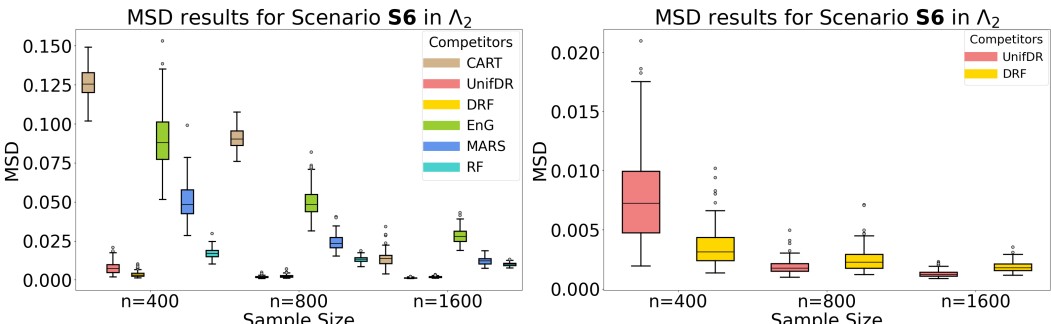

Figure 19: Box plots for MSD in **S6** using evaluation set $\Lambda_2$. The left plot corresponds to all competitors performance, while the right plot corresponds to best two competitors.

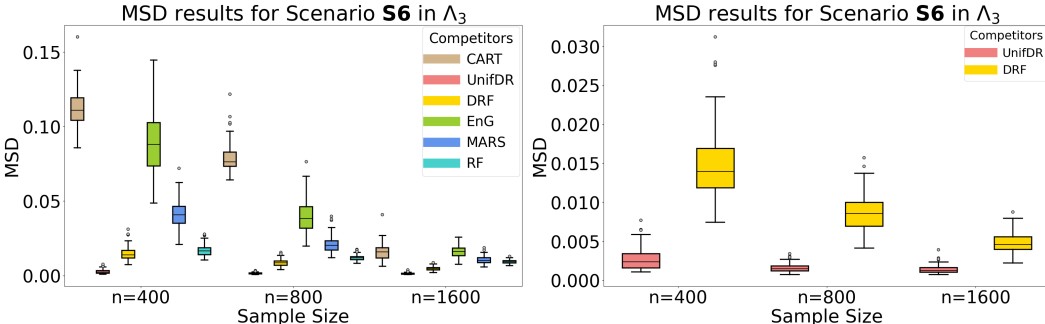

Figure 20: Box plots for MSD in **S6** using evaluation set $\Lambda_3$. The left plot corresponds to all competitors performance, while the right plot corresponds to best two competitors.

# G Proofs

## G.1 Proof of Lemma 1

*Proof.* Let $y_{(1)} \le y_{(2)} \le \ldots \le y_{(n)}$ the order statistics of $y$. Notice that

$$
\begin{aligned}
L(F) &:= \sum_{i=1}^{n} \mathrm{CRPS}(F_i, 1\{y_i \le \cdot\}) \\
&= \int \sum_{i=1}^{n} (F_i(t) - 1\{y_i \le t\})^2 dt \\
&= \sum_{j=1}^{n+1} \int_{A_j} \sum_{i=1}^{n} (F_i(t) - 1\{y_i \le t\})^2 dt
\end{aligned}
$$

where $A_1 = (-\infty, y_{(1)})$, $A_2 = [y_{(1)}, y_{(2)}), \ldots, A_n = [y_{(n-1)}, y_{(n)})$, $A_{n+1} = [y_{(n)}, \infty)$. However, for every $j \in \{1, \ldots, n+1\}$ and $t, t' \in A_j$ we have that

$$
\min_{F(t) \in K} \sum_{i=1}^{n} (F_i(t) - 1\{y_i \le t\})^2 = \min_{F(t') \in K} \sum_{l=1}^{n} (F_i(t') - 1\{y_i \le t'\})^2.
$$

Hence, letting $t_j$ be an element of $A_j$, we obtain that minimizing $L(F)$ with the constraints $F(t) \in K$ for all $t$ is equivalent to solving the independent problems

$$
\min_{F(t_j) \in K} \sum_{i=1}^{n} (F_i(t_j) - 1\{y_i \le t_j\})^2,
$$

and the claim follows. $\qquad\square$

## G.2 Proof of Theorem 1

**Theorem 5.** *[Theorem A.1 in [23]]. There exists a universal positive constant $C > 0$ such that for every $t$,*

$$
\frac{1}{n} \mathbb{E}\left( \sum_{i=1}^{n} \left( \widehat{F}_i(t) - F_i^*(t) \right)^2 \right) \le \frac{C \max\{\eta^2, \max_{i=1,\ldots,n} F_i^*(t)(1 - F_i^*(t))\}}{n},
$$

*for every $\eta > 1$ satisfying*

$$
\mathbb{E}\left[ \sup_{\theta \in K_t : \|\theta - F^*(t)\| \le \eta} \epsilon(t)^\top (\theta - F^*(t)) \right] \le \frac{\eta^2}{2} \tag{26}
$$

*where $\epsilon(t) = w(t) - F^*(t)$.*

In the followig we first present the proof of Theorem 5.

*Proof.* Let $t \in \mathbb{R}$. Define $\sigma^2 = \max_{i=1,\ldots,n} F_i^*(t)(1 - F_i^*(t))$. We consider the following two cases separately based on the value of $\sigma$:

1. $\sigma = 0$,

2. $\sigma \ne 0$.

**Case 1:** $\sigma = 0$.

By definition, $\sigma^2 = \max_{i=1,\ldots,n} F_i^*(t)(1 - F_i^*(t)) = 0$. This implies that for all $i = 1, \ldots, n$, $F_i^*(t)(1 - F_i^*(t)) = 0$. Since $F_i^*(t)(1 - F_i^*(t)) = 0$, it follows that either $F_i^*(t) = 0$ or $F_i^*(t) = 1$ for each $i$. Now observe that for each $i \in \{1, \ldots, n\}$, either $w_i(t) = 0$ or $w_i(t) = 1$. Given that $\mathbb{E}(w_i(t)) = F_i^*(t)$, it follows that $w_i(t) = F_i^*(t)$. Therefore, by the definition of $\widehat{F}$ in Equation 3, we have $\widehat{F} = F^*$. In this case, it holds that

$$
\mathbb{E}\left( \|\widehat{F}(t) - F^*(t)\|_2^2 \right) = 0,
$$

and the result is obtained trivially.

**Case 2:** $\sigma \neq 0$.

Denote by $\Theta_{F^*(t)}(\eta) := \{\theta - F^*(t) \in \Theta_{F^*(t)} : \|\theta - F^*(t)\|_2 \leq \eta\}$, where $\Theta_{F^*(t)} = \{\theta - F^*(t) : \theta \in K_t\}$. Notice that the function $L : \mathbb{R}^n \to \mathbb{R}$ given by

$$\epsilon \to \sup_{\theta \in \Theta_{F^*(t)}(\eta)} \left| \epsilon^\top (\theta - F^*(t)) \right|$$

is $\eta$-Lipschitz. Moreover observe that $L$ is separately convex. In fact, for any $k \in \{1, ...n\}$ we have that

$$(\epsilon_1, ..., \epsilon_{k-1}, t\epsilon_k^{(1)} + (1-t)\epsilon_k^{(2)}, \epsilon_{k+1}, ..., \epsilon_n)$$
$$= (t\epsilon_1, ..., t\epsilon_{k-1}, t\epsilon_k^{(1)}, t\epsilon_{k+1}, ..., t\epsilon_n)$$
$$+ ((1-t)\epsilon_1, ..., (1-t)\epsilon_{k-1}, (1-t)\epsilon_k^{(2)}, (1-t)\epsilon_{k+1}, ..., (1-t)\epsilon_n)$$
$$= t\epsilon_{k,1} + (1-t)\epsilon_{k,2}.$$

Therefore,

$$L\left[ (\epsilon_1, ..., \epsilon_{k-1}, t\epsilon_k^{(1)} + (1-t)\epsilon_k^{(2)}, \epsilon_{k+1}, ..., \epsilon_n) \right]$$
$$= \sup_{\theta \in \Theta_{F^*(t)}(\eta)} \left| (t\epsilon_{k,1} + (1-t)\epsilon_{k,2})^\top (\theta - F^*(t)) \right|$$
$$= \sup_{\theta \in \Theta_{F^*(t)}(\eta)} \left| t(\epsilon_{k,1})^\top (\theta - F^*(t)) + (1-t)(\epsilon_{k,2})^\top (\theta - F^*(t)) \right|$$
$$\leq \sup_{\theta \in \Theta_{F^*(t)}(\eta)} \left[ t \left| (\epsilon_{k,1})^\top (\theta - F^*(t)) \right| + (1-t) \left| (\epsilon_{k,2})^\top (\theta - F^*(t)) \right| \right],$$

where the inequality is followed by triangle inequality. Using the fact that the supremum of the sum is bounded by the sum of the supremums, it follows that

$$L\left[ (\epsilon_1, ..., \epsilon_{k-1}, t\epsilon_k^{(1)} + (1-t)\epsilon_k^{(2)}, \epsilon_{k+1}, ..., \epsilon_n) \right]$$
$$\leq t \sup_{\theta \in \Theta_{F^*(t)}(\eta)} \left| (\epsilon_{k,1})^\top (\theta - F^*(t)) \right| + (1-t) \sup_{\theta \in \Theta_{F^*(t)}(\eta)} \left| (\epsilon_{k,2})^\top (\theta - F^*(t)) \right|$$
$$= tL\left[ (\epsilon_1, ..., \epsilon_{k-1}, \epsilon_k^{(1)}, \epsilon_{k+1}, ..., \epsilon_n) \right] + (1-t)L\left[ (\epsilon_1, ..., \epsilon_{k-1}, \epsilon_k^{(2)}, \epsilon_{k+1}, \epsilon_n) \right].$$

Thus, $L$ is separately convex. Notice that $\epsilon_i(t) = (1\{y_i \leq t\} - F_i^*(t))$ satisfies $|\epsilon(t)| \leq 1$. Hence, by Theorem 3.4 in Wainwright [64], for any $\delta > 0$

$$\sup_{\theta \in \Theta_{F^*(t)}(\eta)} \left| \epsilon(t)^\top (\theta - F^*(t)) \right| \leq \mathbb{E}\left( \sup_{\theta \in \Theta_{F^*(t)}(\eta)} \left| \epsilon(t)^\top (\theta - F^*(t)) \right| \right) + \sigma \eta \delta \qquad (27)$$

with probability at least $1 - e^{-\delta^2 \sigma^2 / 16}$. Next, we have that

$$\left| \epsilon(t)^\top (\theta - F^*(t)) \right| \leq \max\{ \frac{\|\theta - F^*(t)\|_2}{\eta}, 1 \} \widetilde{L}(\eta), \qquad (28)$$

for any $\theta \in K_t$, where $\widetilde{L}(\eta) = \sup_{\theta \in \Theta_{F^*(t)}(\eta)} \left| \epsilon(t)^\top (\theta - F^*(t)) \right|$. This conclusion is derived based on the subsequent line of reasoning. If $\theta \in K_t$ and $\|\theta - F^*(t)\|_2 \leq \eta$, then $\theta - F^*(t) \in \Theta_{F^*(t)}(\eta)$ and inconsequence $|\epsilon(t)^\top (\theta - F^*(t))| \leq \widetilde{L}(\eta)$, by definition of $\widetilde{L}(\eta)$. If $\theta \in K_t$ and $\|\theta - F^*(t)\|_2 > \eta$, then $\frac{\theta - F^*(t)}{\|\theta - F^*(t)\|_2} \cdot \eta \in \Theta_{F^*(t)}(\eta)$ because $\Theta_{F^*(t)}$ is star-shaped, given that $K_t$ is convex. Also $\left\| \frac{\theta - F^*(t)}{\|\theta - F^*(t)\|} \eta \right\|_2 = \eta$. Hence,

$$\left| \epsilon(t)^\top \left( \frac{\theta - F^*(t)}{\|\theta - F^*(t)\|} \eta \right) \right| \leq \widetilde{L}(\eta),$$

which implies,

$$|\epsilon(t)^\top (\theta - F^*(t))| \leq \frac{\|\theta - F^*(t)\|_2}{\eta} \cdot \widetilde{L}(\eta),$$

for any $\theta \in K_n$. Then we observe that by the basic inequality

$$||w(t) - \widehat{F}(t)||_2^2 \le ||w(t) - F^*(t)||_2^2.$$

This implies that

$$\frac{1}{2}||\widehat{F}(t) - F^*(t)||_2^2 \le \epsilon(t)^\top (\widehat{F}(t) - F^*(t)).$$

Given that $\widehat{F}(t) \in K_t$ it follows from inequality (28) that

$$\frac{1}{2}||\widehat{F}(t) - F^*(t)||_2^2 \le \max\{\frac{||\widehat{F}(t) - F^*(t)||_2}{\eta}, 1\}\widetilde{L}(\eta),$$

and by inequality (27),

$$\frac{1}{2}||\widehat{F}(t) - F^*(t)||_2^2 \le \max\{\frac{||\widehat{F}(t) - F^*(t)||_2}{\eta}, 1\} \left( \mathbb{E}\left( \sup_{\theta \in \Theta_{F^*(t)}(\eta)} |\epsilon(t)^\top (\theta - F^*(t))| \right) + \sigma\eta\delta \right),$$

for any $\delta > 0$, with probability at least $1 - e^{-\sigma^2\delta^2/16}$. Thus, for any $\delta > 0$

$$||\widehat{F}(t) - F^*(t)||_2 \le \max\left\{ \frac{2G(\eta, \delta)}{\eta}, \sqrt{2G(\eta, \delta)} \right\},$$

with probability at least $1 - e^{-\sigma^2\delta^2/16}$, where $G(\eta, \delta) = \mathbb{E}\left( \sup_{\theta \in \Theta_{F^*(t)}(\eta)} |\epsilon(t)^\top (\theta - F^*(t))| \right) + \sigma\eta\delta$. Next, by inequality (26),

$$\max\left\{ \frac{2G(\eta, \delta)}{\eta}, \sqrt{2G(\eta, \delta)} \right\} \le \max\left\{ \frac{\eta^2 + 2\sigma\eta\delta}{\eta}, \sqrt{\eta^2 + 2\sigma\eta\delta} \right\} = \max\left\{ \eta + 2\sigma\delta, \sqrt{\eta(\eta + 2\sigma\delta)} \right\} \le \eta + 2\sigma\delta.$$

In consequence for any $\delta > 0$,

$$||\widehat{F}(t) - F^*(t)||_2 \le \eta + 2\sigma\delta, \tag{29}$$

with probability at least $1 - e^{-\sigma^2\delta^2/16}$. Finally, we observe that

$$\mathbb{E}(||\widehat{F}(t) - F^*(t)||_2^2) = \int_0^\infty \mathbb{P}\left( ||\widehat{F}(t) - F^*(t)||_2^2 > s \right) ds$$

$$= \int_0^{(\eta+2\sigma)^2} \mathbb{P}\left( ||\widehat{F}(t) - F^*(t)||_2^2 > s \right) ds + \int_{(\eta+2\sigma)^2}^\infty \mathbb{P}\left( ||\widehat{F}(t) - F^*(t)||_2^2 > s \right) ds$$

$$= I_1 + I_2.$$

To analyze the term $I_1$ we observe that $\mathbb{P}\left( ||\widehat{F}(t) - F^*(t)||_2^2 > s \right) \le 1$, and therefore

$$I_1 \le (\eta + 2\sigma)^2. \tag{30}$$

For the term $I_2$, we perform a change of variables $s = (\eta + 2\sigma\delta)^2$ to obtain

$$I_2 \le \int_1^\infty \mathbb{P}\left( ||\widehat{F}(t) - F^*(t)||_2^2 > (\eta + 2\sigma\delta)^2 \right) 4\sigma(\eta + 2\sigma\delta)d\delta,$$

and by inequality (29),

$$I_2 \le 4\sigma \int_1^\infty e^{-\frac{\sigma^2\delta^2}{16}}(\eta + 2\sigma\delta)d\delta. \tag{31}$$

Moreover,

$$4\sigma \int_1^\infty e^{-\frac{\sigma^2\delta^2}{16}}(\eta + 2\sigma\delta)d\delta = 4\eta\sigma \int_1^\infty e^{-\frac{\sigma^2\delta^2}{16}}d\delta + 8\sigma^2 \int_1^\infty e^{-\frac{\sigma^2\delta^2}{16}}\delta d\delta$$

$$= 2\eta\frac{\sqrt{\pi}\sigma\,\mathrm{erfc}(\sigma)}{\sigma} + 4e^{-\sigma^2}$$

$$\le 2\eta\sqrt{\pi}\,\mathrm{erfc}(\sigma) + 4e^{-\sigma^2}.$$

where $\mathrm{erfc}(z) = \frac{2}{\sqrt{\pi}} \int_z^\infty e^{-\delta^2} d\delta$. From inequality (30) (31), and the fact that $\eta > 1$, we conclude that

$$\mathbb{E}(\|\widehat{F}(t) - F^*(t)\|_2^2) \leq C_1(\eta^2 + \sigma^2),$$

for an absolute positive constant $C_1$. Finally observe that,

$$\eta^2 + \sigma^2 \leq 2 \max\left(\eta^2, \sigma^2\right)$$

Taking $C = 2C_1$ the result is achieved. $\qquad\square$

Now we are ready to start the proof of Theorem 1.

*Proof.* Notice that for all $i$, it holds that $\widehat{F}_i(t) = F^*(t) = 0$ for all $t < \inf\{a \ : \ a \in \Omega\}$ and $\widehat{F}_i(t) = F^*(t) = 1$ for all $t > \sup\{a \ : \ a \in \Omega\}$. Hence,

$$
\begin{aligned}
\mathbb{E}\left(\frac{1}{n} \sum_{i=1}^n \mathrm{CRPS}(\widehat{F}_i, F_i^*)\right) &= \mathbb{E}\left(\frac{1}{n} \sum_{i=1}^n \int_{-\infty}^\infty (\widehat{F}_i(t) - F_i^*(t))^2 dt\right) \\
&= \mathbb{E}\left(\frac{1}{n} \sum_{i=1}^n \int_\Omega (\widehat{F}_i(t) - F_i^*(t))^2 dt\right) \\
&= \int_\Omega \mathbb{E}\left(\frac{1}{n} \sum_{i=1}^n (\hat{F}_i(t) - F_i^*(t))^2\right) dt \\
&\leq \int_\Omega \frac{C \max\{1, \eta^2\}}{n} dt \\
&= \frac{C \max\{1, \eta^2\}}{n} \int_\Omega dt
\end{aligned}
$$

where the inequality follows from Theorem 5, by noticing that (6) and Lemma 4 imply (26). $\qquad\square$

### G.3 Proof of Corollary 1

*Proof.* Throughout we use the notation from Definitions 5 and 6.

First, notice that $\widehat{F}_i(t) = \widetilde{F}_i(t) = 0$ for all $t < y_{(1)}$ and for all $i$. Similarly, $\widehat{F}_i(t) = \widetilde{F}_i(t) = 1$ for all $t \geq y_{(n)}$ and for all $i$. Therefore,

$$\int_{-\infty}^{y_{(1)}} (\widehat{F}_i^+(t) - F_i^*(t))^2 + \int_{y_{(n)}}^\infty (\widehat{F}_i^+(t) - F_i^*(t))^2 = \int_{-\infty}^{y_{(1)}} (\widetilde{F}_i(t) - F_i^*(t))^2 + \int_{y_{(n)}}^\infty (\widetilde{F}_i(t) - F_i^*(t))^2. \tag{32}$$

Next, define

$$\widehat{G}_i(t) := \begin{cases} \widehat{F}_i^+((1-t)(y_{(n)} - y_{(1)}) + y_{(1)}) & \text{for } t \in [0, 1), \\ 0 & \text{otherwise.} \end{cases}$$

Clearly, $\widehat{G}_i(0) = \widehat{F}_i^+(y_{(n)})$ and $\widehat{G}_i(1) = \widehat{F}_i^+(y_{(1)})$. Moreover, recalling that for $t \in [y_{(1)}, y_{(n)})$, we can write

$$\widehat{F}_i^+(t) = \sum_{k=1}^{n-1} a_{i,j_k} 1_{[y_{(j_k)}, y_{(j_k+1)})}(t),$$

then for $t \in [0, 1)$, it holds that

$$\widehat{G}_i(t) := \sum_{k=1}^{n-1} a_{i,j_k} 1_{[u_{j_k+1}, u_{j_k})}(t) \tag{33}$$

where

$$u_l := 1 - \frac{y_{(l)} - y_{(1)}}{y_{(n)} - y_{(1)}}$$

for $l \in \{1, \ldots, n\}$.

Furthermore, let

$$G_i^*(t) := \begin{cases} F_i^*((1-t)(y_{(n)} - y_{(1)}) + y_{(1)}) & \text{if } t \in [0,1), \\ 0 & \text{otherwise.} \end{cases}$$

Now, we observe that

$$\int_0^1 (\widehat{G}_i(t) - G_i^*(t))^2 dt$$

$$= \int_0^1 (\widehat{F}_i^+((1-t)(y_{(n)} - y_{(1)}) + y_{(1)}) - F_i^*((1-t)(y_{(n)} - y_{(1)}) + y_{(1)}))^2 dt$$

$$= \frac{1}{y_{(n)} - y_{(1)}} \int_{y_{(1)}}^{y_{(n)}} (\widehat{F}_i^+(s) - F_i^*(s))^2 ds \tag{34}$$

by making the change of variable $s = (1-t)(y_{(n)} - y_{(1)}) + y_{(1)}$.

Furthermore, by Lemma 6,

$$\int_0^1 |D(G_i^*)(t)|^2 dt = \int_0^\infty |D(G_i^*)(t)|^2 dt$$

$$= \int_0^\infty |G_i^*(t)|^2 dt = \int_0^1 |G_i^*(t)|^2 dt = \frac{1}{y_{(n)} - y_{(1)}} \int_{y_{(1)}}^{y_{(n)}} |F_i^*(s)|^2 ds, \tag{35}$$

and

$$\int_0^1 |D(\widehat{G}_i)(t)|^2 dt = \int_0^\infty |D(\widehat{G}_i)(t)|^2 dt = \int_0^\infty |\widehat{G}_i(t)|^2 dt$$

$$= \int_0^1 |\widehat{G}_i(t)|^2 dt = \frac{1}{y_{(n)} - y_{(1)}} \int_{y_{(1)}}^{y_{(n)}} |\widehat{F}_i^+(s)|^2 ds, \tag{36}$$

Also, by Lemma 6,

$$-\int_0^1 D(\widehat{G}_i)(t) \cdot D(G_i^*)(t) dt = -\int_0^\infty D(\widehat{G}_i)(t) \cdot D(G_i^*)(t) dt$$

$$\leq -\int_0^\infty \widehat{G}_i(t) \cdot G_i^*(t) dt = -\int_0^1 \widehat{G}_i(t) \cdot G_i^*(t) dt \tag{37}$$

which implies

$$-\int_0^1 D(\widehat{G}_i)(t) \cdot D(G_i^*)(t) dt \leq \frac{1}{y_{(n)} - y_{(1)}} \int_{y_{(1)}}^{y_{(n)}} \widehat{F}_i^+(t) \cdot F_i^*(t) dt. \tag{38}$$

Combining (35), (36) and (37), we obtain that

$$\int_0^1 (D(\widehat{G}_i)(t) - D(G_i^*)(t))^2 dt \leq \frac{1}{y_{(n)} - y_{(1)}} \int_{y_{(1)}}^{y_{(n)}} (\widehat{F}_i^+(t) - F_i^*(t))^2 dt. \tag{39}$$

However, since $G_i^*$ is decreasing and continuous in $[0,1)$, then by Lemma 6, it holds that $D(G_i^*)(t) = G_i^*(t)$ for all $t \geq 0$. Thus, from (39),

$$\int_0^1 (D(\widehat{G}_i)(t) - G_i^*(t))^2 dt \leq \frac{1}{y_{(n)} - y_{(1)}} \int_{y_{(1)}}^{y_{(n)}} (\widehat{F}_i^+(t) - F_i^*(t))^2 dt. \tag{40}$$

Now, by Lemma 5 and (33), we obtain that

$$D(\widehat{G}_i)(t) := \sum_{l=1}^{n-1} a_{i,j_l} 1_{[m_{l-1}, m_l)}(t) \tag{41}$$

where

$$m_l := \sum_{k=1}^{l} \frac{y_{(j_k+1)} - y_{(j_k)}}{y_{(n)} - y_{(1)}}, \quad l = 1, \ldots, n-1,$$

and with $m_0 = 0$.

With (41) in hand, we let $H_i(t) = D(\widehat{G}_i)(1 - (t - y_{(1)})/(y_{(n)} - y_{(1)}))$ for all $t \in (y_{(1)}, y_{(n)}]$. Thus, can write

$$H_i(t) = \sum_{l=1}^{n-1} a_{i,j_l} 1_{[v_l, v_{l-1})}(t)$$

for all $t \in (y_{(1)}, y_{(n)})$, where $v_0$ and $v_l = y_{(n)} - \sum_{k=1}^{l}(y_{(j_k+1)} - y_{(j_k)})$ for all $l = 1, \ldots, n-1$. Also, $D(\widehat{G}_i)(t) := H((1-t)(y_{(n)} - y_{(1)}) + y_{(1)})$. Hence, from (40) we obtain that

$$\frac{1}{y_{(n)} - y_{(1)}} \int_{y_{(1)}}^{y_{(n)}} (H_i(t) - F_i^*(t))^2 dt \;=\; \int_0^1 (D(\widehat{G}_i)(t) - G_i^*(t))^2 dt$$

$$\leq \;\frac{1}{y_{(n)} - y_{(1)}} \int_{y_{(1)}}^{y_{(n)}} (\widehat{F}_i^+(t) - F_i^*(t))^2 dt$$

and as a result,

$$\int_{y_{(1)}}^{y_{(n)}} (\widetilde{F}_i(t) - F_i^*(t))^2 dt = \int_{y_{(1)}}^{y_{(n)}} (H_i(t) - F_i^*(t))^2 dt \leq \int_{y_{(1)}}^{y_{(n)}} (\widehat{F}_i^+(t) - F_i^*(t))^2 dt, \quad (42)$$

since $H_i(t) = \widetilde{F}_i(t)$ for all $t \in (y_{(1)}, y_{(n)})$.

Combining (32) and (42), we obtain,

$$\int_{\mathbb{R}} (\widetilde{F}_i(t) - F_i^*(t))^2 dt \;\leq\; \int_{\mathbb{R}} (\widehat{F}_i^+(t) - F_i^*(t))^2 dt \;\leq\; \int_{\mathbb{R}} (\widehat{F}_i(t) - F_i^*(t))^2 dt.$$

The claim then follows. $\qquad\square$

### G.4 Proof of Theorem 2

*Proof.* First, by the basic inequality we have that

$$\frac{1}{2}\sum_{i=1}^{n}(F_i(t) - F_i^*(t))^2 \;\leq\; \sum_{i=1}^{n}(F_i^*(t) - 1_{\{y_i \leq t\}})(F_i^*(t) - F_i(t))$$

for all $F \in \{\kappa F^* + (1-\kappa)\widehat{F} : \kappa \in [0,1]\}$. Hence, for $\xi_1, \ldots, \xi_n$ are independent Rademacher random variables, we have that

$$\mathbb{P}\left(\sup_{t \in \mathbb{R}} \sum_{i=1}^{n}\left(\widehat{F}_i(t) - F_i^*(t)\right)^2 > 2\eta^2\right) \;\leq\; \mathbb{P}\left(\sup_{t \in \mathbb{R}} \sup_{\theta \in K_t : \|\theta - F^*(t)\| \leq \eta} \sum_{i=1}^{n}(F_i^*(t) - 1_{\{y_i \leq t\}})(\theta_i - F_i^*(t)) > \eta^2\right)$$

$$\leq\; \mathbb{P}\left(\sup_{t \in \mathbb{R}} \sup_{\theta \in K : \|\theta - F^*(t)\| \leq \eta} \sum_{i=1}^{n}(F_i^*(t) - 1_{\{y_i \leq t\}})(\theta_i - F_i^*(t)) > \eta^2\right)$$

$$\leq\; \frac{1}{\eta^2}\mathbb{E}\left(\sup_{t \in \mathbb{R}} \sup_{\theta \in K : \|\theta - F^*(t)\| \leq \eta} \sum_{i=1}^{n}(F_i^*(t) - 1_{\{y_i \leq t\}})(\theta_i - F_i^*(t))\right)$$

$$\leq\; \frac{1}{\eta^2}\mathbb{E}\left(\sup_{t \in \mathbb{R}} \sup_{\theta \in K - K : \|\theta\| \leq \eta} \sum_{i=1}^{n}(F_i^*(t) - 1_{\{y_i \leq t\}})\theta_i\right)$$

$$\leq\; \frac{2}{\eta^2}\mathbb{E}\left(\sup_{t \in \mathbb{R}} \sup_{\theta \in K - K : \|\theta\| \leq \eta} \sum_{i=1}^{n}\xi_i 1_{\{y_i \leq t\}}\theta_i\right)$$

$$\leq\; \frac{2}{\eta^2}\mathbb{E}\left(\mathbb{E}\left(\sup_{t \in \mathbb{R}} \sup_{\theta \in K - K : \|\theta\| \leq \eta} \sum_{i=1}^{n}\xi_i 1_{\{y_i \leq t\}}\theta_i \Big| y\right)\right)$$

$$=\; \frac{2}{\eta^2}\mathbb{E}\left(\mathbb{E}\left(\max_{t \in \{y_1, \ldots, y_n\}} \sup_{\theta \in K - K : \|\theta\| \leq \eta} \sum_{i=1}^{n}\xi_i 1_{\{y_i \leq t\}}\theta_i \Big| y\right)\right)$$

where the second inequality follows from Markov's inequality, the fourth by simmetrization. Next, notice that for a fixed $t$ and $y$, the random variables $\{\xi_i 1_{\{y_i \le t\}}\}_{i=1}^n$ are subGaussian(1). Hence, by Lemma 3,

$$
\mathbb{E}\left( \max_{t \in \{y_1, \ldots, y_n\}} \sup_{\theta \in K - K \,:\, \|\theta\| \le \eta} \sum_{i=1}^n \xi_i 1_{\{y_i \le t\}} \theta_i \bigg| y \right) \;\le\; C \int_0^{\eta/4} \sqrt{\log N(\varepsilon, (K - K) \cap B_\eta(0), \|\cdot\|)} d\varepsilon \\
+ C\eta \sqrt{\log n},
$$

for some constant $C > 0$. The claim then follows.

$\square$

## G.5   Proof of Corollary 2

*Proof.* Following the proof of Theorem 2.2 in Chatterjee [13], we obtain that for any positive integer $l$ it holds, for $g \sim N(0, I_n)$, that

$$
\mathbb{E}\left[ \sup_{\theta \in K_t \,:\, \|\theta - F^*(t)\| \le \eta} g^\top (\theta - F^*(t)) \right] \;=\; \mathbb{E}\left[ \sup_{\theta \in K \,:\, \|\theta - F^*(t)\| \le \eta} g^\top (\theta - F^*(t)) \right] \\
\le\; C_1 \left[ 2\sqrt{2^l \eta} n^{1/4} + \frac{\eta^2}{2^{l-1}} \right] \tag{43}
$$

for a positive constant $C_1$. Next, let $L$ the constant in (6). We now choose $l$ large enough such that

$$
\frac{C_1 \eta^2}{2^{l-1}} \;\le\; \frac{\eta^2}{2L}.
$$

Furthermore, for this choice of $l$, we can choose $\eta$ as $\eta \asymp n^{1/6}$ such that

$$
C_1 2\sqrt{2^l \eta} n^{1/4} \;\le\; \frac{\eta^2}{2L}.
$$

Thus, for a choice of $\eta$ satisfying $\eta \asymp n^{1/6}$, we obtain

$$
\sup_{t \in \mathbb{R}} \mathbb{E}\left[ \sup_{\theta \in K_t \,:\, \|\theta - F^*(t)\| \le \eta} g^\top (\theta - F^*(t)) \right] \;\le\; \frac{\eta^2}{L},
$$

and so (9) follows from Theorem 1. Furthermore, the corresponding conclusion for $\{\widetilde{F}(t)\}_{t \in \mathbb{R}}$ follows from Corollary 1.

Finally, we notice that for some positive constant $C_2$

$$
\begin{aligned}
\int_0^{\eta/4} \sqrt{\log N(\varepsilon, (K \cap [a,b] - K \cap [a,b]) \cap B_\eta(0), \|\cdot\|)} d\varepsilon \;&\le\; 2 \int_0^{\eta/4} \sqrt{\log N(\varepsilon/2, (K \cap [a,b]) \cap B_\eta(0), \|\cdot\|)} d\varepsilon \\
&\le\; 2 \int_0^{\eta/4} \sqrt{\log N(\varepsilon/2, (K \cap [a,b]) \cap B_\eta(0), \|\cdot\|)} d\varepsilon \\
&\le\; 2 \int_0^{\eta/4} \sqrt{\frac{2C_2 \sqrt{n}(b-a)}{\varepsilon}} d\varepsilon \\
&\le\; 2\sqrt{2C_2(b-a)} n^{1/4} \eta^{1/2}
\end{aligned}
$$

where the third inequality follows from Lemma 4.20 in Chatterjee [13]. Therefore, the claim in (10) follows form Theorem 2 by taking $\eta$ satisfying $\eta \asymp (b-a)^{1/3} n^{1/6} + \sqrt{\log n}$.   $\square$

## G.6 Proof of Theorem 4

*Proof.* First we observe that by the basic inequality, for all $t \in \mathbb{R}$,

$$\frac{\|F(t) - F^*(t)\|^2}{2} \leq a(t)^\top (F(t) - F^*(t)) + \lambda_t [\mathrm{pen}_t(F^*(t)) - \mathrm{pen}_t(F(t))] \qquad (44)$$

where $a(t) = w(t) - F^*(t)$ for all $t \in \mathbb{R}$ and $i \in \{1, \ldots, n\}$, and where the inequality holds for all

$$F(t) \in \Lambda(t) := \left\{ s\widehat{F}(t) + (1 - s)F^*(t) \ : \ s \in [0, 1] \right\} \subset \mathbb{R}^n.$$

Therefore,

$$\begin{aligned}
\mathrm{pen}_t(F(t)) &\leq \mathrm{pen}_t(F(t)) + \frac{\|F(t) - F^*(t)\|^2}{2\lambda_t} \\
&\leq \frac{a(t)^\top (F(t) - F^*(t))}{\lambda_t} + \mathrm{pen}_t(F^*(t))
\end{aligned}$$

for all $F(t) \in \Lambda(t)$ and $t \in \mathbb{R}$. Hence, by the properties of $\mathrm{pen}_t(\cdot)$, we have that

$$\begin{aligned}
\mathrm{pen}_t(F(t) - F^*(t)) &\leq \mathrm{pen}_t(F(t)) + \mathrm{pen}_t(F^*(t)) \\
&\leq \frac{a(t)^\top (F(t) - F^*(t))}{\lambda_t} + 2\mathrm{pen}_t(F^*(t))
\end{aligned} \qquad (45)$$

for all $F(t) \in \Lambda(t)$ and $t \in \mathbb{R}$.

Now suppose that there exists $F(t) \in \Lambda(t)$ such that

$$\|F(t) - F^*(t)\| \leq \eta^2$$

and $\mathrm{pen}_t(F(t)) \geq 5\mathrm{pen}_t(F^*(t))$. Then

$$\begin{aligned}
\mathrm{pen}_t(F(t) - F^*(t)) &\geq \mathrm{pen}_t(F(t)) - \mathrm{pen}_t(F^*(t)) \\
&\geq 4\mathrm{pen}_t(F^*(t)).
\end{aligned}$$

Hence, we let

$$s := \frac{4\,\mathrm{pen}_T(F^*(t))}{\mathrm{pen}_t(F(t) - F^*(t))} \in [0, 1].$$

Then we set

$$\widetilde{F}(t) := sF(t) + (1 - s)F^*(t) \in \Lambda(t).$$

As a result,

$$\|F^*(t) - \widetilde{F}(t)\|^2 \leq \|F^*(t) - F(t)\|^2 \leq \eta^2.$$

Also,

$$\begin{aligned}
\mathrm{pen}_t(\widetilde{F}(t) - F^*(t)) &= \mathrm{pen}_t(sF(t) + (1 - s)F^*(t) - F^*(t)) \\
&= \mathrm{pen}_t(sF(t) - sF^*(t)) \\
&= s\mathrm{pen}_t(F(t) - F^*(t)) \\
&= 4\mathrm{pen}_t(F^*(t)).
\end{aligned}$$

Therefore,

$$\begin{aligned}
4\mathrm{pen}_t(F^*(t)) &= \mathrm{pen}_t(\widetilde{F}(t) - F^*(t)) \\
&\leq \frac{a(t)^\top (\widetilde{F}(t) - F^*(t))}{\lambda_t} + 2\mathrm{pen}_t(F^*(t)),
\end{aligned}$$

where the inequality follows from (45). This implies that

$$2\mathrm{pen}(F^*(t)) \leq \frac{a(t)^\top (\widetilde{F}(t) - F^*(t))}{\lambda_t}. \qquad (46)$$

Hence, we let

$$\lambda_t = \frac{\eta^2}{4\mathrm{pen}(F^*(t))}.$$

Then from (46) we obtain that

$$\frac{\eta^2}{2} \leq a(t)^\top (\widetilde{F}(t) - F^*(t)).$$

It follows that the events

$$\Omega_1 := \bigcup_{t\in\mathbb{R}}\left\{\sup_{F(t)\in\Lambda(t)\,:\,\|F(t)-F^*(t)\|\leq\eta}\mathrm{pen}_t(F(t)) \geq 5\mathrm{pen}_t(F^*(t))\right\}$$

and

$$\Omega_2 := \left\{\sup_{t\in\mathbb{R}}\sup_{F(t)\in\Lambda(t)\,:\,\|F(t)-F^*(t)\|\leq\eta,\,\mathrm{pen}(F^*(t)-F(t))\leq4\mathrm{pen}(F^*(t))}a(t)^\top(F(t)-F^*(t)) \geq \frac{\eta^2}{2}\right\}$$

satisfy that $\Omega_1 \subset \Omega_2$. Hence, $\mathbb{P}(\Omega_1) \leq \mathbb{P}(\Omega_2)$.

Next, we observe that if $\|\hat{F}(t)-F(t)\| \geq \eta$, then there exists $F(t)\in\Lambda(t)$ such that $\|F(t)-F^*(t)\| = \eta$. This implies, by (44), that

$$\frac{\eta^2}{2} \leq a(t)^\top(F(t)-F^*(t)) + \lambda_t\left[\mathrm{pen}(F^*(t)) - \mathrm{pen}(F(t))\right]$$

and so from our choice of $\lambda_t$

$$\frac{\eta^2}{4} \leq a(t)^\top(F(t)-F^*(t)). \tag{47}$$

Thus, (47) holds for some $F(t)\in\Lambda(t)$ with $\|F(t)-F^*(t)\|\leq\eta$ provided that $\|\hat{F}(t)-F^*(t)\| > \eta$. Therefore,

$$
\begin{aligned}
\mathbb{P}\left(\sup_{t\in\mathbb{R}}\|\hat{F}(t)-F^*(t)\| > \eta\right) &\leq \mathbb{P}\left(\left\{\sup_{t\in\mathbb{R}}\|\hat{F}(t)-F^*(t)\| > \eta\right\}\cap\Omega_1^c\right) + \mathbb{P}(\Omega_1) \\
&\leq \mathbb{P}\left(\sup_{t\in\mathbb{R}}\sup_{F(t)\in\Lambda(t)\,:\,\|F^*(t)-F(t)\|\leq\eta,\,\mathrm{pen}(F(t))\leq5\mathrm{pen}(F^*(t))}a(t)^\top(F(t)-F^*(t))\right.\\
&\qquad\left. \geq \frac{\eta^2}{4}\right) + \mathbb{P}(\Omega_2) \\
&\leq 2\,\mathbb{P}\left(\sup_{t\in\mathbb{R}}\sup_{F(t)\in\Lambda(t)\,:\,\|F(t)-F^*(t)\|\leq\eta,\,\mathrm{pen}(F(t))\leq5\mathrm{pen}(F^*(t))}a(t)^\top(F(t)-F^*(t))\right.\\
&\qquad\left. \geq \frac{\eta^2}{4}\right) \\
&\leq \frac{8}{\eta^2}\mathbb{E}\left(\sup_{t\in\mathbb{R}}\sup_{F(t)\,:\,\|F^*(t)-F(t)\|\leq\eta,\,\mathrm{pen}(F(t))\leq5\mathrm{pen}(F^*(t))}a(t)^\top(F(t)-F^*(t))\right) \\
&\leq \frac{8}{\eta^2}\mathbb{E}\left(\sup_{t\in\mathbb{R}}\sup_{\theta\in K\,:\,\|\theta\|\leq\eta}a(t)^\top\theta\right).
\end{aligned}
\tag{48}
$$

Hence, for $\xi_1,\ldots,\xi_n$ independent Rademacher variables independent of $y$, we have that

$$
\begin{aligned}
\mathbb{P}\left(\sup_{t\in\mathbb{R}}\|\hat{F}(t)-F^*(t)\| > \eta\right) &\leq \frac{16}{\eta^2}\mathbb{E}\left(\sup_{t\in\mathbb{R}}\sup_{\theta\in K\,:\,\|\theta\|\leq\eta}\sum_{i=1}^n\xi_i\mathbf{1}_{\{y_i\leq t\}}\theta_i\right) \\
&= \frac{16}{\eta^2}\mathbb{E}\left(\mathbb{E}\left(\sup_{t\in\mathbb{R}}\sup_{\theta\in K\,:\,\|\theta\|\leq\eta}\sum_{i=1}^n\xi_i\mathbf{1}_{\{y_i\leq t\}}\theta_i\,\middle|\,y\right)\right) \\
&= \frac{16}{\eta^2}\mathbb{E}\left(\mathbb{E}\left(\max_{t\in\{y_1,\ldots,y_n\}}\sup_{\theta\in K\,:\,\|\theta\|\leq\eta}\sum_{i=1}^n\xi_i\mathbf{1}_{\{y_i\leq t\}}\theta_i\,\middle|\,y\right)\right)
\end{aligned}
$$

where the first inequality follows from (48) and symmetrization. However, by Lemma 3,

$$\mathbb{E}\left(\max_{t\in\{y_1,\ldots,y_n\}}\sup_{\theta\in K\,:\,\|\theta\|\leq\eta}\sum_{i=1}^n\xi_i\mathbf{1}_{\{y_i\leq t\}}\theta_i\,\middle|\,y\right) \leq C\int_0^{\eta/4}\sqrt{\log N(\varepsilon,K\cap B_\eta(0),\|\cdot\|)}d\varepsilon + C\eta\sqrt{\log n},$$

for some constant $C > 0$. The claim then follows.

$\square$

### G.7 Proof of Corollary 3

*Proof.* We start by noting that the set $K := \left\{\theta \in \mathbb{R}^n : \mathrm{TV}^{(r)}(\theta) \leq V\right\}$ is convex. Indeed, let $\theta^{(1)}, \theta^{(2)} \in K$, and define $\theta^{(\alpha)} := \alpha\theta^{(1)} + (1-\alpha)\theta^{(2)}$ for $\alpha \in [0,1]$. Then $\mathrm{TV}^{(r)}(\theta^{(\alpha)}) = n^{r-1}\left\|D^{(r)}\theta^{(\alpha)}\right\|_1 = n^{r-1}\left\|\alpha D^{(r)}\theta^{(1)} + (1-\alpha)D^{(r)}\theta^{(2)}\right\|_1$. By convexity of the $\ell_1$-norm, $\mathrm{TV}^{(r)}(\theta^{(\alpha)}) \leq \alpha\mathrm{TV}^{(r)}(\theta^{(1)}) + (1-\alpha)\mathrm{TV}^{(r)}(\theta^{(2)}) \leq V_t$. Hence, $\theta^{(\alpha)} \in K$, proving that $K$ is convex.

Let $g \sim N(0, I_n)$ and set

$$K := \left\{\theta \in \mathbb{R}^n : \mathrm{TV}^{(r)}(\theta) \leq V\right\}. \tag{49}$$

Hence,

$$
\begin{aligned}
\mathbb{E}\left[\sup_{\theta \in K_t \,:\, \|\theta - F^*(t)\| \leq \eta} g^\top(\theta - F^*(t))\right] &\leq \mathbb{E}\left[\sup_{\theta \in K \,:\, \|\theta - F^*(t)\| \leq \eta} g^\top(\theta - F^*(t))\right] \\
&\leq C_r\left[\eta\left(\frac{\sqrt{n}V}{\eta}\right)^{1/2r} + \eta\sqrt{\log n}\right]
\end{aligned}
\tag{50}
$$

where $C_r > 0$ is a constant the second inequality follows from Lemma B.1 in Guntuboyina et al. [23].

Next, notice that

$$C_r\eta\left(\frac{\sqrt{n}V}{\eta}\right)^{1/2r} \leq \frac{\eta^2}{2L}$$

holds if

$$(2LC_r)^{2r/(2r+1)}n^{1/(4r+2)}V^{1/(2r+1)} \leq \eta.$$

Also,

$$C_r\eta\sqrt{\log n} \leq \frac{\eta^2}{2L}$$

provided that $2LC_r\sqrt{\log n} \leq \eta$. Hence, taking

$$\eta = \max\{(2LC_r)^{2r/(2r+1)}n^{1/(4r+2)}V^{1/(2r+1)}, 2LC_r\sqrt{\log n}\}$$

we obtain (13).

The claim for $\widetilde{F}(t)$ follows from Corollary 1.

To show (14), we observe that by the proof of Theorem B.1 in Guntuboyina et al. [23], for $0 < \eta < n$, we have

$$
\begin{aligned}
\int_0^{\eta/4} \sqrt{\log N(\varepsilon, (K-K) \cap B_\eta(0), \|\cdot\|)}d\varepsilon &\leq 2\int_0^{\eta/4} \sqrt{\log N(\varepsilon/2, K \cap B_\eta(0), \|\cdot\|)}d\varepsilon \\
&\leq \tilde{C}_r\left[\eta\left(\frac{\sqrt{n}V}{\eta}\right)^{1/2r} + \eta\sqrt{\log n}\right]
\end{aligned}
$$

for some positive constant $\tilde{C}_r$. Hence, (14) follows with the same argument as above.

$\square$

### G.8 Proof of Theorem 3

*Proof.* Notice that for any $t \in \mathbb{R}$ we have that

$$\|F(t) - H(t)\|^2 \leq 2nB$$

for all $F(t), H(t) \in K_t$. This implies that $\sup_{t \in \mathbb{R}} \|\widehat{F}(t) - G(t)\|^2 \leq 2nB$. To control the probability of large deviations, we restrict attention to the event

$$\left\{\sup_{t \in \mathbb{R}} \|\widehat{F}(t) - G(t)\|^2 > \eta^2\right\} \cap \left\{\sup_{t \in \mathbb{R}} \|\widehat{F}(t) - G(t)\|^2 \leq 2nB\right\},$$

which confines the deviations to the range $(\eta^2, 2nB]$. We then partition this interval into dyadic subintervals $(2^{j-1}\eta^2, 2^j\eta^2]$ for $j = 1, \ldots, J$, where $J$ is the smallest integer such that $2^J\eta^2 \geq 2nB$. Applying a union bound over these dyadic shells yields

$$
\begin{aligned}
\mathbb{P}\left(\sup_{t\in\mathbb{R}}\|\widehat{F}(t) - G(t)\| > \eta\right) &= \mathbb{P}\left(\sup_{t\in\mathbb{R}}\|\widehat{F}(t) - G(t)\|^2 > \eta^2, \sup_{t\in\mathbb{R}}\|\widehat{F}(t) - G(t)\|^2 \leq 2nB\right) \\
&\leq \sum_{j=1}^{J}\mathbb{P}\left(\sup_{t\in\mathbb{R}}\|\widehat{F}(t) - G(t)\|^2 > 2^{j-1}\eta^2, \sup_{t\in\mathbb{R}}\|\widehat{F}(t) - G(t)\|^2 \leq 2^j\eta^2\right) \\
&\leq \sum_{j=1}^{J}\mathbb{P}\bigg(\sup_{t\in\mathbb{R}}2\sum_{i=1}^{n}(\widetilde{F}_i(t) - 1_{\{y_i\leq t\}})(G_i(t) - \widehat{F}_i(t)) > 2^{j-1}\eta^2, \\
&\qquad\qquad\qquad \sup_{t\in\mathbb{R}}\|\widehat{F}(t) - G(t)\|^2 \leq 2^j\eta^2\bigg) \\
&\leq \sum_{j=1}^{J}\mathbb{P}\bigg(\sup_{t\in\mathbb{R}}\sup_{F(t)\in K_t\,:\,\|F(t)-G(t)\|^2\leq 2^j\eta^2}\sum_{i=1}^{n}(G_i(t) - 1_{\{y_i\leq t\}})(G_i(t) - F_i(t)) \\
&\qquad\qquad\qquad > 2^{j-2}\eta^2\bigg),
\end{aligned}
$$

where the first inequality follows from the union bound and the second from the basic inequality. However,

$$
\begin{aligned}
&\mathbb{P}\left(\sup_{t\in\mathbb{R}}\sup_{F(t)\in K_t\,:\,\|F(t)-G(t)\|^2\leq 2^j\eta^2}\sum_{i=1}^{n}(G_i(t) - 1_{\{y_i\leq t\}})(G_i(t) - F_i(t)) > 2^{j-2}\eta^2\right) \\
&\leq \frac{1}{2^{j-2}\eta^2}\mathbb{E}\left(\sup_{t\in\mathbb{R}}\sup_{F(t)\in K_t\,:\,\|F(t)-G(t)\|^2\leq 2^j\eta^2}\sum_{i=1}^{n}(G_i(t) - 1_{\{y_i\leq t\}})(G_i(t) - F_i(t))\right) \\
&\leq \frac{1}{2^{j-2}\eta^2}\mathbb{E}\left(\sup_{t\in\mathbb{R}}\sup_{F(t)\in K_t\,:\,\|F(t)-G(t)\|^2\leq 2^j\eta^2}\sum_{i=1}^{n}(F_i^*(t) - 1_{\{y_i\leq t\}})(G_i(t) - F_i(t))\right) \\
&\quad + \frac{1}{2^{j-2}\eta^2}\sup_{t\in\mathbb{R}}\sup_{F(t)\in K_t\,:\,\|F(t)-G(t)\|^2\leq 2^j\eta^2}\sum_{i=1}^{n}(G_i(t) - F_i^*(t))(G_i(t) - F_i(t)) \\
&\leq \frac{1}{2^{j-2}\eta^2}\mathbb{E}\left(\sup_{t\in\mathbb{R}}\sup_{F(t)\in K_t\,:\,\|F(t)-G(t)\|^2\leq 2^j\eta^2}\sum_{i=1}^{n}(F_i^*(t) - 1_{\{y_i\leq t\}})(G_i(t) - F_i(t))\right) \\
&\quad + \frac{4\sqrt{n}}{2^{j/2}\eta}\sup_{t\in\mathbb{R}}\|G(t) - F^*(t)\|_\infty
\end{aligned}
$$

for some constant $C > 0$, where the first inequality follows from Markov's inequality, and the last inequality holds by Cauchy–Schwarz inequality. Furthermore,

$$
\begin{aligned}
&\mathbb{E}\left(\sup_{t\in\mathbb{R}}\sup_{F(t)\in K_t\,:\,\|F(t)-G(t)\|^2\leq 2^j\eta^2}\sum_{i=1}^{n}(F_i^*(t) - 1_{\{y_i\leq t\}})(G_i(t) - F_i(t))\right) \\
&\leq \mathbb{E}\left(\sup_{t\in\mathbb{R}}\sup_{F(t)\in K_t - K_t\,:\,\|F(t)\|^2\leq 2^j\eta^2}\sum_{i=1}^{n}(F_i^*(t) - 1_{\{y_i\leq t\}})F_i(t)\right) \\
&\leq C\int_0^{2^{j/2}\eta/4}\sqrt{\log N(\varepsilon, (K - K)\cap B_\eta(0), \|\cdot\|)}\,d\varepsilon + C2^{j/2}\eta\sqrt{\log n},
\end{aligned}
$$

where the last inequality follows as in the proof of Theorem 2. Therefore,

$$
\begin{aligned}
\mathbb{P}\left(\sup_{t\in\mathbb{R}}\|\widehat{F}(t)-G(t)\| > \eta\right) \;\le\;& \sum_{j=1}^{J}\frac{1}{2^{j-2}\eta^2}\left[C\int_{0}^{2^{j/2}\eta/4}\sqrt{\log N(\varepsilon,(K-K)\cap B_\eta(0),\|\cdot\|)}d\varepsilon\right. \\
& \left.+C2^{j/2}\eta\sqrt{\log n}\right] + \sum_{j=1}^{J}\frac{4\sqrt{n}}{2^{j/2}\eta}\sup_{t\in\mathbb{R}}\|G(t)-F^*(t)\|_\infty \\
\;\le\;& \frac{C}{\eta^2}\sum_{j=1}^{J}\frac{1}{2^{j-2}}\int_{0}^{2^{j/2}\eta/4}\sqrt{\log N(\varepsilon,(K-K)\cap B_\eta(0),\|\cdot\|)}d\varepsilon + \\
& \frac{4C\sqrt{\log n}}{\eta}\sum_{j=1}^{J}\left(\frac{1}{2^{1/2}}\right)^j + \frac{4\sqrt{n}}{\eta}\sup_{t\in\mathbb{R}}\|G(t)-F^*(t)\|_\infty\sum_{j=1}^{J}\left(\frac{1}{2^{1/2}}\right)^j \\
\;\le\;& \frac{C}{\eta^2}\sum_{j=1}^{J}\frac{1}{2^{j-2}}\int_{0}^{2^{j/2}\eta/4}\sqrt{\log N(\varepsilon,(K-K)\cap B_\eta(0),\|\cdot\|)}d\varepsilon + \\
& \frac{4C\sqrt{\log n}}{\eta}\frac{2^{-1/2}}{1-2^{-1/2}} + \frac{4\sqrt{n}}{\eta}\sup_{t\in\mathbb{R}}\|G(t)-F^*(t)\|_\infty\frac{2^{-1/2}}{1-2^{-1/2}}
\end{aligned}
$$

and the claim follows noticing that

$$
\mathbb{P}\left(\sup_{t\in\mathbb{R}}\|\widehat{F}(t)-F^*(t)\| > \eta+\sqrt{n}\|F^*(t)-G(t)\|_\infty\right) \;\le\; \mathbb{P}\left(\sup_{t\in\mathbb{R}}\|\widehat{F}(t)-G(t)\| > \eta\right).
$$

$\square$

## G.9 Proof of Corollary 4

We begin by restating the corollary to be proved.

**Corollary 8.** *Let $\widehat{F}(t)$ be the estimator from (3) with the set $K_t$ as in (16) for all $t \in \mathbb{R}$ with $F^*(t)$ not necessarily in $K_t$. Suppose that Assumption 1, described in Appendix C, holds. Let*

$$\phi_n = \max_{(p,M) \in \mathcal{P}} n^{\frac{-2p}{(2p+M)}}.$$

*There exists positive constants $c_1$ and $c_2$ such that if*

$$L = \lceil c_1 \log n \rceil \quad \text{and} \quad \nu = \left\lceil c_2 \max_{(p,M) \in \mathcal{P}} n^{\frac{M}{2(2p+M)}} \right\rceil \tag{51}$$

*or*

$$L = \left\lceil c_1 \max_{(p,M) \in \mathcal{P}} n^{\frac{M}{2(2p+M)}} \log n \right\rceil \quad \text{and} \quad \nu = \lceil c_2 \rceil, \tag{52}$$

*then*

$$\sup_{t \in \mathbb{R}} \sum_{i=1}^{n} \frac{1}{n} \left( \widehat{F}_i(t) - F_i^*(t) \right)^2 = O_{\mathbb{P}}\left( \frac{\log n}{n} + \phi_n \log^4 n \right). \tag{53}$$

Then the proof of such result is provided.

*Proof.* Throughout the proof, we condition on the covariates $x_i$'s and omit this dependence from the notation for simplicity. We proceed by using Theorem 3. First, for a vector $v \in \mathbb{R}^n$, we let $\|v\|_n := \|v\|/\sqrt{n}$. Then, by Theorem 3 in Kohler and Langer [37], it holds that

$$\sup_{t \in \mathbb{R}} \|F^*(t) - G(t)\|_\infty \leq C_1 \sqrt{\phi_n}. \tag{54}$$

for some positive constant $C_1$.

Furthermore, as in the proof of Theorem 2 in Zhang et al. [67], see also Lemma 19 in Kohler and Langer [37], we have that

$$\log N(\varepsilon, \mathcal{F}(L,\nu), \|\cdot\|_n) \lesssim L^2 \nu^2 \log(L\nu) \log(\varepsilon^{-1})$$

for $\varepsilon \in (0,1)$. Therefore, for $\epsilon \in (0, 2\sqrt{n})$, we have that

$$\log N(\varepsilon, \mathcal{F}(L,\nu), \|\cdot\|) \lesssim L^2 \nu^2 \log(L\nu) \log(2\varepsilon^{-1}\sqrt{n}).$$

Therefore, for $\eta < \sqrt{n}$, with

$$J = \left\lceil \frac{\log(2n/\eta^2)}{\log 2} \right\rceil,$$

it holds that

$$\frac{C}{\eta^2} \sum_{j=1}^{J} \frac{1}{2^{j-2}} \int_0^{2^{j/2}\eta/4} \sqrt{\log N(\varepsilon, K(\eta), \|\cdot\|)} d\varepsilon + \frac{C\sqrt{\log n}}{\eta} + \frac{C\sqrt{n}}{\eta} \sup_{t \in \mathbb{R}} \|G(t) - F^*(t)\|_\infty$$

$$\leq \frac{C}{\eta^2} \sum_{j=1}^{J} \frac{1}{2^{j-2}} \int_0^{2^{j/2}\eta/4} \sqrt{\log N(\varepsilon, K-K, \|\cdot\|)} d\varepsilon + \frac{C\sqrt{\log n}}{\eta} + \frac{C\sqrt{n}}{\eta} \sup_{t \in \mathbb{R}} \|G(t) - F^*(t)\|_\infty$$

$$\leq \frac{C}{\eta^2} \sum_{j=1}^{J} \frac{1}{2^{j-2}} \int_0^{2^{j/2}\eta/4} \sqrt{\log N(\varepsilon, K-K, \|\cdot\|)} d\varepsilon + \frac{CC_1\sqrt{\log n}}{\eta} + \frac{C\sqrt{n\phi_n}}{\eta}.$$

$$\lesssim \frac{1}{\eta^2} \sum_{j=1}^{J} \frac{1}{2^{j-2}} \int_0^{2^{j/2}\eta/4} \sqrt{\log N(\varepsilon/2, K, \|\cdot\|)} d\varepsilon + \frac{\sqrt{\log n}}{\eta} + \frac{\sqrt{n\phi_n}}{\eta}$$

$$\lesssim \frac{1}{\eta^2} \sum_{j=1}^{J} \frac{1}{2^{j-2}} \int_0^{2^{j/2}\eta/4} \sqrt{L^2 \nu^2 \log(L\nu) \log(4\varepsilon^{-1}\sqrt{n})} d\varepsilon + \frac{\sqrt{\log n}}{\eta} + \frac{\sqrt{n\phi_n}}{\eta}$$

$$\leq \frac{L\nu\sqrt{\log(L\nu)}}{\eta^2} \sum_{j=1}^{J} \frac{1}{2^{j-2}} \int_0^{2^{j/2}\eta/4} \sqrt{\log(n) + 4\epsilon^{-1}} d\varepsilon + \frac{\sqrt{\log n}}{\eta} + \frac{\sqrt{n\phi_n}}{\eta}$$

Moreover,

$$\frac{L\nu\sqrt{\log(L\nu)}}{\eta^2}\sum_{j=1}^{J}\frac{1}{2^{j-2}}\int_0^{2^{j/2}\eta/4}\sqrt{\log(n)+4\epsilon^{-1}}d\varepsilon \ + \ \frac{\sqrt{\log n}}{\eta} \ + \ \frac{\sqrt{n\phi_n}}{\eta}$$

$$\leq \frac{L\nu\sqrt{\log(L\nu)}}{\eta^2}\sum_{j=1}^{J}\frac{1}{2^{j-2}}\int_0^{2^{j/2}\eta/4}(\sqrt{\log(n)}+2\epsilon^{-1/2})d\varepsilon \ + \ \frac{\sqrt{\log n}}{\eta} \ + \ \frac{\sqrt{n\phi_n}}{\eta}$$

$$\leq \frac{L\nu\sqrt{\log(L\nu)}}{\eta^2}\sum_{j=1}^{J}\frac{4\eta\sqrt{\log n}}{2^{j/2}} + \frac{L\nu\sqrt{\log(L\nu)}}{\eta^2}\sum_{j=1}^{J}\frac{1}{2^{j-2}}\int_0^{2^{j/2}\eta/4}2\epsilon^{-1/2}d\varepsilon \ + \ \frac{\sqrt{\log n}}{\eta} \ + \ \frac{\sqrt{n\phi_n}}{\eta}$$

$$\lesssim \frac{L\nu\sqrt{\log(L\nu)\log n}}{\eta} + \frac{L\nu\sqrt{\log(L\nu)}}{\eta^2}\sum_{j=1}^{J}\frac{1}{2^{j-2}}\int_0^{2^{j/2}\eta/4}2\epsilon^{-1/2}d\varepsilon \ + \ \frac{\sqrt{\log n}}{\eta} \ + \ \frac{\sqrt{n\phi_n}}{\eta}$$

$$\lesssim \frac{L\nu\sqrt{\log(L\nu)\log n}}{\eta} + \frac{L\nu\sqrt{\log(L\nu)}}{\eta^2}\sum_{j=1}^{J}\frac{\sqrt{\eta}}{2^{3j/4-2}} \ + \ \frac{\sqrt{\log n}}{\eta} \ + \ \frac{\sqrt{n\phi_n}}{\eta}.$$

$$\lesssim \frac{L\nu\sqrt{\log(L\nu)\log n}}{\eta} + \frac{L\nu\sqrt{\log(L\nu)}}{\eta^{3/2}} \ + \ \frac{\sqrt{\log n}}{\eta} \ + \ \frac{\sqrt{n\phi_n}}{\eta}.$$

$$(55)$$

Moreover, by our choice of $L$ and $\nu$,

$$L\nu \asymp (\log n)\cdot\sqrt{n\phi_n}. \tag{56}$$

Therefore, from (55) and (56),

$$\frac{C}{\eta^2}\sum_{j=1}^{J}\frac{1}{2^{j-2}}\int_0^{2^{j/2}\eta/4}\sqrt{\log N(\varepsilon,K(\eta),\|\cdot\|)}d\varepsilon \ + \ \frac{C\sqrt{\log n}}{\eta} \ + \ \frac{C\sqrt{n}}{\eta}\sup_{t\in\mathbb{R}}\|G(t)-F^*(t)\|_\infty$$

$$\lesssim \frac{\sqrt{n\phi_n}\log^2 n}{\eta} + \frac{\sqrt{.n\phi_n\log(n)}}{\eta^{3/2}} + \frac{\sqrt{\log n}}{\eta} + \frac{\sqrt{n\phi_n}}{\eta}.$$

$$(57)$$

Hence, the claim follows by taking

$$\eta \asymp \sqrt{\log n} \ + \ \sqrt{n\phi_n}\log^2 n.$$

$\square$

### G.10 Proof of Corollary 5

*Proof.* By Theorem 1, it is enough to bound

$$\sup_{t \in \mathbb{R}} \mathbb{E}\left[\sup_{\theta \in K_t \,:\, \|\theta - F^*(t)\| \leq \eta} g^\top (\theta - F^*(t))\right],$$

where where $g \sim N(0, I_n)$. Let, $t \in \mathbb{R}$ and $g \sim \mathcal{N}(0, I_n)$ be a standard Gaussian vector in $\mathbb{R}^n$, and let $K_t \subset \mathbb{R}^n$ be a constraint set. Now we analyze the quantity

$$\mathbb{E}\left[\sup_{\theta \in K_t : \|\theta - F^*(t)\| \leq \eta} \langle g, \theta - F^*(t)\rangle\right].$$

Since $K_t$ is a convex cone it is closed under positive scalar multiplication; that is, $\frac{K_t}{\eta} = K_t$ for any $\eta > 0$. Using this, we have

$$(K_t - F^*(t)) \cap B(0, \eta) = \eta \cdot \left(\left(K_t - \frac{F^*(t)}{\eta}\right) \cap B(0,1)\right).$$

In consequence,

$$\mathbb{E}\left[\sup_{\theta \in K_t : \|\theta - F^*(t)\| \leq \eta} \langle g, \theta - F^*(t)\rangle\right] = \eta \cdot \mathbb{E}\left[\sup_{\theta \in \left(K_t - \frac{F^*(t)}{\eta}\right) \cap B(0,1)} \langle g, \theta\rangle\right]. \tag{58}$$

Now, we consider the tangent cone to a convex set $K_t$ at a point $F^*(t) \in K_t$. This is defined as,

$$T_{K_t}\left(\frac{F^*(t)}{\eta}\right) := \mathrm{cl}\left\{h\left(\theta - \frac{F^*(t)}{\eta}\right) : \theta \in K_t, h > 0\right\}.$$

Now, observe that $K_t := \{\theta \in \mathbb{R}^n : \theta_1 \leq \cdots \leq \theta_n\}$ is a closed convex cone, and the set $K_t - \frac{F^*(t)}{\eta}$ is contained in the *tangent cone* of $K_t$ at $\frac{F^*(t)}{\eta}$, that is,

$$K_t - \frac{F^*(t)}{\eta} \subset T_{K_t}\left(\frac{F^*(t)}{\eta}\right).$$

Therefore,

$$\mathbb{E}\left[\sup_{\theta \in \left(K_t - \frac{F^*(t)}{\eta}\right) \cap B(0,1)} \langle g, \theta\rangle\right] \leq \mathbb{E}\left[\sup_{\theta \in T_{K_t}\left(\frac{F^*(t)}{\eta}\right) \cap B(0,1)} \langle g, \theta\rangle\right]. \tag{59}$$

Furthermore, $\sup_{\theta \in T_{K_t}\left(\frac{F^*(t)}{\eta}\right) \cap B(0,1)} \langle g, \theta\rangle = \sup_{\theta \in T_{K_t}\left(\frac{F^*(t)}{\eta}\right) \cap S^{n-1}} \langle g, \theta\rangle$, where $S^{n-1}$ denotes the unit Euclidean sphere. Then, by Equation (59)

$$\mathbb{E}\left[\sup_{\theta \in \left(K_t - \frac{F^*(t)}{\eta}\right) \cap B(0,1)} \langle g, \theta\rangle\right] \leq \mathbb{E}\left[\sup_{\theta \in T_{K_t}\left(\frac{F^*(t)}{\eta}\right) \cap S^{n-1}} \langle g, \theta\rangle\right]. \tag{60}$$

Now, we define the Gaussian width of subset $\mathcal{K}$ as

$$w(\mathcal{K}) := \mathbb{E}\left[\sup_{v \in \mathcal{K} \cap S^{n-1}} \langle g, v\rangle\right].$$

Therefore, from Equations (58) and (60)

$$\mathbb{E}\left[\sup_{\theta \in K_t : \|\theta - F^*(t)\| \leq \eta} \langle g, \theta - F^*(t)\rangle\right] \leq \eta \cdot w\left(T_{K_t}\left(\frac{F^*(t)}{\eta}\right)\right). \tag{61}$$

Let $\delta(\mathcal{K})$ denote the statistical dimension of a closed convex cone $\mathcal{K} \subset \mathbb{R}^n$, defined as

$$\delta(\mathcal{K}) := \mathbb{E}[D(g; \mathcal{K})], \quad \text{where } D(y; \mathcal{K}) := \sum_{i=1}^n \frac{\partial}{\partial y_i} \hat{\theta}_i(y; \mathcal{K})$$

and $g \sim \mathcal{N}(0, I_n)$. Here, $\hat{\theta}(y; \mathcal{K})$ denotes the Euclidean projection of $y \in \mathbb{R}^n$ onto $\mathcal{K}$, formally defined as

$$\hat{\theta}(y; \mathcal{K}) := \arg\min_{\theta \in \mathcal{K}} \|\theta - y\|^2.$$

Using Proposition 10.1 in [1], the Gaussian width is controlled by the statistical dimension:

$$w\left(T_{K_t}\left(\frac{F^*(t)}{\eta}\right)\right) \leq \sqrt{\delta\left(T_{K_t}\left(\frac{F^*(t)}{\eta}\right)\right)}.$$

We now analyze the statistical dimension of this tangent cone. Let

$$k(t) := \left|\left\{i \in \{1, \ldots, n-1\} : F_i^*(t) < F_{i+1}^*(t)\right\}\right|$$

be the number of strict increases in $F^*(t)$. Let $0 = i_0 < i_1 < \cdots < i_{k(t)} < i_{k(t)+1} = n$ denote the jump points such that $F^*(t)$ is constant over each segment $\{i_{\ell-1} + 1, \ldots, i_\ell\}$. Then, the tangent cone admits the decomposition

$$T_{K_t}\left(\frac{F^*(t)}{\eta}\right) = \bigoplus_{\ell=1}^{k(t)+1} \mathcal{M}^{n_\ell}, \quad \text{where } n_\ell := i_\ell - i_{\ell-1},$$

with each $\mathcal{M}^{n_\ell} := \{\theta \in \mathbb{R}^{n_\ell} : \theta_1 \leq \cdots \leq \theta_{n_\ell}\}$ denoting the isotonic cone in $\mathbb{R}^{n_\ell}$. This decomposition follows from Remark 2.1 in Chatterjee et al. [12]. Therefore, the statistical dimension satisfies

$$\delta\left(T_{K_t}\left(\frac{F^*(t)}{\eta}\right)\right) = \sum_{\ell=1}^{k(t)+1} \delta(\mathcal{M}^{n_\ell}).$$

From Example 2.2 in [12], we have that $\delta(\mathcal{M}^m) \leq \log(em)$. Moreover, applying Jensen's inequality, we conclude

$$\delta\left(T_{K_t}\left(\frac{F^*(t)}{\eta}\right)\right) \leq (1 + k(t)) \cdot \log\left(\frac{en}{1 + k(t)}\right).$$

Hence, the Gaussian width is bounded as

$$w\left(T_{K_t}\left(\frac{F^*(t)}{\eta}\right)\right) \leq \sqrt{(1 + k(t)) \cdot \log\left(\frac{en}{1 + k(t)}\right)}$$

Therefor, taking $\eta = \sup_{t \in \mathbb{R}} \sqrt{(1 + k(t)) \cdot \log\left(\frac{en}{1+k(t)}\right)}$, from Theorem 1 and Equation 61 we conclude that

$$\mathbb{E}\left(\frac{1}{n}\sum_{i=1}^n \mathrm{CRPS}(\widehat{F}_i, F_i^*)\right) \leq C\sup_{t \in \mathbb{R}} \cdot \frac{1 + k(t)}{n} \log\left(\frac{en}{1 + k(t)}\right).$$

$\square$

## G.11 Proof of Corollary 6

*Proof.* By Theorem 1, it is enough to bound

$$\sup_{t \in \mathbb{R}} \mathbb{E} \left[ \sup_{\theta \in K_t \,:\, \|\theta - F^*(t)\| \leq \eta} g^\top (\theta - F^*(t)) \right],$$

where where $g \sim N(0, I_n)$. Let, $t \in \mathbb{R}$ and $g \sim \mathcal{N}(0, I_n)$ be a standard Gaussian vector in $\mathbb{R}^n$, and let $K_t \subset \mathbb{R}^n$ be a constraint set. Now we analyze the quantity

$$\mathbb{E} \left[ \sup_{\theta \in K_t : \|\theta - F^*(t)\| \leq \eta} \langle g, \theta - F^*(t) \rangle \right].$$

To that end, remember that

$$K_t = \left\{ \theta \in R^n : \left\| D^{(r)} \theta \right\|_1 \leq \frac{V_t^*}{n^{r-1}} \right\}.$$

and using Definition 4 we have that $\mathbb{E} \left[ \sup_{\theta \in K_t : \|\theta - F^*(t)\| \leq \eta} \langle g, \theta - F^*(t) \rangle \right]$ is the same as the Gaussian complexity of $\{ \theta \in K_t - F^*(t) : \|\theta\| \leq \eta \}$, this is

$$\mathcal{R} \left( \{ \theta \in K_t - F^*(t) : \|\theta\| \leq \eta \} \right),$$

see Equation (63) for such notation. Then, following the same line of arguments as in the proof of Theorem 2.2 in [23], see Equation (129) in their Supplementary Material, it follows that

$$\mathcal{R} \left( \{ \theta \in K_t - F^*(t) : \|\theta\| \leq \eta \} \right) \leq C \left[ (s+1) \log \left( \frac{en}{s+1} \right) \right]$$

and the desired result is obtained.

$\square$

### G.12 Auxiliary Lemmas

**Definition 3.** *Let $K \subset \mathbb{R}^n$ and $t > 0$. A subset $P$ of $K$ is a packing of $K$ if $P \subset K$ and the set $\{B_t(x)\}_{x \in P}$ is pairwise disjoint. Then, the $t$-packing number of $K$, denoted as $M(t, K, \|\cdot\|)$, is defined as the cardinality of any maximum $t$-packing.*

**Lemma 2.** *[Variant of Dudley's inequality.] Let $S \subset \mathbb{R}^n$ be a finite set and $\epsilon^{(j)} \in \mathbb{R}^n$ be a vector of mean zero independent SubGaussian(1) random variables, for $j = 1 \ldots, m$. Suppose that $0 \in S$ and $v \in S$ implies that $\|v\| \leq D_n/2$ for some $D_n > 0$. Then there exists a constant $C > 0$ such that*

$$\mathbb{E}\left( \max_{j=1,\ldots,m} \max_{v \in S} v^\top \epsilon^{(j)} \right) \leq C \left( D_n \sqrt{\log m} + \int_0^{D_n/4} \sqrt{\log M(\delta, S, \|\cdot\|)} \, d\delta \right)$$

*where $M(\delta, S, \|\cdot\|)$ is the packing number of $S$ with respect to the metric $\|\cdot\|$.*

*Proof.* For $C \geq 1$, let $S_n$ be a maximal $D_n 2^{-l}$-separated subset of $S$, i.e.,

$$\min_{v,u \in S} \|v - u\| > D_n 2^{-l}.$$

By construction, $|S_l| = M(D_n 2^{-l}, S, \|\cdot\|)$. Clearly, because of the maximality,

$$\max_{v \in S} \min_{u \in S_l} \|v - u\| \leq D_n 2^{-l}.$$

Furthermore, $S_l = S$ for large enough $l$. Hence, we let

$$N = \min \{l \geq 1 : S_l = S\}.$$

Also, for $l \geq 1$, let $\pi_l$ be the function that assigns $v \in S$ to the point in $S_l$ closest to $v$. By definition,

$$\|\pi_l(v) - v\| \leq D 2^{-l}$$

for all $v \in S$ and $l \in \mathbb{N}$. We also write $S_0 = \{0\}$ and so $\pi_0(v) = 0$ for all $v \in S$. Next, we observe that

$$v^\top \epsilon^{(j)} = \sum_{l=1}^N (\pi_l(v) - \pi_{l-1}(v))^\top \epsilon^{(j)}$$

for all $v \in S$. It follows that

$$\max_{j=1,\ldots,m} \max_{v \in S} v^\top \epsilon^{(j)} \leq \max_{j=1,\ldots,m} \max_{v \in S} \sum_{l=1}^N (\pi_l(v) - \pi_{l-1}(v))^\top \epsilon^{(j)}$$

$$\leq \sum_{l=1}^N \max_{j=1,\ldots,m} \max_{v \in S} (\pi_l(v) - \pi_{l-1}(v))^\top \epsilon^{(j)}$$

and so

$$\mathbb{E}\left( \max_{j=1,\ldots,m} \max_{v \in S} v^\top \epsilon^{(j)} \right) \leq \sum_{l=1}^N \mathbb{E}\left( \max_{j=1,\ldots,m} \max_{v \in S} (\pi_l(v) - \pi_{l-1}(v))^\top \epsilon^{(j)} \right).$$

However, notice that for all $u > 0$,

$$\mathbb{P}\left( (\pi_l(v) - \pi_{l-1}(v))^\top \epsilon^{(j)} \geq u \right) \leq 2 \exp\left( \frac{-u^2}{2\|\pi_l(v) - \pi_{l-1}(v)\|^2} \right)$$

and

$$\|\pi_l(v) - \pi_{l-1}(v)\| \leq \|\pi_l(v) - v\| + \|\pi_{l-1}(v) - v\|$$
$$\leq D_n 2^{-l} + D_n 2^{l-1}$$
$$\leq 3 D_n 2^{-l}$$

which implies, by the subGaussian maximal inequality, that

$$\mathbb{E}\left( \max_{j=1,\ldots,m} \max_{v \in S} (\pi_l(v) - \pi_{l-1}(v))^\top \epsilon^{(j)} \right) \leq \frac{3 C D_n}{2^l} \sqrt{\log(2m |S_l||S_{l-1}|)}$$

$$\leq \frac{3 C D_n}{2^l} \sqrt{\log(2m |S_l|^2)}$$

$$\leq \frac{3 C D_n}{2^l} \sqrt{\log(2m M(D 2^{-l}, S, \|\cdot\|))}$$

for some constant $C > 0$. Therefore,

$$
\begin{aligned}
\mathbb{E}\left(\max_{j=1,\ldots,m} \max_{v \in S} v^\top \epsilon^{(j)}\right) &\leq 3C \sum_{l=1}^{N} \frac{D_n}{2^l} \sqrt{\log m + \log(2M(D_n 2^l, S, \|\cdot\|))} \\
&\leq 3C\sqrt{\log m} \sum_{l=1}^{N} \frac{D_n}{2^l} + 3C \sum_{l=1}^{N} \frac{D_n}{2^l} \sqrt{\log\left(2M(D2^{-l}, S, \|\cdot\|)\right)} \\
&\leq 3CD_n\sqrt{\log m} + 3C \sum_{l=1}^{N} \frac{D_n}{2^l} \sqrt{\log\left(2M(D_n 2^{-l}, S, \|\cdot\|)\right)} \\
&\leq 3CD_n\sqrt{\log m} + 6C \sum_{l=1}^{N} \int_{D_n/2^{l+1}}^{D/2^l} \sqrt{\log\left(2M(r, S, \|\cdot\|)\right)} dr \\
&= 3CD_n\sqrt{\log m} + 6C \int_{D_n/2^{N+1}}^{D/2} \sqrt{\log\left(2M(r, S, \|\cdot\|)\right)} dr \\
&\leq 3CD_n\sqrt{\log m} + 6C \int_{0}^{D_n/2} \sqrt{\log\left(2M(r, S, \|\cdot\|)\right)} dr \\
&\leq 3CD_n\sqrt{\log m} + 6C \int_{0}^{D_n/4} \sqrt{\log\left(2M(r, S, \|\cdot\|)\right)} dr \\
&\quad + 6C \int_{0}^{D_n/4} \sqrt{\log\left(2M(r + D_n/4, S, \|\cdot\|)\right)} dr \\
&\leq 3CD_n\sqrt{\log m} + 12C \int_{0}^{D_n/4} \sqrt{\log\left(2M(r, S, \|\cdot\|)\right)} dr \\
&\leq 3CD_n\sqrt{\log m} + 24C \int_{0}^{D_n/4} \sqrt{\log\left(M(r, S, \|\cdot\|)\right)} dr
\end{aligned}
$$

and the claim follows.

$\square$

**Lemma 3.** *With the notation and conditions of Lemma 2, if $S \subset \mathbb{R}$ arbitrary (not necessarily finite), then*

$$
\mathbb{E}\left(\max_{j=1,\ldots,m} \sup_{v \in S} v^\top \epsilon^{(j)}\right) \leq C\left(D_n\sqrt{\log m} + \int_{0}^{D_n/4} \sqrt{\log M\left(\delta, S, \|\cdot\|\right)} d\delta\right).
$$

*Proof.* Let $\tilde{S} \subset S$ be a countable set such that

$$
\max_{j=1,\ldots,m} \sup_{v \in S} v^\top \epsilon^{(j)} = \max_{j=1,\ldots,m} \sup_{v \in \tilde{S}} v^\top \epsilon^{(j)}.
$$

Let $\tilde{S}_l$ the set consisting of the first $l$ elements of $\tilde{S}$. Without loss of generality let's assume that $0 \in \tilde{S}_l$ for all $l$. Then by Lemma 2, we have that

$$
\begin{aligned}
\mathbb{E}\left(\max_{j=1,\ldots,m} \sup_{v \in \tilde{S}_l} v^\top \epsilon^{(j)}\right) &\leq C\left(D_n\sqrt{\log m} + \int_{0}^{D_n/4} \sqrt{\log M\left(\delta, \tilde{S}_l, \|\cdot\|\right)} d\delta\right) \\
&\leq C\left(D_n\sqrt{\log m} + \int_{0}^{D_n/4} \sqrt{\log M\left(\delta, S, \|\cdot\|\right)} d\delta\right)
\end{aligned} \tag{62}
$$

for all $l \geq 1$. Hence, the claim follows by letting $l \to \infty$ in (62) and applying the Monotone Convergence Theorem. $\square$

**Definition 4.** *For $t \in \mathbb{R}$ let $\epsilon(t) = w(t) - F^*(t) \in \mathbb{R}^n$. Then for a set $\mathcal{V} \subset \mathbb{R}^n$ define*

$$
R_t(\mathcal{V}) = \mathbb{E}\left(\sup_{v \in \mathcal{V}} \sum_{i=1}^{n} v_i \epsilon_i(t)\right).
$$

*Furthermore, define the Gaussian complexity of $\mathcal{V}$ as*

$$\mathcal{R}(\mathcal{V}) = \mathbb{E}\left(\sup_{v \in \mathcal{V}} \sum_{i=1}^{n} v_i g_i\right), \tag{63}$$

*where $g \sim N(0, I)$.*

**Lemma 4.** *There exists a universal constant such that for any set $\mathcal{V} \subset \mathbb{R}^n$ it holds that*

$$\sup_{t \in \mathbb{R}} R_t(\mathcal{V}) \leq L\mathcal{R}(\mathcal{V}),$$

*where $L > 0$ is a universal constant.*

*Proof.* Fix $t \in \mathbb{R}$. Then for $v \in \mathcal{V}$, let $Y_v = v^\top \epsilon(t)$ and $X_v = v^\top g$. Next, observe that

$$\sup_{u,v \in \mathcal{V}} |Y_u - Y_v| = \sup_{u,v \in \mathcal{V}} (Y_u - Y_v) = \sup_{u \in \mathcal{V}} Y_u + \sup_{v \in \mathcal{V}} -Y_v.$$

Hence, for any $v_0 \in \mathcal{V}$,

$$
\begin{aligned}
\mathbb{E}\left(\sup_{u,v \in \mathcal{V}} |Y_u - Y_v|\right) &= \mathbb{E}\left(\sup_{u \in \mathcal{V}} Y_u\right) + \mathbb{E}\left(\sup_{v \in \mathcal{V}} -Y_v\right) \\
&\geq \mathbb{E}\left(\sup_{u \in \mathcal{V}} Y_u\right) + \mathbb{E}(-Y_{v_0}) \\
&\geq \mathbb{E}\left(\sup_{u \in \mathcal{V}} Y_u\right) \\
&= R_t(\mathcal{V}).
\end{aligned}
\tag{64}
$$

Now, we observe that $\epsilon_i(t)$ is sub-Gaussian(1). Hence, by Theorem 2.1.5 from Talagrand [59], we have that

$$\mathbb{E}\left(\sup_{u,v \in \mathcal{V}} |Y_u - Y_v|\right) \leq L\mathbb{E}(\sup_{v \in \mathcal{V}} v^\top g) = L\mathcal{R}(\mathcal{V}),$$

for a universal constant that does not depend on $t$. Hence, the claim follows.

$\square$

**Definition 5.** *For a measurable function $f : \mathbb{R} \to \mathbb{R}$ we define its distribution with respect to the Lebesgue measure as the function $\mu_f : [0, \infty) \to \mathbb{R}$ given as*

$$\mu_f(\lambda) := \mu(\{x : |f(x)| > \lambda\})$$

*where $\mu$ is the Lebesgue measure.*

**Definition 6.** *For a measurable function $f : \mathbb{R} \to \mathbb{R}$ we define its decreasing rearrangement $D(f) : [0, \infty) \to \mathbb{R}$ given as*

$$D(f)(t) := \inf\{\lambda \geq 0 : \mu_f(\lambda) \leq t\}$$

*where $\mu$ is the Lebesgue measure.*

**Lemma 5.** *Suppose that $f$ can be written as*

$$f(t) = \sum_{l=1}^{n-1} b_l 1_{E_l}(t)$$

*for $b_1 \geq \ldots \geq b_{n-1} \geq 0$ and measurable sets $E_l \subset \mathbb{R}$ that are pairwise disjoint. Then the decreasing rearrangement of $f$ is given by*

$$D(f)(t) = \sum_{l=1}^{n-1} b_l 1_{[m_{l-1}, m_l)}(t)$$

*where*

$$m_l := \sum_{k=1}^{l} \mu(E_k), \quad l = 1, \ldots, n-1,$$

*and with $m_0 = 0$.*

*Proof.* This is Example 1.6 in Bennett and Sharpley [6]. □

**Lemma 6.** *For any integrable functions* $f$ *and* $g$ *the following hold:*

1.

$$\int_{\mathbb{R}} |f(t)|^2 dt = \int_0^\infty |D(f)(t)|^2 dt.$$ (65)

2. *[G.H. Hardy and J.E. Littlewood].*

$$\int_{\mathbb{R}} |f(t)g(t)| dt \leq \int_0^\infty D(f)(s) \cdot D(g)(s) ds.$$ (66)

3. *Suppose that* $f$ *is decreasing and continuous in* $[0, a)$ *for some* $a > 0$, *and* $f(t) = 0$ *otherwise. Then* $f(t) = D(f)(t)$ *for all* $t \in [0, \infty)$.

*Proof.* The claim in (65) follows from Proposition 1.8 in Bennett and Sharpley [6]. The inequality in (66) is the well-known G.H Hardy and J.E Littlewood inequality, see for instance Theorem 2.2 in Bennett and Sharpley [6].

We now prove the final claim. Let $t \geq 0$. Suppose that $t \in [0, a)$. Then

$$\begin{aligned} D(f)(t) &= \inf\{\lambda \geq 0 : \sup\{x : f(x) > \lambda\} \leq t\} \\ &= \inf\{\lambda \geq 0 : \sup\{x \geq 0 : f(x) > \lambda\} \leq t\}. \end{aligned}$$ (67)

Hence, for $0 \leq \lambda < f(t)$, the continuity of $f$ in $[0, a)$, implies that there exists $t' \in (t, a)$ such that $\lambda < f(t') \leq f(t)$. Thus, $\sup\{x \geq 0 : f(x) > \lambda\} > t$. On the other hand, if $f(0) \geq \lambda \geq f(t)$, then, also by the continuity of $f$ in $[0, a)$,

$$\sup\{x \geq 0 : f(x) > \lambda\} = \inf\{x \geq 0 : f(x) = \lambda\}.$$

Hence, from (67), we obtain $f(t) = D(f)(t)$ for $t \in [0, a)$.

Suppose now that $t \in [a, \infty)$. Then

$$\mu_f(0) \leq a \leq t.$$

Hence, $D(f)(t) = 0$ and the claim follows. □

