# OpenReview forum: "Risk Bounds For Distributional Regression"
_NeurIPS.cc/2025/Conference — NeurIPS 2025 poster_

### Official Review · Reviewer_WHfB · 2025-06-05

**Clarity:** 3
**Significance:** 4
**Originality:** 4
**Rating:** 6
**Confidence:** 3

**Summary:**

The paper introduces a general procedure to learn a set of cdfs under constraints, and derive general estimation guarantees for this procedure. They then show several special cases of this approach and their theory, thereby strengthening existing results. Finally they close the paper by showing that their new approach outperforms state of the art methods in real and simulated data.

**Questions:**

- I dont fully understand in the first part whether (3) is new in this full generality of whether this has been done before?
- There seems to be something wrong in the Probability statement of Theorem 3 (also in the proof in the appendix). I guess the > \eta needs to shifted to the right?
- In the proof of Theorem 3, how do you get from <= 2nB to <= 2^j \eta^2?
- It would be great to have a bit more details in Section 4.2. Now the K_t depend on a neural network that I guess needs to be optimized? How is this handled in practice? I am not that interested in the structure of the neural net, but more how the optimization in (3) works together with the optimization of the neural net.
- Maybe a stupid question, but why is cdf learning with a regularization on TV distance called trend filtering?

**Ethical Concerns:**

["NO or VERY MINOR ethics concerns only"]

**Final Justification:**

I think this is a great paper with solid theory, generalizing previous work and providing general estimation guarantees. The authors addressed my questions adequately and also replied to my proof questions in details, which was interesting to read. I also see no unresolved issue and I am thus deciding to leave my score.

**Limitations:**

Yes

**Paper Formatting Concerns:**

MP

**Quality:**

4

**Strengths And Weaknesses:**

Strengths
---------------
- It appears to me the theoretical results are very strong
- The paper is dense but well written. The proofs could benefit from a bit more explanation, but overall it is reasonable.
- Though it is somewhat hidden in the appendix, the simulation and application section appears thorough as well

Weaknesses
----------------
If I had any critique of the paper than that it is almost too dense and appears to follow the trend in statistics to just throw as many results into a paper as possible. In particular, I am not sure what Appendix B is for. The results are great, but it appears to stand completely on it's own just making the paper longer. Similarly, while the main paper has a clear thread and the results have their place, I would wish for a bit more explanation in Section 4.2. Now K_t depends on a neural net as far as I understand; how exactly is this handled? (see also my questions below)

---

> ### Author Rebuttal · Authors · 2025-07-30
>
> We thank the reviewer for the endorsement of our work and for the helpful suggestions to improve clarity and presentation. Below, we address the general concerns and specific questions in turn.
>
> ---
>
> **Q1: Novelty of Equation (3)**
>
> Thank you for the question. To the best of our knowledge, Equation (3), which formulates CDF estimation in distributional regression as a family of projection estimators indexed by $t$ and constrained to general sets $K_t \subset \mathbb{R}^n$, is novel in this level of generality. While projection-based methods have been used in other contexts, such as mean estimation under monotonicity or trend filtering constraints, their application to distributional regression problems has been limited. Our framework unifies these structural constraints within a general formulation for CDF estimation, allowing for a broad class of constraint sets including isotonic, total variation, and neural network-based sets, to be incorporated into a single projection-based estimator. This unified view enables new theoretical analysis and practical extensions for distributional regression.
>
> ---
>
> **Q2: Probability statement in Theorem 3**
>
> Thank you for pointing this out. We confirm that the inequality inside the probability statement in Theorem 3 is correct as written, though we agree it may initially appear unintuitive. The bound controls the $\ell_2$ error $\vert\vert\widehat{F}(t) - F^{\ast}(t)\vert\vert_2$, and the probability is taken over the event that this exceeds a threshold $\eta$ and an additive term involving $\sqrt{n} \cdot \sup_t \vert\vert F^{\ast}(t) - G(t)\vert\vert_\infty$. In applications, the relevant quantity is the average per-sample error, which is normalized by $\sqrt{n}$. Therefore, after normalizing by $\sqrt{n}$, the final bound depends on both the approximation error, $\sup_t \vert\vert F^{\ast}(t) - G(t)\vert\vert_\infty$, and $\eta/\sqrt{n}$. In consequence, it suffices to choose $\eta$ of order $\sqrt{\log n} + \sqrt{n} \cdot \sup_t \vert\vert F^{\ast}(t) - G(t)\vert\vert_\infty$.
>
> This behavior is confirmed in Corollary 4 for our Dense ReLU network example. There, we assume that $G^{\ast}(\cdot, t) \in \mathcal{H}(l, \mathcal{P})$ where $F_i^{\ast}(t) = G^{\ast}(x_i, t) $, see Assumption 1 in Appendix C, and define the constraint set
>
>
> $$
> K_t = \\{ \theta \in \mathbb{R}^n : \theta_i = f(x_i), \text{ for some } f \in \mathcal{F}(L, \nu) \\}
> $$
>
>
> where $\mathcal{F}(L, \nu)$ is a class of ReLU networks with appropriate choice for the number of layers and neurons. In this case, the approximation error scales as $\phi_n = \max_{(p, M) \in \mathcal{P}} n^{-p/(2p + M)}$, and the final rate becomes
>
> $$
> \sup_{t \in \mathbb{R}} \frac{1}{n} \sum_{i=1}^n\left(\widehat{F}_i(t)-F_i^*(t)\right)^2 = O_P\left(\frac{\log n}{n} + \phi_n \log^4 n \right).
> $$
>
> Thus, the resulting rate is meaningful and matches known convergence rates for mean estimation under Gaussian noise using Dense ReLU networks. This shows that our bound is both general and sharp in concrete, practically relevant settings.
>
> ---
>
> **Q3: Step from $\leq 2nB$ to $\leq 2^j \eta^2$ in the proof of Theorem 3**
>
> Thank you for this question. The step involving the transition from the upper bound $\sup_t \vert\vert\widehat{F}(t) - G(t)\vert\vert^2 \leq 2nB$ to the summation over intervals of the form $(2^{j-1}\eta^2, 2^j \eta^2]$ corresponds to a peeling argument.
>
> We briefly clarify the logic. First, we restrict attention to the event
>
> $$
>  \\{\sup_{t \in \mathbb{R}} \vert\vert\widehat{F}(t) - G(t)\vert\vert^2 > \eta^2 \\}
> \cap
>  \\{\sup_{t \in \mathbb{R}} \vert\vert\widehat{F}(t) - G(t)\vert\vert^2 \leq 2nB\\},
> $$
>
> which allows us to consider only deviations above $\eta^2$ and below the known upper bound $2nB$.
>
> We then partition the interval $[\eta^2, 2nB]$ into dyadic subintervals $(2^{j-1} \eta^2, 2^j \eta^2]$ for $j = 1, \ldots, J$, where $J$ is the smallest integer such that $2^J \eta^2 \geq 2nB$. This allows us to write:
>
> $$
> \mathbb{P}\left( \sup_{t \in \mathbb{R}} \vert\vert\widehat{F}(t) - G(t)\vert\vert^2 > \eta^2,
> \sup_{t \in \mathbb{R}} \vert\vert\widehat{F}(t) - G(t)\vert\vert^2 \leq 2nB \right)
> \leq
> \sum_{j=1}^J \mathbb{P}\left( \sup_{t \in \mathbb{R}} \vert\vert\widehat{F}(t) - G(t)\vert\vert^2
> \in (2^{j-1}\eta^2, 2^j \eta^2] \right).
> $$
>
> Each term in the sum corresponds to the event that the supremum falls within a specific dyadic shell.
>
> We have added a brief clarification in the proof to make this peeling step more transparent to the reader.
>
> ---
>
> **Q4: Practical implementation and optimization of neural network-based $K_t$**
>
> Thank you for the question. In practice, we fix the network architecture in advance by selecting the number of layers and neurons ($L$ and $\nu$). For each threshold value $t$, we then solve the projection problem in Equation (3) by training a neural network that maps covariates $x_i$ to estimates $F_i(t)$, subject to the specified architecture.
>
> This training procedure is repeated independently for each $t$, which makes it highly parallelizable and scalable in practice. While the structure of the network remains fixed across $t$, the parameters are learned separately for each projection, effectively allowing us to approximate the conditional CDF at each threshold using a specialized network.
>
> ---
>
> **Q5: Terminology — Why “trend filtering” is used for CDF learning with total variation regularization**
>
> Thank you for the question. The reason we refer to this estimator as “trend filtering” is because the terminology is standard in the literature, see [41, 61, 23], when the constraint set $K_t$ is defined via a bound on total variation of order $r$. We will add a clarifying sentence in the revised manuscript to make this connection explicit.
>
> ---
>
> ***References***
>
> [23] Adityanand Guntuboyina, Donovan Lieu, Sabyasachi Chatterjee, and Bodhisattva Sen. Adaptive risk bounds in univariate total variation denoising and trend filtering. The Annals of Statistics, 48:205–229, 2020.
>
> [41] Enno Mammen and Sara Van De Geer. Locally adaptive regression splines. The Annals of Statistics, 25(1):387–413, 1997.
>
> [61] Ryan J Tibshirani. Adaptive piecewise polynomial estimation via trend filtering. 2014

---

> > ### Comment · Reviewer_WHfB · 2025-08-06
> >
> > Thanks a lot for your detailed and clarifying comments in particular also for the proof of Theorem 3! I also somehow read the probability statement in Theorem 3 wrong, it makes a lot more sense now. With this I am happy to leave my score

---

### Official Review · Reviewer_6oF7 · 2025-06-22

**Clarity:** 3
**Significance:** 3
**Originality:** 2
**Rating:** 4
**Confidence:** 3

**Summary:**

This paper studies risk bounds for distributional regression and proposes a unified framework that accommodates both convex and non-convex constraints. Theoretical convergence guarantees are established, and the framework is shown to be broadly applicable. Extensive numerical experiments are conducted to validate its performance.

**Questions:**

See the questions in weakness. I am happy to increase the score if the authors can solve my question.

**Ethical Concerns:**

["NO or VERY MINOR ethics concerns only"]

**Final Justification:**

Based on all the discussions, I choose to keep my positive score

**Limitations:**

See the questions in weakness.

**Quality:**

3

**Strengths And Weaknesses:**

**Strengths**: The paper is well-written and addresses a fundamental and interesting problem. It extends existing frameworks to the context of distributional regression, with broad applicability. The theoretical bounds derived appear sound, and the authors support their claims with extensive numerical results.

**Weakness**: while the paper is well-written, I have some concerns regarding its novelty.

The authors should more clearly highlight the contributions of this work in comparison to existing literature. It would be helpful to include a structured comparison—such as a table or bullet-point list—detailing differences in problem formulations, theoretical results, and methodological innovations.

I am particularly concerned about the originality of some key components. For instance, the projection technique used in Lemma 1 appears to be standard in this line of research. Moreover, Theorem 1 plays a central role in the paper but seems closely related to Theorem A.1 in [23]. Notably, a large portion of the proof of Theorem 1—approximately 80%—appears to be directly adapted from the proof of Theorem A.1 in [23]. This raises questions about the specific challenges addressed and the novel contributions of Theorem 1.

---

> ### Author Rebuttal · Authors · 2025-07-30
>
> Thank you very much for the comments and appreciation of our work. We provide clarifications and a detailed breakdown of the theoretical and methodological advances in our responses below.
>
>
> ---
>
> **Q1: Contributions and structured comparison with prior work**
>
> Thank you for the helpful suggestion. We have revised the manuscript to more clearly organize and expand our theoretical comparisons, including both existing content and several new additions. We have also created a consolidated summary (Rate, assumption, method) in Table 2, partially presented below.
>
> **Regarding the bounds**
>
> We now explicitly compare the convergence rates obtained for our isotonic, trend filtering, and Dense ReLU network examples to those of corresponding methods in the CDF estimation literature, with a focus on both assumptions and statistical accuracy.
>
> As an example, we reproduce below the updated discussion following Corollary 2 concerning our isotonic estimator:
>
> > The result in Corollary 2 establishes that distributional isotonic regression achieves an estimation rate of $n^{-2/3}$ for both the expected average CRPS and the worst-case MSE, as shown in (10). This result improves upon Theorem 3 in Hänzi et al. [28] in the univariate case, which only demonstrated convergence in probability for isotonic distributional regression. However, we emphasize that Hänzi et al. [28] study the more general setting of multivariate covariates, while our analysis is restricted to the univariate case ($d = 1$).
>
> Due to space constraints in this response, we briefly comment on the new comparisons we have added.
>
> Our isotonic result and Theorem 3.3 of Mösching and Dümbgen [44] are compared. Translating their setup into our notation, their quantity $F_{X_i}(t) := \mathbb{P}(Y_i \leq t \mid X_i)$ corresponds to our $F_i^{\ast }(t) := \mathbb{P}(y_i \leq t)$. The monotonicity assumption on $x \mapsto F_x(t)$ aligns with our use of an isotonic constraint on the sequence $F_1^{\ast }(t) \leq \cdots \leq F_n^{\ast }(t)$. However, our result does not assume Hölder continuity for the map $x \mapsto F_x(t)$, or dense covariates in its domain. Despite these weaker requirements, we achieve a faster rate of $n^{-2/3}$ (up to logarithmic factors) for both average CRPS and worst-case MSE, compared to their $n^{-2\alpha/(2\alpha+1)}$ rate for the worst-case MSE, where $\alpha \in (0,1]$.
>
> We compared our Dense ReLU estimator with DRF [10] and EnG [58], two methods explicitly designed for conditional distribution estimation.
>
> DRF estimates the conditional CDF using forest-based weights to form a weighted empirical distribution. EnG instead models $Y = g^{\ast }(X, \varepsilon)$, with $\varepsilon \sim \text{Unif}[0,1]$, and trains a neural network via energy score minimization; the estimated CDF is obtained by sampling from the learned model. Translating their notation into our notation, both methods assume that $G^{\ast }(x, t) := \mathbb{P}(Y \leq t \mid X = x)$ is Lipschitz in $x$ for each fixed $t$, and that the covariate density is bounded away from zero and infinity.
>
> Our method instead assumes $G^{\ast }(\cdot, t) \in \mathcal{H}(l, \mathcal{P})$, a broader class that includes the Lipschitz case ($\mathcal{H}(1, \\{(1, d_0)\\})$) and mitigates the curse of dimensionality, which typically affects Lipschitz-based methods in high-dimensional settings. Moreover, we do not require assumptions on the covariate density. While DRF and EnG show convergence in probability at fixed $t$, our method yields high-probability bounds on the worst-case MSE uniformly over $t$, with explicit rate $\frac{\log n}{n} + \phi_n \log^4 n$.
>
> **Regarding the tightness of the bounds**
>
> In the revision, we have incorporated a discussion regarding the tightness of the upper bounds. Specifically, the following has been added:
>
> Theorems 1–3 are general results that apply to a wide range of constraint sets, both convex and non-convex, and are not tailored to any specific estimation problem. For this reason, we do not provide lower bounds at the general level.
>
> However, in specific examples like isotonic regression, or total variation, the rates we obtain using these general theorems match known minimax rates for mean estimation with Gaussian noise. In this sense, the bounds we obtain are optimal for these concrete examples.
>
> **Regarding the novelty of the general Theorem 1, 2, and 3 in CDF literature**
>
> The following has been added:
>
> Theorems 1–3 adapt classical techniques from regression, where projection-based estimators have been studied in both convex (e.g., [5, 14]) and non-convex (e.g., [46]) settings. These prior results focus on controlling mean squared error at a fixed point and were not developed for CDF estimation, which requires uniform control over all $t \in \mathbb{R}$. To our knowledge, our extensions provide the first such results for CDF estimation.
>
> ---
>
> **Q2: Originality of the projection technique in Lemma 1**
>
> Thank you for the question. We agree that Lemma 1 is technically straightforward. However, its purpose is not to introduce a novel projection technique per se, but rather to establish a conceptual and formal bridge between the original CRPS-based formulation (1) and the projection estimator (2) that underlies our methodological framework. This connection enables us to reinterpret the constrained distributional regression problem as a sequence of constrained projection problems across $t$, which is key to the development of our general theory. In particular, it provides the foundation for extending the estimator to time-varying constraint sets $K_t$, and for analyzing its performance under different structural assumptions, including those defined by neural networks, trend filtering, and monotonicity.
>
> ---
>
> **Q3: Novelty of Theorem 1 relative to Theorem A.1 in [23]**
>
> Thank you very much for pointing this out. Our result in Theorem 1 is conceptually related to Theorem A.1 of [23], but technically distinct. In their setting, the observations are Gaussian $Y = \mu + Z$, with $Z \sim \mathcal{N}(0, \alpha^2 I_n)$. The estimator $\hat{\mu}$ is the Euclidean projection of $Y$ onto a convex set $K$. Their goal is to bound the squared error $\vert\vert\hat{\mu} - \mu\vert\vert_2^2 = t^{\ast }$ in expectation, which is reduced to be controlled with high probability. For this, the "KEY" step is to show that the squared error $\vert\vert\hat{\mu} - \mu\vert\vert_2^2 = t^{\ast }$ concentrates around a deterministic benchmark $t_\mu$, which characterizes the scale of estimation error. This is proved using a peeling argument that combines the strict concavity of an associated process with Gaussian concentration.
>
> Our setting also relies on convex projection, but differs in several important ways. We observe Bernoulli vectors $w(t) = 1_{\{ y_i \leq t \}}$ (their $Y$), with sub-Gaussian, heteroscedastic noise $\epsilon(t) = w(t) - F^{\ast }(t)$ (their $Z$), and our target is the CRPS rather than squared error. Despite these differences, our analysis also hinges on addressing the same "KEY" step: showing that the squared $\ell_2$ error $\vert\vert\widehat{F}(t) - F^{\ast }(t)\vert\vert_2^2$ (their $t^{\ast }$) concentrates around a complexity-driven benchmark $\eta$ (their $t_\mu$).
>
> However, our proof takes a completely different route. We use (i) a sub-Gaussian concentration inequality for Lipschitz and separately convex functions to control a local empirical process, and (ii) a star-shaped self-normalization argument to extend this control to the full constraint set $K_t$. Because of the differences in observations, loss function, and techniques, our result is not a direct application of theirs.
>
> ---
> ***Table 2***
> Below, we present Table 2 partially, showing only the summary of results for the Isotonic and Dense ReLU Network examples due to space constraints.
>
> ### **Isotonic**
>
> | Work         | Method        | Assumptions                                                   | Convergence Rate                          |
> |--------------|---------------|----------------------------------------------------------------|-------------------------------------------|
> | Prior Work   | [27]          | $K_t$ monotone cone, $F^*(t)\in K_t$                          | $o(1)$                                    |
> | Prior Work   | [44]          | $K_t$ monotone cone, $F^*(t)\in K_t$, dense design, Hölder    | $n^{-2\alpha/(2\alpha+1)}, \alpha\le 1$ (Worst MSE)    |
> | This Work    | General rate  | $K_t$ monotone cone, $F^*(t)\in K_t$                          | $n^{-2/3}$ (CRPS, Worst MSE)              |
> | This Work    | Fast rate     | $K_t$ monotone cone, $F^*(t)\in K_t$, with few strict increases, $k(t)\ll n$                             | $\frac{1 + k(t)}{n} \log \left( \frac{en}{1 + k(t)} \right)$ (CRPS) |
>
> ### **Dense ReLU Networks**
>
> | Work         | Method        | Assumptions                                                                                       | Convergence Rate                          |
> |--------------|---------------|----------------------------------------------------------------------------------------------------|-------------------------------------------|
> | Prior Work   | [36]          | Regression $y_i = f(x_i) + \epsilon_i$, $f \in \mathcal{H}(l, \mathcal{P})$                        | $\phi_n$ (MSE)                            |
> | Prior Work   | [10]          | $F_i^{\ast }(t) = G^{\ast }(x_i, t) = \mathbb{P}(y_i \leq t \mid x_i)$, $G^{\ast }(\cdot, t) \in \mathcal{H}(1, \\{(1,d_0)\\})$, Forest-Weighted empirical CDF | $o(1)$                                    |
> | Prior Work   | [57]          | As above but Neural Net-based sampling approach                                            | $o(1)$                                    |
> | This Work    | General rate  | $F_i^{\ast}(t) = G^{\ast}(x_i, t)$, $G^{\ast}(\cdot, t) \in \mathcal{H}(l, \mathcal{P})$                          | $\frac{\log n}{n} + \phi_n \log^4 n$ (CRPS, Worst MSE) |

---

> > ### Comment · Reviewer_6oF7 · 2025-08-04
> >
> > I am satisfied with the authors' rebuttal and I will maintain my positive score.

---

### Official Review · Reviewer_LnJT · 2025-06-26

**Clarity:** 3
**Significance:** 3
**Originality:** 3
**Rating:** 5
**Confidence:** 2

**Summary:**

This paper investigates the risk bounds of nonparametric distribution regression estimators. By establishing general upper bounds for the continuous ranked probability score (CRPS) and the worst-case mean squared error (MSE) across the domain, it proposes methods with theoretical guarantees. Comprehensive experiments on both simulated and real data validate the theoretical contributions and demonstrate their practical effectiveness.

**Questions:**

1. What are the limitations of the proposed theoretically guaranteed constraints in real-world applications?
2. Is the proposed method effective when applied to large-scale datasets and models?
3. What challenges or open problems remain in distribution regression currently?
4. Is the proposed upper bound sufficiently tight?

**Ethical Concerns:**

["NO or VERY MINOR ethics concerns only"]

**Final Justification:**

My main concern about this work was the limitations of the theory in the real world, and the author's rebuttal effectively addressed my doubts. Therefore, I ultimately gave a positive score.

**Limitations:**

Yes.

**Paper Formatting Concerns:**

N/A.

**Quality:**

3

**Strengths And Weaknesses:**

Strengths:
1. This paper has clear theoretical guarantees.
2. This paper  is well-organized and easy to understand.
3. The experimental results support the theory.

---

> ### Author Rebuttal · Authors · 2025-07-30
>
> We thank the reviewer for the evaluation and thoughtful questions regarding our work. We appreciate the opportunity to clarify and expand on several important aspects of the paper. Our detailed responses follow below.
>
> ---
>
> **Q1: Limitations of theoretically guaranteed constraints in real-world applications**
>
> Thank you for bringing this up. In the revision, we have added a dedicated discussion on the limitations of our theoretically motivated constraints in real-world applications. In particular, we note that structural assumptions such as monotonicity or bounded variation may not hold exactly in practice, and we discuss how misspecification may affect performance. We also comment on the scalability of certain constraint classes and potential robustness issues in the presence of noise or distributional shifts.
>
> ---
>
> **Q2: Scalability of the proposed method to large-scale datasets and models**
>
> Thank you for pointing this out. We confirm that the proposed method is scalable. While our estimator computes a separate projection $\widehat{F}(t)$ for each threshold $t$, these projections can be evaluated in parallel. Moreover, the structural constraints we consider, such as isotonicity, trend filtering, and neural network-based function classes, are individually scalable and well supported by existing optimization tools. In particular, our neural network implementation leverages architectures commonly used in large-scale nonparametric regression, ensuring practical feasibility in high-dimensional settings.
>
> ---
>
> **Q3: Challenges and open problems in distribution regression**
>
> Thank you for the question. An open problem in distributional regression is how to extend the proposed framework to the case of multivariate responses. This remains a challenging setting due to the lack of a natural ordering in multiple dimensions and the need to estimate joint conditional distributions. While this extension is beyond the scope of the current paper, we agree it is an important direction for future research. We have added a note about this in the discussion of limitations in the revised version.
>
> ---
>
> **Q4: Tightness of the proposed upper bound**
>
> Thank you for the question. Theorems 1--3 are general results that apply to a wide range of constraint sets, both convex and non-convex, and are not tailored to any specific estimation problem. For this reason, we do not provide lower bounds at the general level.
>
> However, in specific examples such as isotonic regression and total variation constraints, the convergence rates we obtain using these general theorems match known minimax rates for mean estimation under Gaussian noise. In this sense, the bounds are optimal for these concrete examples.
>
> To better illustrate this, we have added a dedicated discussion in the revision and created Table~2, which summarizes the convergence rates and key assumptions across our proposed estimators (including isotonic, trend filtering, and Dense ReLU networks) alongside relevant prior work.
>
> Due to space constraints in this response, we briefly comment on the new discussion we incorporated.
>
> For instance, our isotonic result and Theorem 3.3 of Mösching and Dümbgen [44] are compared. Translating their setup into our notation, their quantity $F_{X_i}(t) := \mathbb{P}(Y_i \leq t \mid X_i)$ corresponds to our $F_i^{\ast }(t) := \mathbb{P}(y_i \leq t)$. The monotonicity assumption on $x \mapsto F_x(t)$ aligns with our use of an isotonic constraint on the sequence $F_1^{\ast }(t) \leq \cdots \leq F_n^{\ast }(t)$. However, our result does not assume Hölder continuity for the map $x \mapsto F_x(t)$, or dense covariates in its domain. Despite these weaker requirements, we achieve a faster rate of $n^{-2/3}$ (up to logarithmic factors) for both average CRPS and worst-case MSE, compared to their $n^{-2\alpha/(2\alpha+1)}$ rate for the worst-case MSE, where $\alpha \in (0,1]$.
>
> We compared our Dense ReLU estimator with DRF [10] and EnG [57], two methods explicitly designed for conditional distribution estimation.
>
> DRF estimates the conditional CDF using forest-based weights to form a weighted empirical distribution. EnG instead models $Y = g^{\ast }(X, \varepsilon)$, with $\varepsilon \sim \text{Unif}[0,1]$, and trains a neural network via energy score minimization; the estimated CDF is obtained by sampling from the learned model. Translating their notation into our notation, both methods assume that $G^{\ast }(x, t) := \mathbb{P}(Y \leq t \mid X = x)$ is Lipschitz in $x$ for each fixed $t$, and that the covariate density is bounded away from zero and infinity.
>
> Our method instead assumes $G^{\ast }(\cdot, t) \in \mathcal{H}(l, \mathcal{P})$, a broader class that includes the Lipschitz case ($\mathcal{H}(1, \\{(1, d_0)\\})$) and mitigates the curse of dimensionality, which typically affects Lipschitz-based methods in high-dimensional settings. Moreover, we do not require assumptions on the covariate density. While DRF and EnG show convergence in probability at fixed $t$, our method yields high-probability bounds on the worst-case MSE uniformly over $t$, with explicit rate $\frac{\log n}{n} + \phi_n \log^4 n$.
>
>
> ---
> ***Table 2***
> Below, we present a summary of our convergence rates compared to those in the literature.
>
> ### **Isotonic**
>
> | Work         | Method        | Assumptions                                                   | Convergence Rate                          |
> |--------------|---------------|----------------------------------------------------------------|-------------------------------------------|
> | Prior Work   | [27]          | $K_t$ monotone cone, $F^*(t)\in K_t$                          | $o(1)$                                    |
> | Prior Work   | [44]          | $K_t$ monotone cone, $F^*(t)\in K_t$, dense design, Hölder    | $n^{-2\alpha/(2\alpha+1)},\alpha\le1$ (Worst MSE)    |
> | This Work    | General rate  | $K_t$ monotone cone, $F^*(t)\in K_t$                          | $n^{-2/3}$ (CRPS, Worst MSE)              |
> | This Work    | Fast rate     | $K_t$ monotone cone, $F^*(t)\in K_t$, with few strict increases, $k(t)\ll n$                             | $\frac{1 + k(t)}{n} \log \left( \frac{en}{1 + k(t)} \right)$ (CRPS) |
>
> ### **Trend Filtering**
>
> | Work         | Method        | Assumptions                                                                 | Convergence Rate                          |
> |--------------|---------------|------------------------------------------------------------------------------|-------------------------------------------|
> | Prior Work   | [41,61,23]    | $F_r := \\{ \theta \in \mathbb{R}^n : \mathrm{TV}^{(r)}(\theta) \leq V \\}$, $f \in F_r$ with $f$ the regression function | $\frac{V^{2/(2r+1)}}{n^{2r/(2r+1)}}$ (MSE) |
> | Prior Work   | [23]          | Sparse $D^{(r)}f$, min segment length                                       | $\\{\max{\frac{V^*}{n^{r-1}}, 1} \\}\cdot \frac{s+1}{n} \log\left(\frac{en}{s+1}\right)$ (MSE) |
> | This Work    | General rate  | $K_t := \\{ \theta \in \mathbb{R}^n : \mathrm{TV}^{(r)}(\theta) \leq V_t \\}$, $F^*(t) \in K_t$ | $\frac{V^{2/(2r+1)}}{n^{2r/(2r+1)}} + \frac{\log n}{n}$ (CRPS, Worst MSE) |
> | This Work    | Fast rate     | Sparse $D^{(r)}F^*(t)$, min segment length                                 | $\\{\max{\frac{V^*}{n^{r-1}}, 1}\\} \cdot \frac{s+1}{n} \log\left(\frac{en}{s+1}\right)$ (CRPS) |
>
> ### **Dense ReLU Networks**
>
> | Work         | Method        | Assumptions                                                                                       | Convergence Rate                          |
> |--------------|---------------|----------------------------------------------------------------------------------------------------|-------------------------------------------|
> | Prior Work   | [36]          | Regression $y_i = f(x_i) + \epsilon_i$, $f \in \mathcal{H}(l, \mathcal{P})$                        | $\phi_n$ (MSE)                            |
> | Prior Work   | [10]          | $F_i^{\ast }(t) = G^{\ast }(x_i, t) = \mathbb{P}(y_i \leq t \mid x_i)$, $G^{\ast }(\cdot, t) \in \mathcal{H}(1, \\{(1,d_0)\\})$ Forest-Weighted empirical CDF | $o(1)$                                    |
> | Prior Work   | [57]          | $F_i^{\ast }(t) = G^{\ast }(x_i, t) = \mathbb{P}(y_i \leq t \mid x_i)$, $G^{\ast }(\cdot, t) \in \mathcal{H}(1, \\{(1,d_0)\\})$ Neural Net-based sampling                                             | $o(1)$                                    |
> | This Work    | General rate  | $F_i^{\ast}(t) = G^{\ast}(x_i, t)$, $G^{\ast}(\cdot, t) \in \mathcal{H}(l, \mathcal{P})$                          | $\frac{\log n}{n} + \phi_n \log^4 n$ (CRPS, Worst MSE) |
>
>
> ---
> ***References:***
>
> [10] Domagoj Cevid, Loris Michel, Jeffrey Näf, Peter Bühlmann, and Nicolai Meinshausen. Distributional random forests: Heterogeneity adjustment and multivariate distributional regression. Journal of Machine Learning Research, 23(333):1–79, 2022.
>
> [23] Adityanand Guntuboyina, Donovan Lieu, Sabyasachi Chatterjee, and Bodhisattva Sen. Adaptive risk bounds in univariate total variation denoising and trend filtering. The Annals of Statistics, 48:205–229, 2020.
>
> [27] Alexander Henzi, Johanna F Ziegel, and Tilmann Gneiting. Isotonic distributional regression. Journal of the Royal Statistical Society Series B: Statistical Methodology, 83(5):963–993, 2021.
>
> [36] Michael Kohler and Sophie Langer. On the rate of convergence of fully connected deep neural network regression estimates. The Annals of Statistics, 49(4):2231–2249, 2021.
>
> [41] Enno Mammen and Sara Van De Geer. Locally adaptive regression splines. The Annals of Statistics, 25(1):387–413, 1997.
>
> [57] Xinwei Shen and Nicolai Meinshausen. Engression: extrapolation through the lens of distributional regression. Journal of the Royal Statistical Society Series B: Statistical Methodology, page qkae108, 2024.
>
> [61] Ryan J Tibshirani. Adaptive piecewise polynomial estimation via trend filtering. 2014

---

> > ### Comment · Reviewer_LnJT · 2025-08-03
> >
> > Thanks for the author's rebuttal. I will maintain my positive score.

---

### Official Review · Reviewer_iGJs · 2025-07-01

**Clarity:** 3
**Significance:** 2
**Originality:** 3
**Rating:** 4
**Confidence:** 4

**Summary:**

This paper presents a unified theoretical and algorithmic framework for nonparametric distributional regression under both convex and non-convex constraints. It analyzes risk bounds based on Continuous Ranked Probability Score (CRPS) and worst-case MSE, with applications to isotonic regression, trend filtering, and deep neural networks. The authors prove convergence rates and support their claims through simulation and real-data experiments.

**Questions:**

1. While the paper proposes a unified framework for risk bounds in distributional regression, it is not always clear which results represent new theoretical advances versus careful syntheses or extensions of existing work. For instance, the projection-based estimator (Equation 3) and its analysis under convex constraints closely follow the structure of M-estimation with shape constraints, and the CRPS risk bounds leverage tools like Gaussian complexity and entropy bounds that are standard in nonparametric theory. Similarly, the neural network analysis builds directly on Kohler and Langer (2021) with minimal modification to the underlying techniques. The authors should clarify: Which lemmas or theorems are original? Which contributions extend prior work (and how)?
2. Theorem 3's current form of the bound may be of limited practical use. In particular, the deviation bound involves a threshold eta and achieves small probability only if eta is of order sqrt{n}, which would imply an unacceptably large estimation error. This weakens the utility of the bound, especially compared to existing uniform concentration results in empirical process theory.
3. Corollary 4 presents convergence rates for neural network-based distributional regression. However, the result depends on Assumption 1 and specific architecture-related conditions that are only described in Appendix C and F.9. Unfortunately, these assumptions are not summarized or even mentioned in the main text, making it difficult to evaluate the generality or strength of the result.

**Ethical Concerns:**

["NO or VERY MINOR ethics concerns only"]

**Final Justification:**

The referee appreciates the authors' efforts. The original concerns are still valid. It would take efforts to address them. The paper is solid and has its own merits. Nevertheless, the referee retains the same positive score for this manuscript.

**Limitations:**

The authors did not discuss the limitations or potential negative societal impacts of their work. While the technical contributions are clearly presented, the paper omits critical reflections on the following:
1. The methodology assumes prior knowledge of structural constraints (e.g., monotonicity, bounded variation, or function class for neural nets), which may not hold in practice. The authors should acknowledge this and discuss how misspecification could affect estimation accuracy.
2. Although projection-based estimators are efficient in low dimensions, scalability—particularly for trend filtering and neural networks—can be a concern in large-scale or high-dimensional settings. This should be addressed.
3. The method is primarily validated on simulations and a few benchmark datasets. Real-world data often involves missing values, measurement errors, or distribution shifts. A discussion of robustness in such settings is missing.

**Paper Formatting Concerns:**

NA.

**Quality:**

2

**Strengths And Weaknesses:**

Strengths:
1. The paper presents non-asymptotic risk bounds for distributional regression under convex and non-convex structural constraints. This includes uniform bounds over the domain.
2. The framework can add a range of constraints including monotonicity and bounded variation, and extends to neural networks.
3. The bounds are derived using projection-based estimators and Gaussian complexity tools, and they generalize prior results (e.g., isotonic distributional regression).

Weaknesses
(a) While the paper proposes a general projection-based estimator, this method essentially reduces to projection onto known constraint sets (e.g., isotonic cones or TV balls), which are well-studied in nonparametric regression. The core estimator is structurally straightforward and is similar to existing techniques (e.g., M-estimators under constraints).
(b) The theoretical analysis heavily relies on known tools—e.g., Gaussian complexity and covering numbers for the convex case, and peeling arguments for the non-convex case.
(c) Theorem 3 is quite problematic. It seems that eta should be in the magnitude of sqrt{n} to ensure that the probability to be small. This implies that the error should be quite significant. Can you improve it?

---

> ### Author Rebuttal · Authors · 2025-07-30
>
> We thank the reviewer for careful reading with valuable feedback. Below we would like to address the comments and questions raised by the reviewer point-by-point.
>
> ---
>
> **Q1 and Weakness (a) and (b): Simplicity of projection-based estimator and use of standard theoretical tools**
>
> Thank you for the thoughtful comments. We agree that projection-based estimators onto known constraint sets, such as isotonic cones or total variation balls, are well-studied in nonparametric regression under squared loss. Our contribution lies not in proposing a new estimator form, but in adapting this framework to the setting of CDF estimation.
>
> In particular, Theorems 1–3 build on classical techniques from regression, where projection estimators have been studied in both convex (e.g., [5, 14]) and non-convex (e.g., [36]) settings. These prior results focus on pointwise mean squared error control and were not developed for CDF estimation, which requires uniform control over all $t \in \mathbb{R}$. To our knowledge, our results provide the first such uniform risk bounds for projection-based estimators in the distributional regression setting.
>
> While the analysis uses standard tools, such as Gaussian complexity bounds in the convex case and peeling arguments in the non-convex case, the novelty lies in the adaptation of these techniques to the CDF estimation setting and in the derivation of uniform-in-$t$ risk bounds.
>
> Moreover, the general theorems we establish allow us to analyze concrete examples, such as isotonic regression and total variation denoising, where we show that the resulting rates match known minimax rates for squared loss under Gaussian noise. In this sense, the bounds we obtain are optimal for these structured cases, but now in the more general context of distributional estimation. For a summary comparing our convergence rates with those in the literature, see Table~2 below, which we have incorporated into the revised manuscript.
>
> ---
>
> **Weakness (c) and Q2: Concerns about Theorem 3 and scaling of $\eta$**
>
> Thank you for highlighting this aspect. We clarify that the appearance of the $\sqrt{n}$ factor in Theorem 3 is not due to looseness in the proof. Rather, it reflects the fact that Theorem 3 controls the $\ell_2$ error over $n$ coordinates, i.e., $\vert\vert\widehat{F}(t)-F^*(t)\vert\vert_2$. In applications, the relevant quantity is the average per-sample error, which is normalized by $\sqrt{n}$.
>
> Moreover, the $\sqrt{n}$ term appears inside the event in the probability bound as a multiplier of the approximation error term $\sup_t \vert\vert F^{\ast }(t) - G(t)\vert\vert_\infty$. This quantity measures how well the true distribution vector $F^{\ast }(t)$ can be approximated by elements in the constraint set $K_t$. A similar approximation term appears in related results in the literature, such as Kohler and Langer [36]. After normalizing by $\sqrt{n}$, the final bound depends on both the approximation error and $\eta/\sqrt{n}$. To ensure the bound holds with high probability, it suffices to choose $\eta$ of order $\sqrt{\log n} + \sqrt{n} \cdot \sup_t \vert\vert F^*(t) - G(t)\vert\vert_\infty$.
>
> This behavior is confirmed in Corollary 4 for our Dense ReLU network example. There, we assume that $G^{\ast }(\cdot, t) \in \mathcal{H}(l, \mathcal{P})$ where $F_i^{\ast }(t) = G^{\ast }(x_i, t)$, see Assumption 1 in Appendix C, and define the constraint set
>
> $$
> K_t = \\{ \theta \in \mathbb{R}^n : \theta_i = f(x_i), \text{ for some } f \in \mathcal{F}(L, \nu) \\},
> $$
>
> where $\mathcal{F}(L, \nu)$ is a class of ReLU networks with appropriate number of Layers and Neurons. In this case, the approximation error scales as $\phi_n = \max_{(p, M) \in \mathcal{P}} n^{-p/(2p + M)}$, and the final rate becomes
>
> $$
> \sup_{t \in \mathbb{R}} \frac{1}{n} \sum_{i=1}^n\left(\widehat{F}_i(t)-F_i^*(t)\right)^2 = O_P\left(\frac{\log n}{n} + \phi_n \log^4 n \right).
> $$
>
> Thus, the resulting rate is meaningful and matches known convergence rates for mean estimation under Gaussian noise using Dense ReLU networks. This shows that our bound is both general and sharp in concrete, practically relevant settings.
>
> ---
>
> **Q3: Assumptions behind Corollary 4 and missing summaries in main text**
>
> Thank you for pointing this out. In the revision, we have moved the key elements of Assumption 1 and the architecture-related conditions from the Appendix to the main text to improve clarity and allow readers to better assess the scope and generality of Corollary 4.
>
> ---
>
> **Limitations: Assumptions, scalability, and robustness**
>
> Thank you for the helpful suggestion. In the revision, we have added a dedicated discussion of limitations at the end of the main text. This includes comments on structural constraint assumptions, scalability, and robustness to real-world data issues.
>
>
> ---
> ***Table 2***
> Below, we present a summary of our convergence rates compared to those in the literature.
>
> ### **Isotonic**
>
> | Work         | Method        | Assumptions                                                   | Convergence Rate                          |
> |--------------|---------------|----------------------------------------------------------------|-------------------------------------------|
> | Prior Work   | [27]          | $K_t$ monotone cone, $F^*(t)\in K_t$                          | $o(1)$                                    |
> | Prior Work   | [44]          | $K_t$ monotone cone, $F^*(t)\in K_t$, dense design, Hölder    | $n^{-2\alpha/(2\alpha+1)}, \alpha\le 1$ (Worst MSE)    |
> | This Work    | General rate  | $K_t$ monotone cone, $F^*(t)\in K_t$                          | $n^{-2/3}$ (CRPS, Worst MSE)              |
> | This Work    | Fast rate     | $K_t$ monotone cone, $F^*(t)\in K_t$, with few strict increases, $k(t)\ll n$                             | $\frac{1 + k(t)}{n} \log \left( \frac{en}{1 + k(t)} \right)$ (CRPS) |
>
> ### **Trend Filtering**
>
> | Work         | Method        | Assumptions                                                                 | Convergence Rate                          |
> |--------------|---------------|------------------------------------------------------------------------------|-------------------------------------------|
> | Prior Work   | [41,61,23]    | $F_r := \\{ \theta \in \mathbb{R}^n : \mathrm{TV}^{(r)}(\theta) \leq V \\}$, $f \in F_r$ with $f$ the regression function | $\frac{V^{2/(2r+1)}}{n^{2r/(2r+1)}}$ (MSE) |
> | Prior Work   | [23]          | Sparse $D^{(r)}f$, min segment length                                       | $\\{\max{\frac{V^*}{n^{r-1}}, 1} \\}\cdot \frac{s+1}{n} \log\left(\frac{en}{s+1}\right)$ (MSE) |
> | This Work    | General rate  | $K_t := \\{ \theta \in \mathbb{R}^n : \mathrm{TV}^{(r)}(\theta) \leq V_t \\}$, $F^*(t) \in K_t$ | $\frac{V^{2/(2r+1)}}{n^{2r/(2r+1)}} + \frac{\log n}{n}$ (CRPS, Worst MSE) |
> | This Work    | Fast rate     | Sparse $D^{(r)}F^*(t)$, min segment length                                 | $\\{\max{\frac{V^*}{n^{r-1}}, 1}\\} \cdot \frac{s+1}{n} \log\left(\frac{en}{s+1}\right)$ (CRPS) |
>
> ### **Dense ReLU Networks**
>
> | Work         | Method        | Assumptions                                                                                       | Convergence Rate                          |
> |--------------|---------------|----------------------------------------------------------------------------------------------------|-------------------------------------------|
> | Prior Work   | [36]          | Regression $y_i = f(x_i) + \epsilon_i$, $f \in \mathcal{H}(l, \mathcal{P})$                        | $\phi_n$ (MSE)                            |
> | Prior Work   | [10]          | $F_i^{\ast }(t) = G^{\ast }(x_i, t) = \mathbb{P}(y_i \leq t \mid x_i)$, $G^{\ast }(\cdot, t) \in \mathcal{H}(1, \\{(1,d_0)\\})$ Forest-Weighted empirical CDF | $o(1)$                                    |
> | Prior Work   | [57]          | $F_i^{\ast }(t) = G^{\ast }(x_i, t) = \mathbb{P}(y_i \leq t \mid x_i)$, $G^{\ast }(\cdot, t) \in \mathcal{H}(1, \\{(1,d_0)\\})$ Neural Net-based sampling                                             | $o(1)$                                    |
> | This Work    | General rate  | $F_i^{\ast}(t) = G^{\ast}(x_i, t)$, $G^{\ast}(\cdot, t) \in \mathcal{H}(l, \mathcal{P})$                          | $\frac{\log n}{n} + \phi_n \log^4 n$ (CRPS, Worst MSE) |
>
>
> ---
> ***References:***
>
> [5] Pierre C Bellec. Sharp oracle inequalities for least squares estimators in shape restricted regression. The Annals of Statistics, 46(2):745–780, 2018.
>
> [10] Domagoj Cevid, Loris Michel, Jeffrey Näf, Peter Bühlmann, and Nicolai Meinshausen. Distributional random forests: Heterogeneity adjustment and multivariate distributional regression. Journal of Machine Learning Research, 23(333):1–79, 2022.
>
> [14] Sourav Chatterjee. Matrix estimation by universal singular value thresholding. 2015.
>
> [23] Adityanand Guntuboyina, Donovan Lieu, Sabyasachi Chatterjee, and Bodhisattva Sen. Adaptive risk bounds in univariate total variation denoising and trend filtering. The Annals of Statistics, 48:205–229, 2020.
>
> [27] Alexander Henzi, Johanna F Ziegel, and Tilmann Gneiting. Isotonic distributional regression. Journal of the Royal Statistical Society Series B: Statistical Methodology, 83(5):963–993, 2021.
>
> [36] Michael Kohler and Sophie Langer. On the rate of convergence of fully connected deep neural network regression estimates. The Annals of Statistics, 49(4):2231–2249, 2021.
>
> [41] Enno Mammen and Sara Van De Geer. Locally adaptive regression splines. The Annals of Statistics, 25(1):387–413, 1997.
>
> [57] Xinwei Shen and Nicolai Meinshausen. Engression: extrapolation through the lens of distributional regression. Journal of the Royal Statistical Society Series B: Statistical Methodology, page qkae108, 2024.
>
> [61] Ryan J Tibshirani. Adaptive piecewise polynomial estimation via trend filtering. 2014

---

> > ### Comment · Reviewer_iGJs · 2025-08-05
> >
> > The referee appreciates the authors' efforts. My concerns
> >  ``(a) While the paper proposes a general projection-based estimator, this method essentially reduces to projection onto known constraint sets (e.g., isotonic cones or TV balls), which are well-studied in nonparametric regression. The core estimator is structurally straightforward and is similar to existing techniques (e.g., M-estimators under constraints). (b) The theoretical analysis heavily relies on known tools—e.g., Gaussian complexity and covering numbers for the convex case, and peeling arguments for the non-convex case. (c) Theorem 3 is quite problematic. It seems that eta should be in the magnitude of sqrt{n} to ensure that the probability to be small. This implies that the error should be quite significant. Can you improve it?"
> > are still valid. It would take efforts to address them. Nevertheless, the referee retains the same positive score for this manuscript.

---

### Official Review · Reviewer_c7JP · 2025-07-04

**Clarity:** 2
**Significance:** 3
**Originality:** 2
**Rating:** 4
**Confidence:** 4

**Summary:**

This paper studies the problem of estimating multiple cumulative distribution functions (CDFs) using samples from those distributions under structural constraints. Specifically, given $n$ points sampled from $n$ distributions, the authors propose a distributional regression method which minimizes the averaged CRPS scores between the estimating functions and the one-sample empirical CDFs under functional constraints. They show that the solution corresponds to the $\ell_2$ projection of the empirical CDF values to the constraint set. Upper bounds of the CRPS and mean squared error (MSE) between the estimator and the true CDFs are proved under various settings of the constraints (convex and non-convex, containing the true CDFs and not containing the true CDFs). They also apply their method to the specific case of isotonic regression, trend filtering, and dense ReLU network by identifying the corresponding constraints to establish risk upper bounds. Finally, they compare the performance of their distributional regression method with other existing methods on simulated and real-world datasets.

**Questions:**

1. It would be better to explicitly summarize the existing bounds to compare to the proved bounds, preferably in a table. How does the general results in Theorem 1, 2, and 3 compare to existing results of CDF estimation? For Theorem 1, it seems that the results follow from Theorem A.1 in [23]. Then, what is the technical challenges of this paper considering the results in [23]? What improvements have been made comparing to the results in [23]? I would appreciate more discussion on the comparison to [23] in the paper.

2. Are there lower bounds for the considered problem? Without lower bounds, it is unclear how tight the upper bounds are.

3. I do not find explicit statement that $K_t$ is convex in Section 3. Please emphasize on this point in Section 3.
I am curious about the convexity of $K_t$ in eq. (12). Is it because TV${}^{(r)}$ satisfies triangle inequality? If so, please include a proof or a reference.
Besides, I would recommend the authors to explain the benefits of convexity in the proofs and how the proof for convex constraints extends to nonconvex constraints.

4. For Theorem 2, $N(\epsilon, (K-K)\cap B_\epsilon(0))=1$. It seems from the proof that this quantity should be $N(\epsilon, (K-K)\cap B_\eta(0))$. Similar issue exists for Theorem 3.

5. In the conclusion, the authors claim that "structured constraints" leads to "improved estimation accuracy", but it seems that those results under the listed structured constraints are just direct corollaries of the general results. Then, why are there improved estimation accuracy?

6. For the experiments, please explain more on how to apply the proposed method to test data. For training, we can just estimate CDFs of those training variables. Then, how do we use the trained functions in test data?

7. There are some typos and unclear parts. (1) In line 42, since $F^*(t)\in\mathbb{R}^n$, we should have $K_t\subset\mathbb{R}^n$.

   (2) In line 132, it should be "for some fixed".

   (3) Do we need the assumption that $F^*(t)\in K_t$ for all $t$ for Theorem 1, Corollary 1, and Theorem 2? Is so, please make this assumption explicit in the statement whenever it is needed.

   (4) Does $C$ in line 132 depends on $\Omega$ and other variables? If so, please make this explicit.

   (5) In line 153 and 158, what is $K-K$?

   (6) Please explain the notation $O_{\mathbb{P}}$.

   (7) In line 245, a wrong spelling "unctions".

   (8) Please explain the notation $\mathcal{P}$ in line 253.

   (9) Please add a reference to the "hierarchical composition class" in line 257.

**Ethical Concerns:**

["NO or VERY MINOR ethics concerns only"]

**Final Justification:**

I appreciate it that the authors could revise the paper in response to my suggestions and questions. As more comparisons with existing work on CDF estimation and explanations of the technical challenges compared to the existing theoretical results are needed to revise the current submission, I will maintain my current score.

**Limitations:**

Yes.

**Paper Formatting Concerns:**

No.

**Quality:**

3

**Strengths And Weaknesses:**

I do not check the details of the proofs, but the paper seems to be technically sound. Both theoretical and empirical results of the paper are complete.

The paper is clearly written and well-organized, but it would be better to include some technical overview of the proofs and more details of the experimental setting. See my questions below for details.

The results of the paper are significant for CDF estimation and its downstream application. It allows various function classes through the flexibility in the choice of the constraints and thus considerable potential for application. It would be better to make the comparisons with existing theoretical results explicit by summarizing their bounds. Since lower bounds are missing, it is unclear how tight the results of this paper are. See my questions below for details.

Though the idea of distributional regression based on one-sample empirical CDFs are not new, the authors provide a general and practical recipe for this problem with various application settings. The theoretical work of this paper is based on existing results and standard techniques of covering numbers and uniform convergence.

For other weaknesses, please refer to my questions below.

---

> ### Author Rebuttal · Authors · 2025-07-30
>
> We thank the reviewer for careful reading with valuable feedback. Below we would like to address the comments and questions raised by the reviewer point-by-point.
>
>
> ---
>
> **Q1 & Weakness 2: Comparison and Novelty Relative to Prior Work**
>
> Thank you for all your valuable comments. We expanded and clarified the theoretical comparisons in the revision. Table~2 summarizes the convergence rates and assumptions (partially shown here due to space constraints).
>
> **Bounds**
>
> Due to space constraints in this response, we briefly comment on the new comparisons we have added.
>
> Our isotonic result and Theorem 3.3 of Mösching and Dümbgen [44] are compared. Translating their setup into our notation, their quantity $F_{X_i}(t) := \mathbb{P}(Y_i \leq t \mid X_i)$ corresponds to our $F_{i}^{\ast}(t) := \mathbb{P}(y_i \leq t)$. The monotonicity assumption on $x \mapsto F_x(t)$ aligns with our use of an isotonic constraint on the sequence $F_1^{\ast }(t) \leq \cdots \leq F_n^{\ast }(t)$. However, our result does not assume Hölder continuity for the map $x \mapsto F_x(t)$, or dense covariates in its domain. Despite these weaker requirements, we achieve a faster rate of $n^{-2/3}$ (up to logarithmic factors) for both average CRPS and worst-case MSE, compared to their $n^{-2\alpha/(2\alpha+1)}$ rate for the worst-case MSE, where $\alpha \in (0,1]$.
>
> We compared our Dense ReLU estimator with DRF [10] and EnG [57], two methods explicitly designed for conditional distribution estimation.
>
> DRF estimates the conditional CDF using forest-based weights to form a weighted empirical distribution. EnG instead models $Y = g^{\ast }(X, \varepsilon)$, with $\varepsilon \sim \text{Unif}[0,1]$, and trains a neural network via energy score minimization; the estimated CDF is obtained by sampling from the learned model. Translating their notation into our notation, both methods assume that $G^{\ast }(x, t) := \mathbb{P}(Y \leq t \mid X = x)$ is Lipschitz in $x$ for each fixed $t$, and that the covariate density is bounded away from zero and infinity.
>
> Our method instead assumes $G^{\ast }(\cdot, t) \in \mathcal{H}(l, \mathcal{P})$, a broader class that includes the Lipschitz case ($\mathcal{H}(1, \\{(1, d_0)\\})$) and mitigates the curse of dimensionality, which typically affects Lipschitz-based methods in high-dimensional settings. Moreover, we do not require assumptions on the covariate density. While DRF and EnG show convergence in probability at fixed $t$, our method yields high-probability bounds on the worst-case MSE uniformly over $t$, with explicit rate $\frac{\log n}{n} + \phi_n \log^4 n$.
>
> **Theorem 1-3 in CDF literature:**
>
> Thank you for the question. Theorems 1–3 adapt classical techniques from regression, where projection-based estimators have been studied in both convex (e.g., [5, 14]) and non-convex (e.g., [46]) settings. These prior results focus on controlling mean squared error at a fixed point and were not developed for CDF estimation, which requires uniform control over all $t \in \mathbb{R}$. To our knowledge, our extensions provide the first such results for CDF estimation.
>
> **Clarification regarding Theorem A.1 in [23]:**
>
> Our result in Theorem 1 is conceptually related to Theorem A.1 of [23], but technically distinct. In their setting, the observations are Gaussian $Y = \mu + Z$, with $Z \sim \mathcal{N}(0, \alpha^2 I_n)$. The estimator $\hat{\mu}$ is the Euclidean projection of $Y$ onto a convex set $K$. Their goal is to bound the squared error $\vert\vert \hat{\mu} - \mu\vert\vert_2^2 = t^{\ast }$ in expectation, which is reduced to be controlled with high probability. For this, the "KEY" step is to show that the squared error $t^{\ast }$ concentrates around a deterministic benchmark $t_\mu$, which characterizes the scale of estimation error. This is proved using a peeling argument that combines the strict concavity of an associated process with Gaussian concentration.
>
> Our setting also relies on convex projection, but differs in several important ways. We observe Bernoulli vectors $w(t) = 1_{\\{ y_i \leq t \\}}$ (their $Y$), with sub-Gaussian, heteroscedastic noise $\epsilon(t) = w(t) - F^{\ast }(t)$ (their $Z$), and our target is the CRPS rather than squared error. Despite these differences, our analysis also hinges on addressing the same "KEY" step: showing that the squared $\ell_2$ error $\vert\vert \widehat{F}(t) - F^{\ast }(t)\vert\vert_2^{2}$ (their $t^{\ast }$) concentrates around a complexity-driven benchmark $\eta$ (their $t_\mu$).
>
> However, our proof takes a completely different route. We use (i) a sub-Gaussian concentration inequality for Lipschitz and separately convex functions to control a local empirical process, and (ii) a star-shaped self-normalization argument to extend this control to the full constraint set $K_t$. Because of the differences in observations, loss function, and techniques, our result is not a direct application of theirs.
>
> **Q2: Lower bounds**
>
> Thank you for the question. Theorems 1–3 are general results that apply to a wide range of constraint sets, both convex and non-convex, and are not tailored to any specific estimation problem. For this reason, we do not provide lower bounds at the general level.
>
> However, in specific examples like isotonic regression, or total variation, the rates we obtain using these general theorems match known minimax rates for mean estimation with Gaussian noise. In this sense, the bounds we obtain are optimal for these concrete examples.
>
> **Q3: Convexity discussion**
>
> Thank you for pointing this out.
>
> **Statement of convexity in Section 3:**
>
> We have now added an explicit statement in Section 3.1 clarifying that the sets $K_t \subset \mathbb{R}^n$ are assumed to be convex throughout the section. This assumption was previously implicit, and we agree it is important to make it clear.
>
> **Convexity of set in (12):**
>
> We added a short formal proof of the convexity of the set $K_t$ in Equation (12) in Appendix F.7, which contains the proof of Corollary 3 in the revised manuscript. As you pointed out, the key argument is that $\mathrm{TV}^{(r)}(\theta) = n^{r-1}\\| D^{(r)} \theta\\|_1$ is convex. We copy it here for your convenience:
>
> We start by noting that the set $K := \\{ \theta \in \mathbb{R}^n : \mathrm{TV}^{(r)}(\theta) \leq V \\}$ is convex. Indeed,
> let $\theta^{(1)}, \theta^{(2)} \in K$, and define $\theta^{(\alpha)} := \alpha \theta^{(1)} + (1 - \alpha)\theta^{(2)}$ for $\alpha \in [0,1]$. Then
>
> $\mathrm{TV}^{(r)}(\theta^{(\alpha)}) = n^{r-1} \\| D^{(r)} \theta^{(\alpha)} \\|_1 = n^{r-1} \\| \alpha D^{(r)} \theta^{(1)} + (1 - \alpha) D^{(r)} \theta^{(2)} \\|_1$.
>
> By convexity of the $\ell_1$-norm,
>
> $\mathrm{TV}^{(r)}(\theta^{(\alpha)}) \leq \alpha \mathrm{TV}^{(r)}(\theta^{(1)}) + (1 - \alpha)\mathrm{TV}^{(r)}(\theta^{(2)}) \leq V$.
>
> Hence, $\theta^{(\alpha)} \in K_t$, proving that $K$ is convex.
>
>
> **Benefits of convexity and extensions to nonconvex settings:**
>
> Thank you for this helpful comment. Convexity plays a central role in both defining the estimator and analyzing its error. In the convex case (Theorems 1 and 2), the convexity guarantees that interpolation between the estimator and the truth remains within the constraint set. This allows us to apply a basic inequality and focus the analysis on a neighborhood around $F^*(t)$, leading to clean bounds based on sub-Gaussian complexity in Theorem 1 and on entropy in Theorem 2.
>
> In the nonconvex case (Theorem 3), these tools are not available. The proof instead uses a nearby point $G(t) \in K_t$ and separates the total error into two parts: the estimation error around $G(t)$, and the approximation error between $G(t)$ and $F^*(t)$. The estimation is then controlled using peeling. The final bound includes an extra approximation term that does not appear in the convex case.
>
> We have added a paragraph in the revision to clarify these points.
>
> **Q6: Applying the Method to Test Data**
>
> Thank you for the question. We have clarified this point in the revision. For the Dense ReLU estimator in Section 4.2, the network is trained on the training set, and predictions on test data are obtained by evaluating the fitted model at the test covariates to estimate the conditional CDF at each threshold. These are then compared against the indicators of the responses.
>
> **Q5: Structured Constraints and Accuracy**
>
> Thank you for the helpful comment. We have clarified the conclusion, which now reads: “In different settings, when structured assumptions are met, our results suggest that estimation accuracy can be achieved. This is the case, for instance, in isotonic regression if we assume that we are estimating a monotonic signal, and similarly for total variation.”
>
> **Q4 and Q7: Typos**
>
> Thank you for these detailed comments. We have corrected the typos, clarified the notation, and made the relevant assumptions explicit in the revised version.
>
> **Q4 and Q7: Table 2**
>
> Below, we present Table 2 partially, showing only the summary of results for the Isotonic case due to space constraints.
>
> ### **Isotonic**
>
> | Work         | Method        | Assumptions                                                   | Convergence Rate                          |
> |--------------|---------------|----------------------------------------------------------------|-------------------------------------------|
> | Prior Work   | [27]          | $K_t$ monotone cone, $F^*(t)\in K_t$                          | $o(1)$                                    |
> | Prior Work   | [44]          | $K_t$ monotone cone, $F^*(t)\in K_t$, dense design, Hölder    | $n^{-2\alpha/(2\alpha+1)}, \alpha\le 1$ (Worst MSE)    |
> | This Work    | General rate  | $K_t$ monotone cone, $F^*(t)\in K_t$                          | $n^{-2/3}$ (CRPS, Worst MSE)              |
> | This Work    | Fast rate     | Few number of strict increases, $k(t)\ll n$                                         |$\frac{1 + k(t)}{n} \log \left( \frac{en}{1 + k(t)} \right)$ (CRPS)             |

---

> > ### Comment · Reviewer_c7JP · 2025-08-08
> >
> > Thank you for your detailed feedback.
> >
> > **Q1.**
> > Please make the discussion on the comparison to Theorem A.1 in [23] explicit in the paper. In the current version, by line 129, Theorem 1 is "a consequence of a modified version of Theorem A.1", and in line 761, Theorem 5 is captioned with Theorem A.1 in [23], which leaves the reader the impression that Theorem 5 is just a restatement of Theorem A.1 for completeness.
> >
> > **Q2.**
> > We can always consider the minimax rate for an estimation problem. For the problem considered in this paper, given observed responses, we estimate their CDFs such that they lie in some constraint set to minimize risks like CRPS or $L_{\infty}$-norm. There is a well-defined minimax rate for it. For example, a researcher could start from the case where the constraint set is convex. I understand that this is beyond the scope of this paper, but it is a well-defined theoretical problem worth studying.
> >
> > Besides, for those specific problems you mentioned, could you refer to those existing minimax rates in the revisions? That will at least provide readers some sense on the quality of the results. However, since those rates are for mean estimation, I do not believe that they immediately translate to minimax rates on CDF estimation.

---

> > > ### Author Response · Authors · 2025-08-08
> > >
> > > Thank you very much for the detailed and constructive feedback. We are grateful for your suggestions and have addressed each point in the revised version of the manuscript. Below, please find our responses.
> > >
> > > ---
> > >
> > > ### **Response to Q1**
> > >
> > > We agree that a more explicit comparison with Theorem A.1 of [23] is valuable for clarifying the contribution of our Theorem 1. We have added the following remark to the revised version of the paper:
> > >
> > > >Our result in Theorem 1 is conceptually related to Theorem A.1 of [23], but technically distinct. In their setting, the observations are Gaussian $Y = \mu + Z$, with $Z \sim \mathcal{N}(0, \alpha^2 I_n)$. The estimator $\hat{\mu}$ is the Euclidean projection of $Y$ onto a convex set $K$. Their goal is to bound the squared error $\vert\vert \hat{\mu} - \mu\vert\vert_2^2 = t^{\star}$ in expectation, which is then controlled with high probability. For this, the **key** step is to show that the squared error $t^{\star}$ concentrates around a deterministic benchmark $t_\mu$, which characterizes the scale of the estimation error. This is proved using a peeling argument that combines the strict concavity of an associated process with Gaussian concentration.
> > >
> > > >Our setting also relies on convex projection, but differs in several important ways. We observe Bernoulli vectors $w(t)$ (their $Y$), with sub-Gaussian, heteroscedastic noise $\epsilon(t) = w(t) - F^{\star}(t)$ (their $Z$), and our target is the CRPS rather than squared error. Despite these differences, our analysis also hinges on addressing the same **key** step: showing that the squared $\ell_2$ error $\vert\vert \widehat{F}(t) - F^{\star}(t)\vert\vert_2^2$ (their $t^{\star}$) concentrates around a complexity-driven benchmark $\eta$ (their $t_\mu$).
> > >
> > > >However, our proof takes a completely different route. We use (i) a sub-Gaussian concentration inequality for Lipschitz and separately convex functions to control a local empirical process, and (ii) a star-shaped self-normalization argument to extend this control to the full constraint set $K_t$. Because of the differences in observations, loss function, and techniques, our result is not a direct application of theirs.
> > >
> > > ---
> > >
> > > ### **Response to Q2**
> > >
> > > We agree that understanding the minimax properties of the proposed estimators is of both theoretical and practical interest. While the current work does not establish general minimax lower bounds for CDF estimation, we have now added clarifying remarks in the revised version to guide readers on how the obtained rates compare to known minimax results in related settings.
> > >
> > > In particular, after Corollary 3, we added the following regarding the Trend Filtering example:
> > >
> > > >The convergence rate in Corollary 3, $V^{1/(2r+1)} n^{-2r/(2r+1)}$, matches the minimax rate for estimating a mean function in a one-dimensional nonparametric regression setting with Gaussian errors and in the class of signals $\theta$ with $\mathrm{TV}^{(r)}(\theta) \leq V$; see Section 5.1 in [61]. This alignment suggests that our estimator may also be minimax-optimal (up to logarithmic factors) for the distributional regression setting with CDF-valued outputs. However, formally establishing such a lower bound for CRPS or squared error in CDF estimation would require technical extensions of known minimax lower bound arguments from mean estimation to the distributional setting, which involves discrete, heteroscedastic observations and a different loss function. We leave this direction to future work.
> > >
> > > A similar remark has also been added in the revised version following Corollary 2 to highlight the rate in the Isotonic example.
> > >
> > > ---
> > >
> > > ### **References**
> > >
> > > [61] Ryan J. Tibshirani. *Adaptive piecewise polynomial estimation via trend filtering*. 2014.

---

> > > > ### Comment · Reviewer_c7JP · 2025-08-09
> > > >
> > > > Thank you for your response. I will maintain my positive score.

---

### Note · Authors · 2025-08-11

We sincerely thank all reviewers for their constructive and thoughtful feedback throughout the review process. Your comments have been invaluable in clarifying our presentation, strengthening the theoretical comparisons, and improving the exposition of our results. We will incorporate all the changes and clarifications we committed to in our rebuttal and discussion responses.

We also appreciate the careful reading of the manuscript and the recognition of the contributions, as well as the effort to maintain fair and consistent assessments. Your feedback has not only helped us improve the current version but will also guide future extensions of this work.

---

### Decision · Program_Chairs · 2025-09-17

**Decision:**

Accept (poster)

**Comment:**

The paper develops risk bounds for distributional regression (estimating conditional CDFs) under structural constraints. It gives general upper bounds on CRPS and worst-case MSE under convex constraints (with applications to isotonic and trend filtering) and also provides a general upper bound for certain non-convex constraints, illustrated with neural network estimators. Experiments support the theory.

Reviewers highlighted that it is
* clearly written and well-organised;
* presents significant results addressing an important problem; and
* is theoretically sound.

Reviewers asked for clarifications on the novelty relative to prior work and on the scaling and tightness of the bounds. The authors’ detailed responses addressed these points satisfactorily. I concur with the reviewers and recommend acceptance.